# SORA: Free Second-Order Attacks in Fast Adversarial Training

**Mazdak Teymourian** [1] [*]  **Ramtin Moslemi** [1] [*]  **Farzan Rahmani** [1]  **Mohammad Hossein Rohban** [1]

## Abstract

Adversarial Training (AT) is a leading defense against adversarial examples but often suffers from *Catastrophic Overfitting* (CO) in efficient single-step variants, where robustness to multi-step attacks collapses despite high single-step performance. We address this failure mode with two contributions. First, we formalize *Epsilon Overfitting* (EO), a perspective in which fixed perturbation magnitudes and directions exacerbate CO, and show that introducing perturbation variability significantly improves robust generalization across different architectures and datasets. Second, we propose **PertAlign** (Perturbation Alignment), a theoretically grounded, computationally negligible metric that predicts CO onset by measuring gradient alignment across attack stages. Leveraging these insights, we introduce **SORA**, an adaptive step-size AT method that dynamically adjusts perturbations based on loss surface geometry. SORA consistently prevents CO, achieves state-of-the-art robustness and clean accuracy, and generalizes across datasets and architectures using a single fixed set of hyperparameters, which is essential for applicability in fast AT. Extensive experiments on diverse datasets and architectures show that SORA matches or surpasses the robustness of prior methods while delivering higher clean accuracy and superior efficiency. Code is available at https://github.com/SecondOrderAT/SORA.

## 1. Introduction

Deep neural networks have achieved remarkable success across a wide range of domains. However, their vulnerability to adversarial examples poses significant challenges to their reliability and security (Szegedy et al., 2014). These adversarial examples are often subtle and imperceptible to the human eye, yet can cause models to make incorrect predictions with high confidence.

Adversarial Training (AT) (Madry et al., 2018) has emerged as one of the most effective defenses against adversarial examples (Athalye et al., 2018). By training models on adversarially perturbed samples, AT improves robustness to attacks during deployment. Despite these benefits, multi-step AT methods such as Projected Gradient Descent (PGD) (Madry et al., 2018) are computationally expensive, as they require several gradient ascent steps to generate adversarial examples at each training iteration. This cost severely limits their scalability to large datasets and deep architectures.

To mitigate this overhead, single-step adversarial training approaches (Wong et al., 2020) have been proposed. While significantly more efficient, these methods introduce new challenges. Notably, Wong et al. (2020) identified a failure mode in single-step AT, commonly observed when using the Fast Gradient Sign Method (FGSM) (Goodfellow et al., 2015), that has been termed *Catastrophic Overfitting* (CO); related limitations of one-step attacks were also discussed in earlier AT work (Madry et al., 2018). In this setting, robust accuracy against multi-step attacks such as PGD collapses to nearly $0\%$, while FGSM accuracy can rise sharply and even exceed the clean accuracy. This suggests that the model overfits to the specific attack used in training and becomes brittle against stronger iterative attacks.

Although many methods have been proposed to address this challenge, we argue that an effective fast AT method should satisfy the following properties:

- **Robustness across datasets and architectures.** It should reliably prevent CO on a diverse range of datasets and models. As shown in Section 6 and Appendix K, our proposed method, Second-Order Adaptive (SORA), succeeds not only on commonly used datasets but also on challenging benchmarks where existing single-step techniques often fail.

- **High clean and robust accuracy with low cost.** A known trade-off exists between clean and robust accuracy (Tsipras et al., 2019; Zhang et al., 2019b). Many prior methods achieve strong robustness by restrict-

---
[*]Equal contribution  [1]Department of Computer Engineering, Sharif University of Technology, Tehran, Iran. Correspondence to: Mohammad Hossein Rohban <rohban@sharif.edu>.

*Proceedings of the $43^{rd}$ International Conference on Machine Learning*, Seoul, South Korea. PMLR 306, 2026. Copyright 2026 by the author(s).

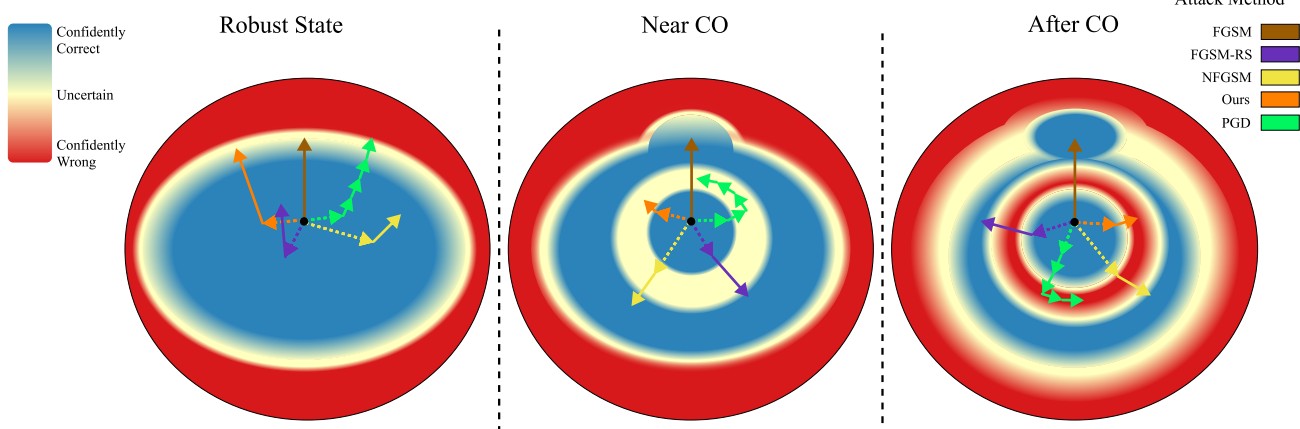

*Figure 1.* Evolution of the loss landscape geometry around a sample during FGSM training. **Left:** Early in training, before CO, the model is robust, with FGSM and PGD accuracies closely matched. The loss surface is approximately semilinear. **Middle:** A few batch updates before CO, the decision boundary begins to wrap around the original and adversarial examples, forming a nonlinear region that is not yet misclassified. **Right:** This nonlinearity escalates, leading to misclassification in that region, which stronger attacks such as PGD exploit. Perturbations from other attacks and our method (SORA) are also visualized.

ing model flexibility (e.g., stronger regularization or reduced capacity), which harms clean accuracy. In contrast, SORA maintains high clean accuracy and state-of-the-art robustness while adding negligible computational and memory overhead by efficiently choosing the attack step-size distribution.

- **Dataset and architecture agnostic hyperparameters.** Several methods require extensive, dataset- and model-specific hyperparameter tuning to avoid CO (Appendix H), which undermines the original goal of Fast Adversarial Training (FAT): robustness with minimal training cost. In our experiments (Section 6 and Appendix K), we fix SORA's hyperparameters across all settings and adopt the single best configuration reported in prior work for a fair comparison, demonstrating strong generalization without costly tuning for each setting.

To the best of our knowledge, no metric has been explicitly designed to predict CO without significant cost. Nevertheless, several existing measures can serve as indirect early indicators of CO onset (Andriushchenko & Flammarion, 2020; Lin et al., 2023; Rocamora et al., 2024). These indicators often incur high computational cost, requiring extra backpropagation or forward passes. To address this gap, we propose PertAlign, a fast, gradient-based metric that is theoretically linked to the curvature of the loss landscape and can predict CO with negligible overhead. It uses gradients from the first and second attack iterations. We further show that PertAlign can anticipate CO earlier than existing indicators.

In addition, we identify a previously overlooked phenomenon, which we term *Epsilon Overfitting* (Figure 1).

We show in Section 3 that variability in the attack step-size plays a crucial role in the emergence of CO. By incorporating this insight, SORA achieves improved robustness across a wider range of settings compared to previous baselines. Our experiments demonstrate that methods incorporating perturbation variability generalize more effectively and consistently avoid CO.

**Our contributions are as follows:**

- We introduce and analyze *Epsilon Overfitting*, demonstrating its importance for robust generalization and CO.

- We propose *PertAlign*, a theoretically grounded and computationally efficient metric for early and reliable CO prediction.

- We present *SORA*, an efficient training paradigm that leverages these insights to adaptively determine attack step-sizes. We show that it matches or surpasses state-of-the-art generalization across datasets and architectures with minimal hyperparameter tuning.

- We identify key criteria for making fast methods deployable in practice and evaluate them on well-known domains, as well as on important but less frequently tested medical domains.

## 2. Background and Related Work

### 2.1. Adversarial Training

Adversarial Training (AT) can be formulated as a min–max optimization problem, where the inner maximization seeks adversarial perturbations that maximize the training loss,

while the outer minimization updates model parameters to minimize the loss on these adversarial examples (Madry et al., 2018). Formally, this can be expressed as:

$$\min_\theta \mathbb{E}_{(x,y)\sim\mathcal{D}} \left[ \max_{\delta\in\Delta} \mathcal{L}\left(f_\theta(x+\delta), y\right) \right], \qquad (1)$$

where $\theta$ denotes the parameters of the model $f$, $(x, y)$ are the training data drawn from the distribution $\mathcal{D}$, $\mathcal{L}(\cdot, \cdot)$ denotes the cross-entropy loss, and $\delta$ is the perturbation confined within a given boundary $\Delta$.

In the standard multi-step AT by (Madry et al., 2018), the loss function used is the cross-entropy loss and $\Delta$ is defined as the $\ell_\infty$-norm ball with radius $\epsilon$, and the inner maximization is solved via the Projected Gradient Descent (PGD) attack. While effective, this iterative approach significantly increases the computational cost for large-scale models and datasets. Zhang et al. (2019b) use a regularization term that minimizes the Kullback–Leibler (KL) divergence between the output distributions of clean and adversarial examples.

Shafahi et al. (2019) use the gradient used for generating adversarial examples for updating the model weights at the same time and therefore eliminate the overhead of the attack.

Following Wong et al. (2020), a family of methods, collectively referred to as Fast Adversarial Training (FAT), emerged with the goal of producing robust models at a fraction of the cost of multi-step AT. However, while these methods improve efficiency, fast single-step AT approaches are prone to a distinctive failure mode known as Catastrophic Overfitting (CO). Understanding and mitigating CO has thus become a critical research direction.

### 2.2. Catastrophic Overfitting

CO is a phenomenon observed during AT, more prevalently during single-step AT, where robustness against multi-step attacks such as PGD suddenly decreases to 0% just within a few epochs, whereas robustness against the FGSM attack rapidly increases. Many studies have focused on explaining CO and developing FAT variants that avoid it. Broadly, prior works can be grouped by their main mitigation strategy:

**Randomization.** Randomization (noise-based) methods inject stochasticity into adversarial example generation to prevent harmful overfitting. Wong et al. (2020) demonstrated that adding random starts to FGSM training can prevent CO. de Jorge Aranda et al. (2022) extended this by increasing the magnitude of random noise and removing the clipping step to boost accuracy.

**Regularization.** Regularization-based methods constrain model behavior to discourage CO. Andriushchenko &

Flammarion (2020) proposed maximizing gradient alignment within the perturbation set. Sriramanan et al. (2020) used Euclidean norm regularization while Sriramanan et al. (2021) used nuclear norm regularization to enforce function smoothing in the vicinity of data samples. Lin et al. (2023) identified *abnormal* adversarial examples as a source of CO and mitigated them via regularization. Similarly, Rocamora et al. (2024) encouraged local linearity in the loss surface.

**Adaptive Step-Sizes.** These methods modify the attack step-size based on training dynamics. Nie et al. (2021) used reinforcement learning to determine step-sizes. Huang et al. (2023) adapted them using gradient norms inspired by Zheng et al. (2020). Zhao et al. (2025) determined the step-size for each sample based on the similarity between the input noise and the gradient direction.

**Other Approaches.** Some works address CO from alternative perspectives. For instance, Golgooni et al. (2023) omitted updates with negligible gradient magnitudes to stabilize training, while Jia et al. (2022; 2024) use prior-guided adversarial initialization to generate stronger adversarial examples by using perturbations from the historical training process. Zhang et al. (2022); Wang et al. (2025) view FAT from the perspective of bi-level optimization.

### 2.3. Second-Order Optimization

Second-order methods have been used to enhance adversarial robustness. Inspired by the relation between curvature and robustness (Fawzi et al., 2016), Moosavi-Dezfooli et al. (2019); Ma et al. (2021); Tsiligkaridis & Roberts (2020) train robust models using curvature regularization. Alternatively Singla & Feizi (2020) use second-order information to improve certifiable robustness and Qian et al. (2022) use second-order optimization to generate stronger adversarial examples to improve AT. However, none of these methods fall within the FAT paradigm.

## 3. Epsilon Overfitting

Although numerous studies have examined CO from various perspectives, a comprehensive understanding of its underlying mechanisms remains elusive. One well-established viewpoint associates CO with changes in the geometry of the model's loss landscape, with several works showing that, at its onset, the loss surface becomes highly distorted (Kim et al., 2021; Kang & Moosavi-Dezfooli, 2021). However, a more precise characterization of this distortion is still lacking. We build on this perspective by analyzing the loss landscape and robustness across different training states, offering a clearer depiction of how the surface evolves throughout CO. Specifically, we investigate whether this distortion follows a consistent, characteristic pattern.

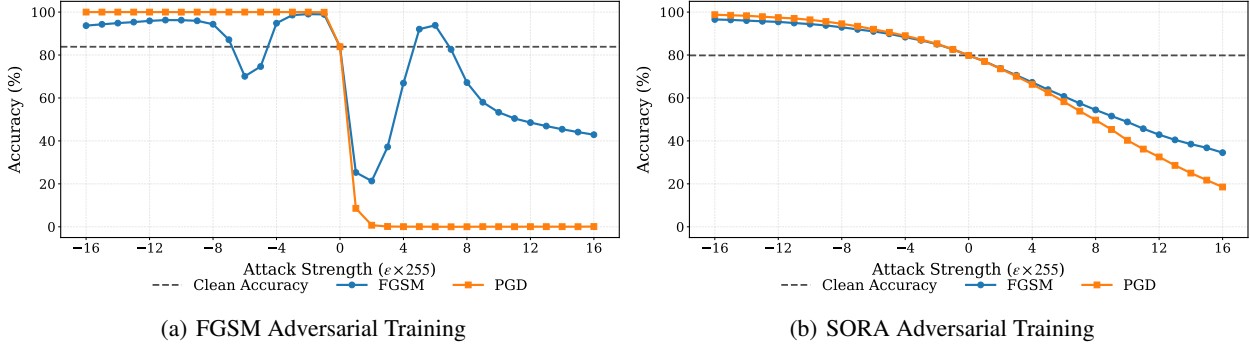

(a) FGSM Adversarial Training    (b) SORA Adversarial Training

*Figure 2.* Comparison of a model exhibiting CO (a) with a robust model (b), evaluated using FGSM and PGD-10.

### 3.1. Evolution of Loss Landscape Geometry

A striking property of CO is that it does not degrade clean accuracy, yet it often causes a substantial increase in FGSM accuracy, sometimes even surpassing clean accuracy on the test set. This counter-intuitive effect prompted us to examine FGSM accuracy across a range of $\epsilon$ values for a model trained with a fixed perturbation budget (e.g., $\epsilon = 8/255$), as shown in Figure 2(a). See Appendix F.5 for more examples.

Two patterns can be observed in this plot. First, there exist certain $\epsilon$ values for which FGSM accuracy drops sharply (e.g., $\epsilon = 2/255$, $\epsilon = -6/255$), while for other values it remains unexpectedly high (e.g., $\epsilon = -2/255$, $\epsilon = 6/255$). Such high variation in FGSM accuracies indicates pronounced nonlinearity in the loss surface, even along the FGSM perturbation direction used during training. Since the model effectively *overfits* to specific $\epsilon$ values, performing well for those perturbation magnitudes but poorly for others, we refer to this as *Epsilon Overfitting* (EO).

Epsilon Overfitting arises when the attack perturbation magnitude is fixed and relatively large, with **limited diversity in magnitude or direction**. Under such constraints, the model can learn to reduce the loss only within the narrow regions reached by the fixed perturbation, rather than across the broader $\ell_\infty$-norm ball. This results in a peculiar form of loss distortion on the onset of CO. See Appendix G for a 3D visualization of the loss surface.

B.S. & Babu (2020); Lin et al. (2023) identified *Abnormal Adversarial Examples* (AAEs), which have lower loss than the original clean samples and become more frequent in the presence of CO. From Figure 1, we see that most AAEs arise in the presence of EO when adversarial examples are generated with the same step-size used during training. In this case, the model enlarges its decision boundary around the adversarial example relative to that around the clean example, producing the observed FGSM accuracy spike, which can even exceed clean accuracy.

### 3.2. Overcoming Epsilon Overfitting

These findings, along with prior work done by Ding et al. (2020); Huang et al. (2023), highlight the importance of adaptive step-size methods. Such methods adjust step-sizes in response to local loss curvature, avoiding interpolation over highly nonlinear regions between the clean and adversarial points and instead targeting the local maxima along the perturbation direction. Randomization-based methods also try to avoid EO as discussed in Appendix C.1.

The FGSM trend in Figure 2(a), where accuracy evolves in a smooth yet non-monotonic manner for small $\epsilon$, indicates that the loss landscape near the clean point is not purely jagged but exhibits structured curvature. This motivates approximating the local neighborhood around $\epsilon = 0$ with a second-order (quadratic) model, thereby enabling curvature-aware step-size selection for loss maximization.

Although second-order methods can explicitly solve the maximization problem via local curvature estimation, their computational cost is prohibitive for large-scale problems due to Hessian computation (Ghojogh et al., 2021). Consequently, most practical approaches infer curvature indirectly to strike a balance between cost and performance. In the following section, we present a technique for approximating loss surface curvature with **nearly zero** computational overhead.

## 4. Perturbation Alignment

One of the key characteristics of CO is that, in most cases, it arises very quickly, often within a single epoch. Once CO occurs, most single-step attack methods cannot recover after the robust accuracy drop, and recovery via multi-step evaluation (e.g., PGD) is computationally expensive. Thus, predicting CO **before** single-step and PGD accuracies diverge can enable training methods to take suitable, low-cost corrective actions. Ideally, we seek an accurate and reliable prognosis tool that can detect when optimization has gone off course, well before the onset of CO.

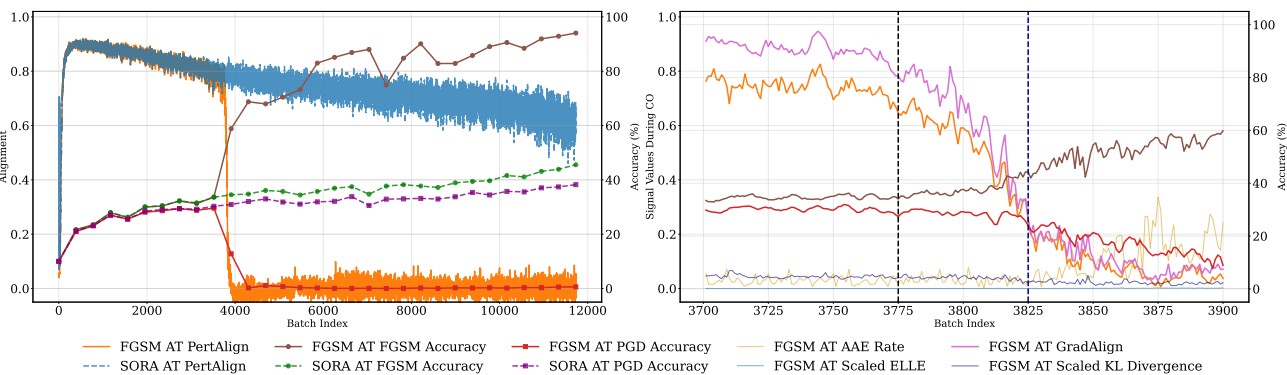

*Figure 3.* **Left:** Tracking PertAlign during FGSM AT and SORA AT for CIFAR-10 dataset, PertAlign collapses on the occurrence of CO. **Right:** At batch 3775 of FGSM AT, PertAlign and GradAlign begin to drop, forecasting CO, while FGSM and PGD accuracies visibly diverge only around batch 3825. Other metrics, AAE Share of the Batch, Scaled KL Divergence from TRADES, and Scaled ELLE nonlinearity measure, react later in response to the model updates. For more details see Appendix J.

We propose a novel metric, **PertAlign**, which can predict CO by measuring the nonlinearity of the loss surface with negligible additional computational cost. PertAlign is defined as the cosine similarity between:

1. The gradient of the loss with respect to the input, computed during adversarial example generation, and

2. The backpropagation gradient, propagated back to the input layer, computed when performing backpropagation on the adversarial example passed through the model.

Formally **PertAlign** is defined as:

$$\cos\left(\nabla_x \mathcal{L}\big(f_\theta(x'), y\big), \nabla_x \mathcal{L}\big(f_\theta(x' + \delta), y\big)\right), \quad (2)$$

where $x' = x + \eta$ and $\eta \sim \mathcal{U}(-\epsilon, +\epsilon)^d$ is the random start, and $\delta = \alpha v$ is a perturbation of scalar step-size $\alpha$ in direction $v$. The perturbation direction $v$ is typically chosen as

$$g = \nabla_x \mathcal{L}\big(f_\theta(x'), y\big) \quad \text{or} \quad p = \text{sign}\left(\nabla_x \mathcal{L}\big(f_\theta(x'), y\big)\right).$$

It can be shown that PertAlign captures the local nonlinearity of the loss surface (see Appendix A).

**Lemma 4.1.** *Let $f_\theta : \mathbb{R}^d \to \mathbb{R}^C$ be a classifier and let $g = \nabla_x \mathcal{L}(f_\theta(x), y)$, $H = \nabla_x^2 \mathcal{L}(f_\theta(x), y)$. Then for $\alpha \ll 1$, PertAlign can be approximated by:*

$$1 - \text{PertAlign} \approx \frac{\alpha^2}{2} \|h_{\perp g}\|^2, \quad h = \frac{Hv}{\|g\|}$$

*where $h_{\perp g}$ denotes component of h, orthogonal to g.*

More specifically, the misalignment measured by PertAlign correlates with the component of $Hv$ (the Hessian $H$

applied to $v$) that is orthogonal to the gradient. This misalignment is minimized when the gradient is aligned with $Hv$, i.e.,

$$v \parallel H^\dagger g, \quad (3)$$

where $H^\dagger$ is the Moore-Penrose pseudo-inverse of $H$. This condition closely resembles Newton's method and implies that the perturbation direction should align with the second-order optimal update step in the loss landscape.

In practice, setting $v = p$ yields consistent and reliable behavior, as shown in Figure 3. When CO occurs, PertAlign drops sharply toward zero, indicating that $p$ is no longer aligned with the optimal second-order direction, signaling a failure of the inner maximization step. Conversely, when the loss surface remains approximately linear, PertAlign remains stable rather than collapsing.

As further discussed in Section 3, since at the onset of CO, **before** PGD accuracy declines, the local loss surface becomes highly nonlinear and unstable (Andriushchenko & Flammarion, 2020; Kim et al., 2021; Rocamora et al., 2024), by PertAlign directly measuring this nonlinearity, it can predict CO earlier than the divergence of single-step and PGD accuracies (Figure 3). Moreover, PertAlign consistently foreshadows CO earlier than other proposed metrics.

Note that PertAlign relies only on two quantities: $\nabla_x \mathcal{L}(f_\theta(x), y)$, computed during all single-step methods, and $\nabla_x \mathcal{L}(f_\theta(x + \delta), y)$, the backpropagation gradient extended by one layer to include the input layer, which is already obtained during the model weight update. As these gradients are already part of standard AT, PertAlign incurs **no additional computational overhead**, making it a fast and reliable metric for CO prediction and **can therefore be applied to any other single-step method**. Shafahi et al. (2019); Zhang et al. (2019a) use a similar trick to speed up AT. The information it provides can also be used to mitigate CO, which our proposed AT method leverages in Section 5.

---

**Algorithm 1** **S**econd-**Or**der **A**daptive Method (SORA)

---

**Inputs:** # epochs $T$, # batches $M$, radius $\epsilon$, step-size $\alpha$, exponential average coefficient $\beta$.

1:   $v \leftarrow 0.99$              {Initialize moving linearity coefficient}
2:   **for** Epoch $t = 1, \ldots, T$ **do**
3:      **for** Batch $i = 1, \ldots, M$ **do**
4:         $\boldsymbol{\eta} \sim \mathcal{U}(-\epsilon, \epsilon)^d$            {Random start}
5:         $\boldsymbol{g} \leftarrow \nabla_{\boldsymbol{x}_i} \mathcal{L}\left(f_{\boldsymbol{\theta}}(\boldsymbol{x}_i + \boldsymbol{\eta}), y_i\right)$
6:         $\boldsymbol{\alpha}_i \sim \mathcal{U}\left(0, \alpha^*\right)^d$         {Element-wise step sampling}
7:         $\boldsymbol{x}'_i \leftarrow \boldsymbol{x}_i + \boldsymbol{\eta} + \boldsymbol{\alpha}_i \odot \mathrm{sign}\left(\boldsymbol{g}\right)$
8:         $\boldsymbol{x}'_i = \Pi_{[0,1]}(\boldsymbol{x}'_i)$        {Project onto the valid pixel range}
9:         $\boldsymbol{g}', \boldsymbol{g}_{\boldsymbol{\theta}} \leftarrow \nabla_{[\boldsymbol{x}'_i, \theta]} \mathcal{L}\left(f_{\boldsymbol{\theta}}(\boldsymbol{x}'_i), y_i\right)$        {Backpropagation}
10:       $\boldsymbol{\theta} \leftarrow \mathrm{optimizer}(\boldsymbol{\theta}, \boldsymbol{g}_{\boldsymbol{\theta}})$      {Standard parameters update, (e.g. SGD)}
11:       Calculate optimal step-size for next batch:

$$\alpha^* \leftarrow \begin{cases} \min\left(\alpha_{\max}, \frac{\alpha_0}{1-v}\right), & v < 1, \\ \alpha_{\max}, & \text{otherwise.} \end{cases}$$

12:       $v \leftarrow (1 - \beta) \cdot v + \beta \cdot \frac{\boldsymbol{p}^T \boldsymbol{g}'}{\|\boldsymbol{g}\|_1}$       {Update moving linearity coefficient}

---

We further examine related approaches, including detailed PertAlign versus GradAlign (Andriushchenko & Flammarion, 2020) comparison in Appendix C.2, and ZeroGrad and MultiGrad (Golgooni et al., 2023) from a second-order prespective in Appendix C.3. Additional properties of PertAlign, alongside an ablation study, are presented in Appendix A.

## 5. Methodology

In Section 3, we demonstrated the importance of using an adaptive step-size in mitigating EO and thus CO. This observation motivates us to explore adaptive step-size strategies more systematically. However, this exploration naturally raises a key question:

*What is the optimal perturbation step-size, given the current state of the model?*

When loss maximization is viewed purely as a first-order optimization problem, our options for determining a suitable step-size are limited. To address this limitation, we instead analyze single-step loss maximization from a second-order optimization perspective.

It can be shown that, by locally approximating the loss landscape with a quadratic function, the optimal step-size for maximizing the loss is given by (see Appendix A for the full derivation):

**Lemma 5.1.** *For a second order loss function, let the perturbation of the input be in the direction of* $\mathrm{sign}(g)$:

$$p = \mathrm{sign}(g) \in \{-1, 0, 1\}^d, \quad v = \alpha p.$$

*Then the optimal step-size that maximizes this loss is*

$$\alpha^* = \begin{cases} \min\left(\alpha_{\max}, \frac{\alpha_0}{1 - \frac{p^T g'}{\|g\|_1}}\right), & \frac{p^T g'}{\|g\|_1} < 1, \\ \alpha_{\max}, & \text{otherwise.} \end{cases}$$

*where* $g' = g + \alpha H p$, $\|\cdot\|_1$ *is the* $\ell_1$ *norm, and* $\alpha_{\max}$ *is the maximum step-size budget, and* $\alpha_0$ *is a fixed numerator.*

Using this formula as a basis for our attack step-size provides a strong initial estimate of an effective perturbation magnitude. To improve stability in cases where the second-order approximation of the loss landscape is inaccurate, and to further mitigate EO, we introduce per-pixel channel diversification: for each channel, we uniformly sample its attack step-size from the range $[0, \alpha^*]$. This stochasticity significantly increases the diversity of perturbations, improving both the robustness and universality of the attack, as demonstrated in Section 6. A summary of our method is provided in Algorithm 1.

Intuitively, SORA adapts to the evolving state of the model by leveraging the backpropagation gradient from the previous batch. By combining this gradient with the general perturbation direction, it estimates the local nonlinearity of the loss surface and adjusts the step-size accordingly.

This dynamic adjustment, illustrated in Figure 4, mitigates overshooting, avoids generating AAEs, and prevents the emergence of highly nonlinear regions in the loss landscape. The theoretical derivation of the optimal step-size along the gradient direction is provided in Appendix A.

To match the time complexity of truly fast methods (Wong et al., 2020), we apply our theoretically optimal step size

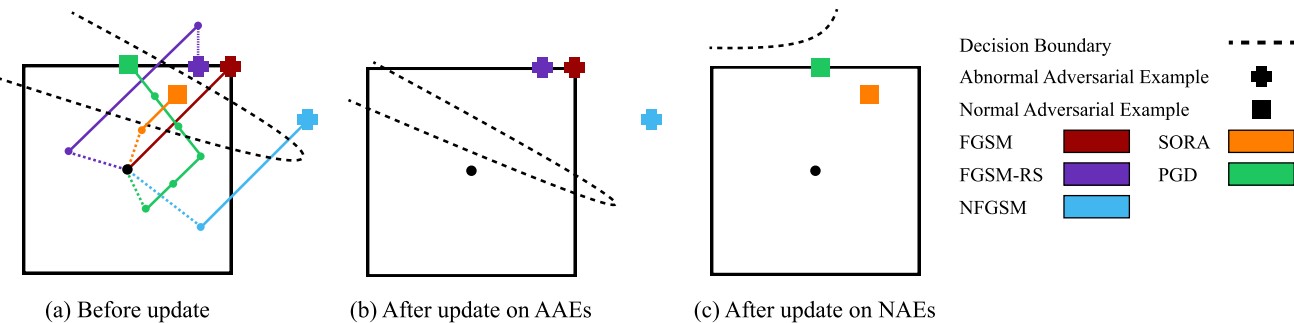

(a) Before update      (b) After update on AAEs      (c) After update on NAEs

*Figure 4.* When the decision boundary becomes distorted, single-step attacks may produce AAEs, whereas multi-step attacks such as PGD can still reliably generate NAEs. By adapting the attack step-size to the local linearity of the loss surface, SORA can also produce NAEs in such scenarios. Training on AAEs tends to exacerbate distortion in the loss surface, while training on NAEs can guide the model toward recovery.

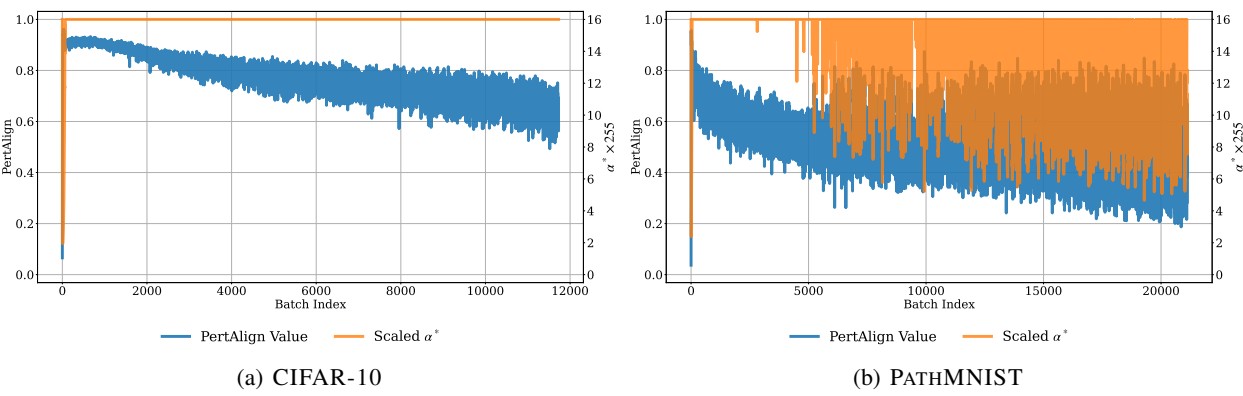

(a) CIFAR-10           (b) PATHMNIST

*Figure 5.* Theoretically derived and self-adaptive $\alpha^*$ during SORA AT with $\epsilon = {}^8/_{255}$ on CIFAR-10 (a) and PATHMNIST (b).

with a *temporal gap*. Similar to the approach of Shafahi et al. (2019), which generates adversarial examples with a temporal gap, we use the optimal step-size with a lag, enabling our method to exploit second-order information essentially for *free*. Since the weights change only slightly after each update, we expect the validity of our theoretical foundation to be preserved. We also apply the optimal step-size to a different batch; such extrapolation across batches is not unprecedented in AT (Shafahi et al., 2019; 2020). We believe our theoretical results remain applicable because all batches are drawn from the same training distribution.

The success of our method supports the observation by Wong et al. (2020) that CO is sudden, drastic, and *universal*, compromising all data points, not isolated outliers. Because the problem is global, we adopt a similarly global corrective mechanism. To handle temporal shifts in weights and batches, we stabilize via exponential moving averages as shown in Algorithm 1.

We also tracked the theoretically optimal step-size $\alpha^*$ (Lemma 5.1) during training on CIFAR-10 and PATHM-NIST. Figure 5(a) shows that, except for a few early batches, SORA selects the maximum possible step-size throughout

training on CIFAR-10, suggesting that the loss landscape is sufficiently linear and does not require step-size reduction to avoid CO. In contrast, on PATHMNIST, SORA consistently readjusts its step-size to achieve robustness (Figure 5(b)). Together with PertAlign's lower average performance on this dataset, this suggests that achieving robustness while avoiding CO is more challenging on PATHMNIST than on CIFAR-10. Thus, failing to incorporate a curvature-aware mechanism in FAT methods may lead to CO on such datasets, as discussed in Section 6.3 and Figure 6. This self-correcting mechanism also enables SORA to perform well with fixed hyperparameters, whereas less flexible methods often require costly setting-specific tuning to avoid CO.

## 6. Experiments

### 6.1. Settings

**Baselines.** We compare our method against a wide range of single-step AT methods, including Free AT (Shafahi et al., 2019), FGSM-RS (Wong et al., 2020), GradAlign (Andriushchenko & Flammarion, 2020), NuAT (Sriramanan et al., 2021), ZeroGrad and MultiGrad (Golgooni et al., 2023), N-FGSM (de Jorge Aranda et al., 2022),

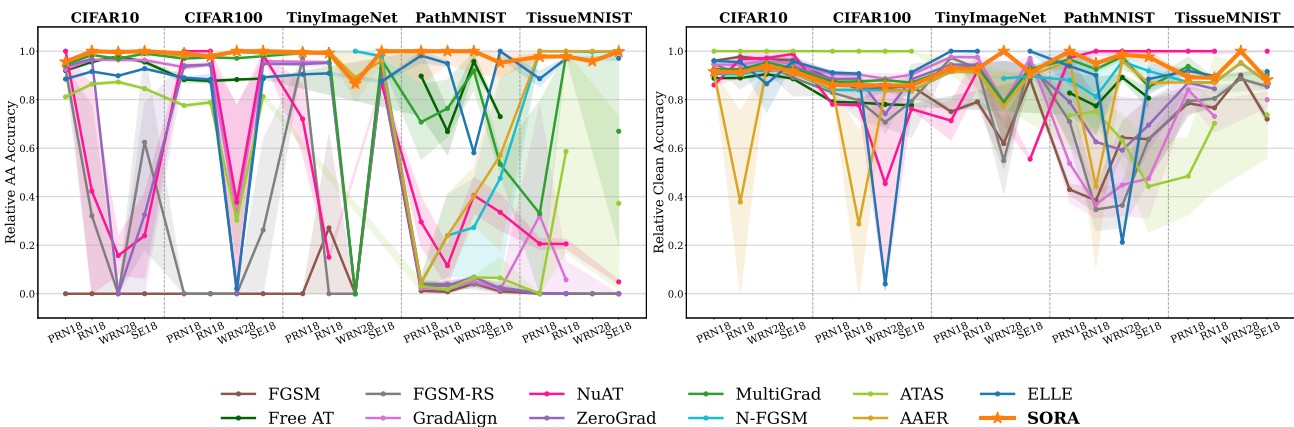

*Figure 6.* Evaluation of different methods across datasets and architectures. **Left:** SORA attains the highest robust accuracy among single-step methods. **Right:** Corresponding clean accuracy.

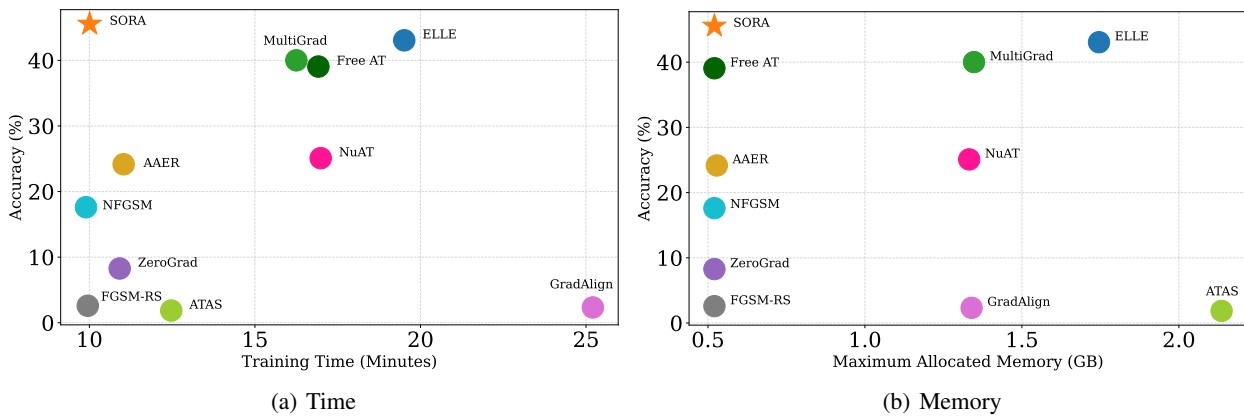

*Figure 7.* Training time vs. memory usage on PATHMNIST with PreActResNet-18, trained for 30 epochs, measured on an NVIDIA GeForce RTX 4090 GPU. The ★ marks SORA. The vertical axis in both figures represents PGD-10 accuracy.

ATAS (Huang et al., 2023), AAER (Lin et al., 2023), and ELLE (Rocamora et al., 2024). We also evaluate multi-step AT methods, including PGD-2 and PGD-10, as well as TRADES (Zhang et al., 2019b), which serve as upper-bound baselines representing idealized performance. We provide more information in Appendix B.

**Datasets and Model Architectures.** The training datasets include CIFAR-10/100 (Krizhevsky, 2009), and TINY-IMAGENET (Le & Yang, 2015), as well as IMAGENET-100 (Russakovsky et al., 2015). To further evaluate the generalization capability of single-step AT methods, we additionally assess our method and the baselines on PATHM-NIST and TISSUEMNIST from the MEDMNIST collection (Yang et al., 2023). More information about these datasets is available in Appendix D. We evaluate on multiple architectures, including ResNet (He et al., 2016a), PreActResNet (He et al., 2016b), and WideResNet (Zagoruyko & Komodakis, 2017) from the ResNet family, as well as SENet (Hu et al., 2018) and ViT (Dosovitskiy et al., 2021).

These architectures are discussed in Appendix E.

**Hyperparameters.** Unless otherwise stated, we set the perturbation budget to $\epsilon = \,^8/_{255}$ and follow the training setup of Wong et al. (2020), using the SGD optimizer with momentum $0.9$ and weight decay $5 \times 10^{-4}$. For our method we set $\alpha_0 = 0.02$, $\beta = 0.05$, and $\alpha_{\max} = 2\epsilon$. Exact hyperparameter values and training details are included in Appendices B, D, and E.

### 6.2. Results

To evaluate the generalizability of single-step AT methods without model- or dataset-specific hyperparameter tuning, we compute each method's AutoAttack (AA) (Croce & Hein, 2020) and clean accuracy relative to the best-performing method in each setting (Figure 6). A relative accuracy closer to 1 indicates performance closer to the state-of-the-art (SOTA) in that setting. For comprehensive results, see Appendix K where we also reported AA for all baselines.

*Table 1.* Training results for PreActResNet-18 on PATHMNIST and IMAGENET-100 for $\epsilon = 8/255$. The performance gap of each method with respect to SORA is also shown in the parentheses.

| Method | PATHMNIST | | IMAGENET-100 | |
|---|---|---|---|---|
| | Clean | AutoAttack | Clean | AutoAttack |
| FGSM (Goodfellow et al., 2015) | 46.06 (-39.87%) | 0.42 (-32.49%) | 15.94 (-41.32%) | 0.00 (-18.56%) |
| N-FGSM (de Jorge Aranda et al., 2022) | 74.89 (-11.04%) | 1.65 (-31.26%) | 49.40 (-7.86%) | 15.52 (-3.04%) |
| AAER (Lin et al., 2023) | 83.54 (-2.39%) | 2.48 (-30.43%) | 48.26 (-9.00%) | 17.18 (-1.38%) |
| **SORA** (Ours) | **85.93%** | **32.91%** | **57.26%** | **18.56%** |

SORA is the only fast method that consistently matches or surpasses the SOTA across all settings, while maintaining higher clean accuracy than competing methods with comparable robust accuracy, and minimal computational overhead (Figure 7). This demonstrates a superior trade-off between robustness, clean accuracy and efficiency.

SORA's effectiveness becomes even more pronounced on more challenging datasets with finer image textures and higher resolution, where it consistently outperforms strong baselines such as N-FGSM and AAER in terms of both clean and robust accuracy, as shown in Table 1.

### 6.3. Discussion on Universality

Figure 6 reveals a pronounced performance gap between prior methods on both standard datasets (CIFAR-10/100, TINYIMAGENET) and more challenging datasets (PATHMNIST, TISSUEMNIST). This highlights a **lack of generalization** in previous approaches (see Appendix H).

Robustness on PATHMNIST and TISSUEMNIST is particularly challenging due to finer image textures, shorter inter-class distances compared to datasets such as CIFAR-10 (Figure 9), and class imbalance (Figure 8). Many methods that avoid CO and perform well on standard benchmarks suffer noticeable drops in robust accuracy on these harder datasets. Furthermore, approaches that do maintain robustness, such as ELLE and MultiGrad, incur substantially higher computational or memory costs (Figure 7) or sacrifice clean accuracy such as N-FGSM (Table 1).

Even when we conduct hyperparameter search for these baselines, we are unable to find any configuration which performs well on PATHMNIST or even when such a configuration exists, it results in a significant performance drop on other datasets such as CIFAR-10, indicating the lack of a universal set of hyperparameters necessary for a FAT method. For more details see Appendices H and M.3.

### 6.4. Contribution of SORA's Components

We perform an ablation study to evaluate the contribution of each component of SORA to its overall performance.

Experiments are conducted using PreActResNet-18 on the PATHMNIST dataset, which is more susceptible to CO than CIFAR-10/100, and thus better exposes the effect of each component. As shown in Table 2, increasing the variability of the step-size magnitude significantly improves robust accuracy; most notably, removing the optimal step-size selection results in the lowest PGD-10 accuracy.

*Table 2.* Ablation study on the components of SORA.

| Configuration | Clean | FGSM | PGD-10 |
|---|---|---|---|
| **SORA (baseline)** | 84.69 | 57.51 | 45.56 |
| – Without Random Sampling | 85.67 | 58.89 | 45.35 |
| – Clamping Step-Size | 86.82 | 52.02 | 34.08 |
| – Without Optimal Step-Size | 89.93 | 28.30 | 17.23 |

These results highlight the importance of both attack strength and directional variability, as discussed in Section 3, and demonstrate the necessity of adaptive step-size selection for challenging datasets.

## 7. Conclusion

We analyzed loss surface distortion in CO, revealing the critical role of the attack perturbation distribution in Epsilon Overfitting (EO) and its close connection to CO. We further introduced PertAlign, a novel, cost-free metric for reliably predicting CO based on loss surface curvature. Building on insights from EO and PertAlign, we developed SORA, an adaptive step-size method that dynamically adjusts its perturbation distribution according to local linearity.

Extensive experiments across 15 baselines, 6 datasets, and 4 architectures demonstrate that SORA consistently prevents CO, achieves SOTA performance, and generalizes robustly across datasets and architectures, settings in which many existing methods fail. We believe that PertAlign, together with the criteria highlighted in this work, provides a principled foundation for developing improved fast adversarial training methods in future research, enabling better generalization and more reliable evaluation through broader and more diverse experimental validation.

## Acknowledgments

The authors thank the reviewers, program chairs, and area chairs of ICML 2026 for their constructive feedback and efforts.

We also thank Zeinab Golgooni for helpful discussions and comments.

We are especially grateful to Sadra Teymourian for his valuable support, for providing GPU resources that enabled several of our experiments, and for his essential assistance with the submission process.

## Impact Statement

This paper presents work whose goal is to advance the field of trustworthy machine learning by solving a key practicality problem in adversarial robustness. The high computational cost of robust training is a major barrier to its widespread adoption, leaving many real-world systems vulnerable. We show that many existing *fast* methods hide large tuning costs, which are frequently a necessary response to instabilities like **Catastrophic Overfitting**. To address this issue, we propose **SORA**, a method that offers true training efficiency and stability. By significantly lowering the barrier to obtaining robust models, this work could help secure vital ML applications. We consciously acknowledge that efficient training still consumes energy, and that improved robustness, while critical for security, is one component of a broader effort needed to build safe and responsible AI.

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

## Appendix Guide

In Appendix A, we present the **theoretical foundations** of our work.

Details about the baselines, datasets, and architectures (including their selection criteria) are provided in Appendices B, D, and E respectively.

Appendix C discusses related work in greater depth and **expands the theoretical justifications for prior approaches**. In Appendix C.1 we examine the success of randomization-based approaches through the lens of our new theoretical and empirical insights. In Appendix C.2 we explain the key differences between GradAlign and our proposed PertAlign metric. In Appendix C.3 we provide a **new perspective** on ZeroGrad and MultiGrad, and elaborate how they manage to mitigate CO. Appendix C.4 examines the relation between SORA and multi-step adversarial training in a simplified setting and in Appendix C.5 we discuss FGSM-CKPT as another adaptive step-size method and compare it to SORA.

Our ablation studies are presented in Appendix F.

We have visualized the loss surface surrounding origin for different datasets and training methods in Appendix G to demonstrate the structure of the loss distortion on the onset of CO.

In Appendix H we perform a grid search on hyperparameters for some competitive baselines to illustrate how some of these methods either **cannot yield competitive results** on some datasets, or **require extensive tuning to be competitive**, or both.

In Appendix I we provide an $\ell_2$-norm variant of our SORA method, building on our theoretical work.

In Appendix J we offer further insights into how and why PertAlign is effective, and explain why **PertAlign reliably detects Catastrophic Overfitting**.

Comprehensive results are available in Appendix K. For brevity, a summary is also provided in Appendix K.1. Additional details about the computational complexity of our method and other baselines is also available in Appendix K.2. Our results on the ViT architecture are available in Appendix L.

We outline our **Research Statement** in Appendix M. The limitations of our work are addressed in Appendix M.1. Guidelines for reproducing our main results, figures, and ablations are in Appendix M.2. Our approach to **tuning SORA and baselines while ensuring generalizability** is described in Appendix M.3. Finally, our use of LLMs and the scope of their involvement are documented in Appendix M.4.

## Contents

# A. Theoretical Insights on PertAlign and Optimal Step-Size

## A.1. Overview

In this appendix we begin with the general relation in Lemma 4.1, showing how misalignment scales quadratically with step-size and depends on the distance between the Hessian-transformed step direction and its projection onto the gradient. Remark A.2 notes that this result is robust to small random noise, while Corollary A.3 specializes it to the gradient direction and Remark A.4 provides an eigenvector-based interpretation. Lemma A.5 then establishes an upper bound on possible misalignment. Proposition A.6 and Proposition A.7 use these findings to derive optimal step-size formulas for gradient and sign-gradient directions, respectively. Finally, Lemma A.8 introduces an alternative PertAlign variant.

## A.2. Notation

We summarize the notations used throughout this paper:

- $(x, y) \in \mathbb{R}^d \times \{0, 1\}^{\mathcal{C}}$: Input vector and one-hot label, $\mathcal{C}$ classes.

- $f_\theta(\cdot) : \mathbb{R}^d \to \mathbb{R}^{\mathcal{C}}$: Output logits of a neural network classifier parameterized by $\theta$.

- $\mathcal{L}(\cdot, \cdot)$: Cross-entropy loss.

- $D_{\mathrm{KL}}(\cdot \| \cdot)$: Kullback–Leibler (KL) divergence.

- $\|\cdot\|$: Euclidean norm $\ell_2$ unless otherwise stated.

- $\cos(a, b)$: Cosine similarity between vectors $a$ and $b$ defined as $\frac{a^T b}{\|a\| \cdot \|b\|}$.

- $g = \nabla_x \mathcal{L}(f_\theta(x), y)$: Gradient of the loss w.r.t. input $x$.

- $p = \mathrm{sign}\left(\nabla_x \mathcal{L}(f_\theta(x), y)\right)$: Sign of the gradient of the loss w.r.t. input $x$.

- $I$: Identity matrix.

- $H = \nabla_x^2 \mathcal{L}(f_\theta(x), y)$: Hessian matrix.

- $H^\dagger$: Moore-Penrose pseudo-inverse of matrix $H$.

- $\mathcal{U}(a, b)$: A uniform distribution on $[a, b]$.

- $a \odot b$: The Hadamard product between vectors $a$ and $b$.

- $\eta \sim \mathcal{U}(-k\epsilon, +k\epsilon)$: Denotes random noise.

- $\delta$: Input perturbation.

- $\Pi(\cdot)$: Projection operator.

- $\mathcal{O}(\cdot)$: Asymptotic upper bound.

## A.3. PertAlign–Hessian Correlation

**Lemma A.1** (PertAlign–Hessian relationship for arbitrary step direction $v$). *Let $f_\theta : \mathbb{R}^d \to \mathbb{R}^{\mathcal{C}}$ be a classifier and let $g = \nabla_x \mathcal{L}(f_\theta(x), y)$, $H = \nabla_x^2 \mathcal{L}(f_\theta(x), y)$. Then for $\alpha \ll 1$, PertAlign can be approximated by:*

$$1 - \mathrm{PertAlign} \approx \frac{\alpha^2}{2} \|h_{\perp g}\|^2, \quad h = \frac{Hv}{\|g\|}$$

*where $h_{\perp g}$ denotes component of $h$, orthogonal to $g$.*

*Proof.* **Note that in practice, in cases where $v$ is computed from $x$ (e.g. $v = g$ or $v = p$), we detach $v$ from the PyTorch computation graph after calculating its value. As a result, its functionality as a function of $x$ is removed, and in the Taylor expansion shown below we can assume $v$ is fixed with respect to $x$.**

For the sake of brevity we let $L(x) := \mathcal{L}(f_\theta(x), y)$. Using a second-order Taylor expansion,

$$\nabla_x L(x + \delta) = \nabla_x L(x) + H\delta + \mathcal{O}(\|\delta\|^2) = g + \alpha H v + R, \quad R = \mathcal{O}\left(\alpha^2\right).$$

Let $g' := \nabla_x L(x + \delta)$, then:

$$g^T g' = \|g\|^2 + \alpha g^T H v + g^T R, \tag{4}$$

$$\|g'\|^2 = \|g\|^2 + 2\alpha g^T H v + \alpha^2 \|Hv\|^2 + 2g^T R + \mathcal{O}\left(\alpha^3\right). \tag{5}$$

Using the cosine formula from Equation 2:

$$\text{PertAlign} = \frac{g^T g'}{\|g\| \cdot \|g'\|}.$$

Substituting Equations 4 and 5 and defining

$$k := \frac{g^T H v}{\|g\|^2}, \quad \gamma := \frac{g^T R}{\|g\|^2},$$

we obtain

$$\text{PertAlign} = \frac{1 + \alpha k + \gamma}{\sqrt{1 + 2\alpha k + 2\gamma + \alpha^2 \frac{\|Hv\|^2}{\|g\|^2} + \mathcal{O}(\alpha^3)}}.$$

Applying the expansion $\sqrt{1 + \epsilon} \approx 1 + \frac{\epsilon}{2} - \frac{\epsilon^2}{8}$ and setting $\epsilon = 2\alpha k + 2\gamma + \alpha^2 \frac{\|Hv\|^2}{\|g\|^2} + \mathcal{O}(\alpha^3)$, we get

$$\text{PertAlign} = \frac{1 + \alpha k + \gamma}{1 + \alpha k + \gamma + \alpha^2 \frac{\|Hv\|^2}{2\|g\|^2} - \frac{\alpha^2 k^2}{2} + \mathcal{O}(\alpha^3)}$$

Let

$$\epsilon_1 = \alpha k + \gamma, \qquad \epsilon_2 = \alpha k + \gamma + \alpha^2 \frac{\|Hv\|^2}{2\|g\|^2} - \frac{\alpha^2 k^2}{2} + \mathcal{O}(\alpha^3)$$

Then,

$$\frac{1 + \epsilon_1}{1 + \epsilon_2} = (1 + \epsilon_1)\left(1 - \epsilon_2 + \epsilon_2^2 + \mathcal{O}(\epsilon_2^3)\right)$$

$$= 1 + \epsilon_1 - \epsilon_2 - \epsilon_1 \epsilon_2 + \epsilon_2^2 + \mathcal{O}(\alpha^3)$$

$$= 1 + \alpha k + \gamma - \alpha k - \gamma - \alpha^2 \frac{\|Hv\|^2}{2\|g\|^2} + \frac{\alpha^2 k^2}{2} - \alpha^2 k^2 + \alpha^2 k^2 + \mathcal{O}(\alpha^3)$$

$$= 1 - \alpha^2 \frac{\|Hv\|^2}{2\|g\|^2} + \frac{\alpha^2 k^2}{2} + \mathcal{O}(\alpha^3)$$

Thus,

$$\text{PertAlign} = 1 - \frac{\alpha^2}{2}\left(\frac{\|Hv\|^2}{\|g\|^2} - k^2\right) + \mathcal{O}(\alpha^3) \tag{6}$$

We can rewrite Equation 6 in another form:

$$\frac{\|Hv\|^2}{\|g\|^2} - k^2 = \frac{(Hv)^T(Hv) - (kg)^T(kg)}{\|g\|^2}$$

$$= \frac{(Hv)^T(Hv) - 2k(kg^T g) + (kg)^T(kg)}{\|g\|^2}$$

$$= \frac{(Hv)^T(Hv) - 2kg^T H^T v + (kg)^T(kg)}{\|g\|^2}$$

In the last equality, we used the symmetry of the Hessian matrix.

$$\frac{(Hv)^T(Hv) - 2(Hv)^T(kg) + (kg)^T(kg)}{\|g\|^2} = \frac{(Hv - kg)^T(Hv - kg)}{\|g\|^2} = \frac{\|Hv - kg\|^2}{\|g\|^2}$$

Which leads us to

$$\text{PertAlign} = 1 - \frac{\alpha^2}{2}\left\|\frac{Hv}{\|g\|} - \frac{kg}{\|g\|}\right\|^2 + \mathcal{O}(\alpha^3)$$

$$= 1 - \frac{\alpha^2}{2}\left\|\frac{Hv}{\|g\|} - \frac{g^T Hvg}{\|g\|^3}\right\|^2 + \mathcal{O}(\alpha^3)$$

Defining $h = \frac{Hv}{\|g\|}$ we can simplify

$$\text{PertAlign} = 1 - \frac{\alpha^2}{2}\left\|h - \frac{g^T hg}{\|g\|^2}\right\|^2$$

$$= 1 - \frac{\alpha^2}{2}\left\|h - (h^T\hat{g})\hat{g}\right\|^2$$

where $\hat{g} = \frac{g}{\|g\|}$ is the unit gradient direction. Further, we can write

$$1 - \text{PertAlign} \approx \frac{\alpha^2}{2}\|h_{\perp g}\|^2$$

$\square$

*Remark* A.2. The conclusion of Lemma 4.1 also holds when the input is augmented with small random noise, i.e., $x' = x + \eta$, since such perturbations do not alter the logic of the proof.

**Corollary A.3** (Special case $v = g$). *If $v$ is aligned with the gradient, Lemma 4.1 reduces to*

$$\text{PertAlign} = 1 - \frac{\alpha^2}{2}\|H\hat{g} - k\hat{g}\|^2 + \mathcal{O}(\alpha^3), \tag{7}$$

*where $\hat{g} = g/\|g\|$ and $k = \frac{g^T Hg}{\|g\|^2}$.*

*Remark* A.4. In this case, if the loss surface is locally linear in the gradient direction, Equation 7 can be rewritten as

$$1 - \text{PertAlign} \approx \frac{\alpha^2}{2}\|(H - kI)\hat{g}\|^2.$$

Thus the gradient satisfies $Hg \approx kg$, i.e., $\hat{g}$ is approximately an eigenvector of $H$, and PertAlign $\approx 1$. If $\hat{g}$ is not aligned with any eigenvector, then $H\hat{g}$ points in a different direction from $\hat{g}$, implying the gradient direction changes after a small step.

**Lemma A.5** (Upper bound on misalignment). *For $v = g$,*

$$1 - \text{PertAlign} \leq \frac{\alpha^2}{2}\left(\|H\|^2 - k^2\right) \leq \frac{\alpha^2}{2}\|H\|^2.$$

*Proof.* This follows directly from Equation 6, using

$$\max_g \frac{\|Hg\|^2}{\|g\|^2} = \|H\|^2,$$

and the fact that $k^2 \geq 0$. $\square$

## A.4. Step-Size Selection from Alignment Approximation

**Proposition A.6** (Optimal $\alpha$ for gradient ascent $v = \alpha g$). *Approximating the loss landscape locally as quadratic,*

$$\alpha^* = \begin{cases} \min\left(\alpha_{\max}, \dfrac{\alpha_0}{1 - \frac{\|g'\|}{\|g\|} \cdot \text{PertAlign}}\right), & \dfrac{\|g'\|}{\|g\|} \cdot \text{PertAlign} < 1, \\ \alpha_{\max}, & \text{otherwise.} \end{cases}$$

*where $g' = g + \alpha H g$ is the perturbed gradient, $\alpha_{\max}$ is the maximum step-size budget, and $\alpha_0$ is a fixed numerator.*

*Proof.* By approximating the loss landscape as a quadratic function, we can use alignment to find better candidates for step-size ($\alpha$):

$$L(x + \delta) = L(x) + g^T \delta + \frac{1}{2}\delta^T H \delta$$

Using $\delta = \alpha g$, substitute in the Equation above:

$$L(x + \alpha g) = L(x) + \alpha\|g\|^2 + \frac{\alpha^2}{2}g^T H g$$

We can find the optimal $\alpha$ as

$$\frac{\partial L(x + \alpha g)}{\partial \alpha} = 0 = \|g\|^2 + \alpha g^T H g \implies \alpha^* = -\frac{\|g\|^2}{g^T H g}$$

Using Taylor's approximation of the gradient:

$$\nabla_x L(x + \alpha g) = \nabla_x L(x) + \nabla_x^2 L(x)\delta = g' = g + \alpha H g$$

$$g^T g' = \|g\|^2 + \alpha g^T H g \implies g^T H g = \frac{g^T g' - \|g\|^2}{\alpha}$$

Substituting this into the step-size equation:

$$\alpha^* = -\frac{\alpha\|g\|^2}{g^T g' - \|g\|^2} = \frac{\alpha}{1 - \frac{g^T g'}{\|g\|^2}}$$

Note that

$$\frac{g^T g'}{\|g\|^2} = \frac{\|g'\|}{\|g'\|}\frac{g^T g'}{\|g\|\|g\|} = \frac{\|g'\|}{\|g\|} \cdot \text{PertAlign}$$

Which gives us,

$$\alpha^* = \frac{\alpha}{1 - \frac{\|g'\|}{\|g\|} \cdot \text{PertAlign}}.$$

Note that, since our objective is to *maximize* the loss, the optimal step-size must be strictly positive (i.e., $\alpha^* > 0$). If the $\alpha^*$ obtained from $\frac{dL}{d\alpha} = 0$ is negative, this implies $g^T H g > 0$, corresponding to an *upward curvature* in the perturbation direction. In such cases, the computed $\alpha^*$ would lead toward a local minimum of the loss rather than a maximum. Therefore, to ensure loss maximization, we set the step-size to the largest permissible value, namely, the maximum perturbation budget.

Because the quadratic approximation underlying this derivation becomes unreliable for large step-sizes, we explicitly cap $\alpha^*$ by the perturbation budget. Moreover, to further stabilize the approximation in practice, we fix the numerator of the step-size formula to a constant value $\alpha_0$. This choice does not affect the validity of the proof as long as $\alpha_0$ remains sufficiently small.

Putting these considerations together, the final practical form of $\alpha^*$ is:

$$\alpha^* = \begin{cases} \min\left(\alpha_{\max}, \dfrac{\alpha_0}{1 - \dfrac{\|g'\|}{\|g\|} \cdot \text{PertAlign}}\right), & \dfrac{\|g'\|}{\|g\|} \cdot \text{PertAlign} < 1, \\ \alpha_{\max}, & \text{otherwise,} \end{cases}$$

where $\alpha_{\max}$ is the perturbation budget, $\alpha_0$ is the fixed numerator, $g'$ is the backpropagation gradient on the adversarial example, and PertAlign denotes the alignment metric defined in Equation 2.

$\square$

To demonstrate the effectiveness of SORA in $\ell_2$ norm, we have provided some empirical results in Appendix I.

**Proposition A.7** (Optimal $\alpha$ for sign-gradient $v = \alpha p$). *For perturbation $v = \alpha p = \alpha \operatorname{sign}(g)$,*

$$\alpha^* = \begin{cases} \min\left(\alpha_{\max}, \frac{\alpha_0}{1 - \frac{p^T g'}{\|g\|_1}}\right), & \frac{p^T g'}{\|g\|_1} < 1, \\ \alpha_{\max}, & otherwise. \end{cases}$$

*where $g' = g + \alpha H p$, $\|\cdot\|_1$ is the $\ell_1$ norm, $\alpha_{\max}$ is the maximum step-size budget, and $\alpha_0$ is a fixed numerator.*

*Proof.* The loss after perturbation becomes:

$$L(x + \alpha p) = L(x) + \alpha g^T p + \frac{\alpha^2}{2} p^T H p.$$

Solving $\frac{dL}{d\alpha} = 0$ yields:

$$\alpha^* = -\frac{g^T p}{p^T H p}.$$

Using $g' = g + \alpha H p$, we have:

$$p^T g' = p^T g + \alpha p^T H p \quad \Longrightarrow \quad p^T H p = \frac{p^T g' - p^T g}{\alpha}.$$

Substituting yields:

$$\alpha^* = \frac{\alpha g^T p}{p^T g - p^T g'}.$$

Since $p^T g = \|g\|_1$:

$$\alpha^* = \frac{\alpha}{1 - \frac{p^T g'}{\|g\|_1}}.$$

Since our objective is to *maximize* the loss, the optimal step-size must satisfy $\alpha^* > 0$. If the $\alpha^*$ obtained from the stationary condition $\frac{dL}{d\alpha} = 0$ is negative, this implies $p^T H p > 0$, corresponding to an *upward curvature* along the perturbation direction. In such cases, the computed $\alpha^*$ would lead toward a local minimum of the loss rather than its maximum. To avoid this, we set the step-size to the largest permissible value, namely the perturbation budget.

Because the quadratic approximation underlying this derivation becomes unreliable for large step-sizes, we explicitly cap $\alpha^*$ by the perturbation budget. Moreover, to further stabilize the approximation in practice, we fix the numerator of the step-size formula to a constant value $\alpha_0$. This choice does not affect the validity of the proof as long as $\alpha_0$ remains sufficiently small.

Putting these considerations together, the final practical form of $\alpha^*$ is:

$$\alpha^* = \begin{cases} \min\left(\alpha_{\max}, \frac{\alpha_0}{1 - \frac{p^T g'}{\|g\|_1}}\right), & \frac{p^T g'}{\|g\|_1} < 1, \\ \alpha_{\max}, & otherwise, \end{cases}$$

where $\alpha_{\max}$ denotes the perturbation budget, $\alpha_0$ is the fixed numerator, $g'$ is the backpropagation gradient computed on the adversarial example, and $p$ is the perturbation direction.

$\square$

## A.5. Alternative PertAlign Variant (Ablation)

**Lemma A.8** (First-order Hessian correlation for sign alignment). *Let $p = \text{sign}(g)$. Define*

$$\text{AltPertAlign} = \beta \cos(p, g'), \quad g' = g + \alpha Hp.$$

*For $\alpha \ll 1$,*

$$1 - \text{AltPertAlign} \approx \alpha \left( \frac{p^T Hp}{\|g\|_1} - \frac{g^T Hp}{\|g\|^2} \right),$$

*with $\beta = \frac{\sqrt{d}\|g\|}{\|g\|_1}$.*

*Proof.* Let $L(x) := \mathcal{L}(f_\theta(x), y)$. Using a second-order Taylor expansion,

$$\nabla_x L(x + \delta) = \nabla_x L(x) + H\delta + \mathcal{O}(\|\delta\|^2) = g + \alpha Hp + R, \quad R = \mathcal{O}(\alpha^2).$$

Let $g' := \nabla_x L(x + \delta)$. Then,

$$p^T g' = p^T g + \alpha p^T Hp + p^T R, \tag{8}$$

$$\|g'\|^2 = \|g\|_2^2 + 2\alpha g^T Hp + \alpha^2 \|Hp\|^2 + 2g^T R + \mathcal{O}(\alpha^3). \tag{9}$$

Using Equations 8 and 9,

$$\text{AltPertAlign} = \beta \frac{p^T g + \alpha p^T Hp + p^T R}{\|p\| \sqrt{\|g\|^2 + 2\alpha g^T Hp + \alpha^2 \|Hp\|^2 + 2g^T R + \mathcal{O}(\alpha^3)}}$$

$$= \beta \frac{\|g\|_1}{\|p\| \|g\|} \frac{1 + \alpha \frac{p^T Hp}{\|g\|_1} + \frac{p^T R}{\|g\|_1}}{\sqrt{1 + 2\alpha \frac{g^T Hp}{\|g\|^2} + \alpha^2 \frac{\|Hp\|^2}{\|g\|^2} + 2 \frac{g^T R}{\|g\|^2} + \mathcal{O}(\alpha^3)}}$$

Using the expansion $\frac{1}{\sqrt{1+\epsilon}} = 1 - \frac{\epsilon}{2} + \mathcal{O}(\epsilon^2)$ and setting $\epsilon = 2\alpha \frac{g^T Hp}{\|g\|^2} + \alpha^2 \frac{\|Hp\|^2}{\|g\|^2} + 2\frac{g^T R}{\|g\|^2} + \mathcal{O}(\alpha^3)$,

$$\text{AltPertAlign} \approx \beta \frac{\|g\|_1}{\|p\| \|g\|} \left( 1 + \alpha \frac{p^T Hp}{\|g\|_1} + \frac{p^T R}{\|g\|_1} \right) \left( 1 - \alpha \frac{g^T Hp}{\|g\|^2} - \frac{\alpha^2}{2} \frac{\|Hp\|^2}{\|g\|^2} - \frac{g^T R}{\|g\|^2} \right)$$

Neglecting second-order terms,

$$\text{AltPertAlign} \approx \beta \frac{\|g\|_1}{\|p\| \|g\|} \left( 1 + \alpha \left( \frac{p^T Hp}{\|g\|_1} - \frac{g^T Hp}{\|g\|^2} \right) \right)$$

Choosing $\beta = \frac{\|p\|\|g\|}{\|g\|_1}$, we obtain

$$1 - \text{AltPertAlign} \approx \alpha \left( \frac{g^T Hp}{\|g\|^2} - \frac{p^T Hp}{\|g\|_1} \right)$$

Further, since $p \in \{-1, 0, 1\}^d$,

$$\beta = \frac{\sqrt{d}\|g\|_2}{\|g\|_1} \tag{10}$$

$\square$

# B. Baselines

We provide details about the hyperparameters of the baseline models here.

**Baseline Selection Criteria**

Excluding SORA, we included 11 single-step methods, 3 multi-step methods, and a benign training baseline, all published in top-tier venues. Our goal was to include both competitive recent methods such as N-FGSM (NeurIPS 2022), AAER (NeurIPS 2024), and ELLE (ICLR 2024), as well as established methods such as Free AT (NeurIPS 2019), FGSM-RS (ICLR 2020), GradAlign (NeurIPS 2020), and ZeroGrad / MultiGrad (Intelligent Systems with Applications 2023) for single-step training, and TRADES (ICML 2019) for multi-step training, in addition to the standard FGSM and PGD methods.

We also included ATAS (IEEE TIP 2022), which is another adaptive step-size method but based on a different mechanism. Given the large number of methods proposed in this area and our goal of evaluating across many dataset–architecture combinations, to be as diverse and extensive as possible.

We included ATAS instead of ATTA (Zheng et al., 2020) since ATAS is a more advanced and competitive extension of ATTA. In addition, the ATAS paper analyzes the limitations of ATTA in detail.

## B.1. FGSM

The Fast Gradient Sign Method (FGSM) introduced by Goodfellow et al. (2015) can be used directly in Equation 1. FGSM consists of a single-step update using the sign of the gradient:

$$x_{\text{adv}} = x + \epsilon \cdot \text{sign}\left(\nabla_x \mathcal{L}(f_\theta(x), y)\right)$$

To ensure that the image pixels reside within a legitimate range, an additional clipping step is added to discard values less than 0 or greater than 1 (or 255 if the images are not normalized). Madry et al. (2018) and Wong et al. (2020) have observed that FGSM adversarial training suffers from Catastrophic Overfitting. We include FGSM adversarial training despite this limitation as baseline for other models ability to overcome CO.

## B.2. Free AT

Shafahi et al. (2019) introduce Free Adversarial Training. Their main idea is to use the same gradients used for generating adversarial examples to update the parameters of the neural network as well. This way, generating attacks don't really incur much additional cost since they will be incorporated in the main training process. For this trick to work, instead of seeing each batch only once during each epoch, they use multiple minibatch replays. Each minibatch replay corresponds to a single iteration of the attack. Additionally, due to the existence of Universal Adversarial Perturbations (Moosavi-Dezfooli et al., 2017), Shafahi et al. (2019) also use the perturbations from one batch as the initialization for the next batch similar to Shafahi et al. (2020).

Following the recommendations of Shafahi et al. (2019) and Wong et al. (2020) we use $m = 8$ minibatch replays and train the model for 10 epochs. Unlike what we do for other baselines, we reduce the maximum learning rate of the optimizer to 0.04 following the advice of Wong et al. (2020) once more.

Shafahi et al. (2019) provide their TensorFlow code at `https://github.com/ashafahi/free_adv_train` and provide their PyTorch code at `https://github.com/mahyarnajibi/FreeAdversarialTraining`.

Wong et al. (2020) have also implemented Free AT using PyTorch at `https://github.com/locuslab/fast_adversarial`.

## B.3. FGSM-RS

Wong et al. (2020) introduced Fast Adversarial Training, in which each attack starts by adding uniform noise $\eta \sim \mathcal{U}(-\epsilon, +\epsilon)^d$ to the clean input. This perturbed sample then serves as the starting point for a standard FGSM update. The resulting perturbation $\delta$ is constrained within the $\ell_\infty$-norm ball of radius $\epsilon = {}^8\!/_{255}$. The method also requires a step-size hyperparameter, for which the authors recommend $\alpha = {}^{10}\!/_{255}$.

Wong et al. (2020) provide their code at `https://github.com/locuslab/fast_adversarial`.

## B.4. GradAlign

Andriushchenko & Flammarion (2020) proposed a regularization method that maximizes gradient alignment within the perturbation set. They define the following local linearity metric of the loss function:

$$\texttt{GradAlign}(x, y, \theta) := \cos\left(\nabla_x \mathcal{L}\big(f_\theta(x), y\big), \nabla_x \mathcal{L}\big(f_\theta(x + \eta), y\big)\right), \tag{11}$$

where $\eta \sim \mathcal{U}(-\epsilon, +\epsilon)^d$. They introduce a regularizer $\Omega(x, y, \theta) = 1 - \texttt{GradAlign}(x, y, \theta)$ and optimize the objective $\mathcal{L} + \lambda \Omega$, with $\lambda = 0.2$ fixed across all architectures and datasets, following the authors' recommendations for CIFAR-10 and CIFAR-100. Here, $\mathcal{L}$ denotes the cross-entropy loss computed on adversarial examples generated by FGSM-RS with $\alpha = 1.25 \times \epsilon$.

Andriushchenko & Flammarion (2020) have made their code publicly available at https://github.com/tml-epfl/understanding-fast-adv-training.

## B.5. NuAT

Inspired by Sriramanan et al. (2020), Sriramanan et al. (2021) introduced Nuclear-Norm Adversarial Training or NuAT. Their proposed defense incorporates a regularizer based on the nuclear norm in both attack generation and adversarial training:

$$\mathcal{L}(f_\theta(x), y) + \lambda \cdot \|f_\theta(x_{\text{adv}}) - f_\theta(x)\|_*$$

where $\|\cdot\|_*$ denotes the nuclear norm, defined as the sum of the singular values, and $x_{\text{adv}}$ is generated via the FGSM-RS attack. Sriramanan et al. (2021) generally use $\lambda = 4$, as this works for datasets that have around 10 classes. However, since for datasets with more classes the training collapses, we use $\lambda = 1$ so that the model can learn.

Sriramanan et al. (2021) provide their code at https://github.com/val-iisc/NuAT.

## B.6. ZeroGrad & MultiGrad

Golgooni et al. (2023) proposed ZeroGrad, a method that sets a threshold $q$ and zeros out the components of the input gradient whose absolute value falls below this threshold, thereby producing a more robust single-step perturbation. The method uses a random initialization $\eta \sim \mathcal{U}(-\epsilon, +\epsilon)^d$ and a step-size $\alpha$.

ZeroGrad is highly sensitive to the choice of $q$ across datasets, architectures, and training settings. For a fair comparison of generalizability among baselines, we follow the authors' recommendations and set $\alpha = 2 \times \epsilon$ and $q = 0.35$ for CIFAR-10, and $q = 0.45$ for CIFAR-100 and all other datasets.

Golgooni et al. (2023) also introduced MultiGrad, in which an identical batch is concatenated $N$ times, each copy initialized with $\eta \sim \mathcal{U}(-\epsilon, +\epsilon)^d$. Perturbations are then retained only in directions where all samples agree on the gradient sign. MultiGrad is less sensitive to hyperparameter choices, so we set $N = 3$ and $\alpha = 2 \times \epsilon$ in all experiments as per recommended by authors.

Golgooni et al. (2023) have made their implementation publicly available at https://github.com/rohban-lab/catastrophic_overfitting.

In Appendix C.3 we analyze these methods through the lens of second-order optimization, providing a fresh new look explaining how ZeroGrad and MultiGrad mitigate CO.

## B.7. N-FGSM

de Jorge Aranda et al. (2022) proposed an FGSM variant that initializes with larger random noise and omits clipping of the final perturbation $\delta$ to the $\ell_\infty$-norm ball of radius $\epsilon$. Following the authors' recommendation, we set $k = 2 \times \epsilon$ for the initialization $\eta \sim \mathcal{U}(-k, +k)^d$ and use a step-size of $\alpha = \epsilon$.

Please note that Schwinn et al. (2020) also have a method with the same name, which also doesn't clip the perturbation.

de Jorge Aranda et al. (2022) provide their code at https://github.com/pdejorge/N-FGSM.

## B.8. ATAS

Inspired by Zheng et al. (2020), Huang et al. (2023) introduced Adversarial Training with Adaptive Step-Sizes (ATAS), which maintains a moving average of the squared $\ell_2$-norm of the gradient:

$$v_i^j = \beta v_i^{j-1} + (1 - \beta) \left\| \nabla_{\tilde{x}_i} \mathcal{L}\big( f_\theta(\tilde{x}_i), y_i \big) \right\|_2^2,$$

where $\tilde{x}_i$ is the initialization of $x_i$, and $\beta$ is a momentum factor that stabilizes the step-size.

The per-example step-size $\alpha_i^j$ at epoch $j$ is then adjusted inversely to $v_i^j$:

$$\alpha_i^j = \frac{\gamma}{c + \sqrt{v_i^j}},$$

where $\gamma$ is a predefined learning rate and $c$ is a constant that prevents $\alpha_i^j$ from becoming excessively large. Following the authors' recommendations for CIFAR-10 and CIFAR-100, we set $\beta = 0.5$, $c = 0.01$, and $\gamma = 2c\epsilon$.

Huang et al. (2023) provide their code at `https://github.com/HuangZhiChao95/ATAS`.

## B.9. AAER

Lin et al. (2023) identified *Abnormal Adversarial Examples* (AAEs) as a primary cause of CO. Unlike *Normal Adversarial Examples* (NAEs), AAEs exhibit lower loss than their corresponding clean samples:

$$x^{\text{AAE}} := \mathcal{L}\big( f_\theta(x + \eta), y \big) > \mathcal{L}\big( f_\theta(x + \eta + \delta), y \big),$$
$$x^{\text{NAE}} := \mathcal{L}\big( f_\theta(x + \eta), y \big) \leq \mathcal{L}\big( f_\theta(x + \eta + \delta), y \big).$$

To mitigate the adverse effect of AAEs, they introduce a regularizer. For a batch of size $m$, let $n$ be the number of AAEs. The following terms penalize anomalous variation in AAEs and disparities in logits:

$$\text{AAE-CE} = \frac{1}{n} \sum_{i=1}^{n} \left[ \mathcal{L}\left( f_\theta(x_i^{\text{AAE}} + \eta), y_i \right) - \mathcal{L}\left( f_\theta(x_i^{\text{AAE}} + \eta + \delta), y_i \right) \right],$$

$$\text{AAE-L2} = \frac{1}{n} \sum_{i=1}^{n} \left\| f_\theta(x_i^{\text{AAE}} + \eta + \delta) - f_\theta(x_i^{\text{AAE}} + \eta) \right\|_2^2,$$

$$\text{NAE-L2} = \frac{1}{m - n} \sum_{j=1}^{m-n} \left\| f_\theta(x_j^{\text{NAE}} + \eta + \delta) - f_\theta(x_j^{\text{NAE}} + \eta) \right\|_2^2.$$

The *Abnormal Adversarial Examples Regularization* (AAER) term is defined as:

$$\text{AAER} = \left( \lambda_1 \cdot \frac{n}{m} \right) \cdot \left( \lambda_2 \cdot \text{AAE-CE} + \lambda_3 \cdot \max\big( \text{AAE-L2} - \text{NAE-L2}, 0 \big) \right).$$

Two variants were proposed: RS-AAER, based on FGSM-RS (Wong et al., 2020), and N-AAER, based on N-FGSM (de Jorge Aranda et al., 2022), both augmented with the AAER regularization term. In our experiments, we adopt N-AAER as the best-performing variant. Following the authors' recommendations for a fair generalizability comparison, we fix $\lambda_1 = 1$, $\lambda_2 = 1.5$, and $\lambda_3 = 0.15$ across all datasets and architectures.

Lin et al. (2023) provide their code at `https://github.com/tmllab/2023_NeurIPS_AAER`.

## B.10. ELLE

Rocamora et al. (2024) proposed *Efficient Local Linearity Enforcement* (ELLE), a regularization term designed to encourage local linearity. ELLE mitigates CO not only in standard AT evaluations, but also in more challenging scenarios such as large adversarial perturbations and extended training schedules. The regularization term is theoretically linked to the curvature of the loss function and requires three forward passes to compute, as follows:

$$x_a, x_b \sim x + \mathcal{U}\big( -\epsilon, +\epsilon \big)^d, \qquad \alpha \sim \mathcal{U}\big( 0, 1 \big),$$

where $d$ is the input dimensionality. A convex combination of $x_a$ and $x_b$ is then formed:

$$x_c = (1 - \alpha) \cdot x_a + \alpha \cdot x_b.$$

The ELLE penalty is given by:

$$E_{\text{lin}} = \big| \mathcal{L}\big(f_\theta(x_c), y^i\big) - (1 - \alpha) \cdot \mathcal{L}\big(f_\theta(x_a), y^i\big) - \alpha \cdot \mathcal{L}\big(f_\theta(x_b), y^i\big) \big|^2 .$$

Because the ELLE term can take on large values, an excessively high coefficient may cause numerical overflow in the model weights. Following the authors' recommendations for CIFAR-10 and CIFAR-100, we set the regularization coefficient to $\lambda = 1000$ in all experiments.

Rocamora et al. (2024) provide their code at `https://github.com/LIONS-EPFL/ELLE`.

### B.11. PGD

Madry et al. (2018) proposed Projected Gradient Descent (PGD) as a multi-step adversarial attack, widely regarded as a strong first-order adversary for both evaluation and training. PGD iteratively applies the gradient-sign method to maximize the loss with respect to the input, projecting intermediate updates back onto the perturbation set to ensure the adversarial example remains within the allowed $\ell_p$-norm constraint.

Given a clean example $x$, PGD generates an adversarial example by initializing from a random perturbation within $\Delta$ and performing $K$ steps:

$$x^{t+1} = \Pi_{x+\Delta} \left( x^t + \alpha \cdot \text{sign} \left( \nabla_x \mathcal{L}(f_\theta(x^t), y) \right) \right),$$

where $\Pi_{x+\Delta}(\cdot)$ denotes the projection onto $x + \Delta$, and $\alpha$ is the step-size.

In our experiments, we adopt PGD-2 and PGD-10 for generating adversarial examples during training with random starts. For PGD-2 we use $\alpha = {}^4/_{255}$ and for PGD-10 we use $\alpha = {}^2/_{255}$.

Madry et al. (2018) provide their code and pre-trained models at `https://github.com/MadryLab/mnist_challenge` and `https://github.com/MadryLab/cifar10_challenge`.

### B.12. TRADES

Zhang et al. (2019b) proposed TRadeoff-inspired Adversarial DEfense via Surrogate-loss minimization (TRADES), a regularized adversarial training framework that explicitly balances clean accuracy and robust accuracy. The method augments the standard adversarial training objective with a penalty that minimizes the Kullback–Leibler (KL) divergence between the model's output distributions on clean and adversarial examples, thereby encouraging prediction consistency under perturbations.

Formally, TRADES solves:

$$\min_\theta \mathbb{E}_{(x,y)\sim\mathcal{D}} \left[ \mathcal{L}\big(f_\theta(x), y\big) + \frac{1}{\lambda} \max_{\delta\in\Delta} D_{\text{KL}} \left( f_\theta(x) \,\|\, f_\theta(x + \delta) \right) \right], \tag{12}$$

where $\mathcal{L}$ denotes the standard cross-entropy loss on clean inputs, $\Delta$ is typically an $\ell_\infty$-norm ball of radius $\epsilon$, and $\lambda$ is a trade-off hyperparameter controlling the balance between natural and robust accuracy.

In practice, the inner maximization is performed using PGD-10, but with the KL divergence term replacing the cross-entropy loss used in standard PGD-based adversarial training. Following the authors' recommendations for CIFAR-10, we set $\beta = {}^1/\lambda = 6$ in all experiments.

Zhang et al. (2019b) release their code and trained models at `https://github.com/yaodongyu/TRADES`.

# C. Reinterpreting Previous Work via Our Framework

Here we dive deeper into previous works and and focus on some closely related ideas.

## C.1. Randomization-Based Methods

A key strength of randomization (noise-based) methods is their *step-size variability*. Andriushchenko & Flammarion (2020) justify the relative success of FGSM-RS (Wong et al., 2020) compared to FGSM (Goodfellow et al., 2015) by noting that the random noise and the subsequent clipping result in a smaller effective magnitude of the adversarial perturbation. This is further supported by the fact that adversarial perturbations with higher magnitudes often exacerbate CO. This, however, does not explain the success of N-FGSM (de Jorge Aranda et al., 2022), which only clips pixel values to the $[0, 1]$ range and ignores the general clipping step used to limit the total perturbation budget.

Evaluating these methods through the lens of Epsilon Overfitting (EO), alongside Table 2, clarifies the importance of step-size variability. Because randomization-based methods use a larger uniform noise magnitude, scaling this magnitude by a factor of $k > 1$ makes the resulting space exponentially larger, starting from a space that is $k^d$ times larger. Since $d$ is often quite large (e.g. for CIFAR-10 we have $d = 3 \times 32 \times 32 = 3072$), the effects of increasing $k$ even by a small amount are substantial (Vershynin, 2026). Omitting the clipping step combined with this expanded space significantly increases the diversity of perturbations, explaining the remarkable success of N-FGSM. These results indicate that the **variability** of the step-size is crucial. Specifically, the variation introduced through the sampling step makes the adversarial perturbation magnitudes more diverse, further limiting the possibility of EO occurring.

Moreover, the Epsilon Overfitting perspective argues that the model can memorize the exact perturbation budget and overfit to perturbations from that specific distribution, while exhibiting low accuracy on adversarial examples that share the same perturbation direction but have lower magnitudes (see Figure 2(a)). This observation suggests that for a given perturbation direction, there is a range of epsilon values where the model performs well, and another where it fails. Methods such as FGSM-RS (Wong et al., 2020) and N-FGSM (de Jorge Aranda et al., 2022) use randomization in the initial step to avoid deceptively good regions in favor of realistically bad regions, resulting in more robust models. Our method, however, does not address this problem randomly; instead, it relies on a systematic second-order approximation to find the optimal step-size and introduce variability. SORA utilizes second-order information about the loss landscape to determine whether a deceptively good region has emerged. When such a region is detected, SORA adapts by adjusting the step-size to target the vulnerable regions, thereby restoring the model's robustness.

## C.2. GradAlign versus PertAlign

Andriushchenko & Flammarion (2020) introduced the GradAlign regularizer, which measures the cosine similarity between the gradient of the loss with respect to the clean input and the gradient with respect to the clean input perturbed by random noise:

$$\texttt{GradAlign} := \cos\left(\nabla_x \mathcal{L}(f_\theta(x), y), \nabla_x \mathcal{L}(f_\theta(x + \eta), y)\right), \qquad \eta \sim \mathcal{U}(-\epsilon, +\epsilon)^d.$$

Lemma 4.1, which forms the theoretical basis for PertAlign, can also be applied to GradAlign. Without loss of generality, we can omit the explicit $\eta$ in Equation 2, viewing the noise $\eta$ in GradAlign as the arbitrary perturbation $v$ in Equation 2. This allows us to interpret GradAlign through a second-order approximation lens. We believe this perspective explains why PertAlign and GradAlign exhibit similar trends in Figure 3.

However, despite the similarity of their underlying metrics, our method differs from GradAlign in three key aspects:

1. **Motivation.** The primary motivation behind GradAlign is to increase gradient alignment *within* the perturbation budget. In contrast, our focus is on promoting the *linearity* of the loss surface along gradient directions. This imposes fewer constraints on the model and facilitates robustness in critical directions while better preserving clean accuracy.

2. **Usage.** GradAlign is designed as a *training regularizer*. In our case, PertAlign is used solely as a monitoring metric, enabling low-cost (practically zero) per-batch prediction of the CO status without influencing the training dynamics directly.

3. **Computation.** Since GradAlign serves as a regularizer, it must be computed before backpropagation to update the weights. In contrast, PertAlign leverages gradients already computed for attack generation and backpropagation

(extended one layer to include the inputs), adding virtually no extra computational or memory overhead (see Figures 7 and 16). In essence, we are using a similar trick to that of Shafahi et al. (2019) but avoid the hurdles of minibatch replaying since we don't use the free gradients directly in attack generation.

Fawzi et al. (2017); Moosavi-Dezfooli et al. (2019) note that adversarial directions correspond to high-curvature directions. This explains why GradAlign often has higher values in Figure 3 compared to PertAlign as the former measures the average curvature by sampling and random vector and the latter measures the curvature in the high-curvature adversarial directions. This also highlights the efficiency of PertAlign compared to GradAlign, even though from Figure 3 it could be inferred that during CO, the model suffers from high curvature in almost all directions as the gap between random and adversarial directions (GradAlign and PertAlign respectively) disappear. It should be noted that GradAlign acts as a regularizer, which overconstrains the model and reduces clean accuracy, especially when the direction is not adversarial. In contrast PertAlign is used only to identify nonlinearity, by indicating whether the chosen step-size has exceeded the locally smooth region, and adaptively modify the attack, without restricting the model.

### C.3. ZeroGrad & MultiGrad from a Second-Order Perspective

Golgooni et al. (2023) proposed ZeroGrad as an adversarial training method to prevent Catastrophic Overfitting (CO). It works by zeroing the perturbation components in directions where the corresponding gradient elements have very small magnitude. In this section, we examine ZeroGrad from a second-order theoretical perspective.

The ZeroGrad perturbation can be modeled as:

$$v = \alpha \odot p, \quad \alpha = \begin{pmatrix} \alpha_1 \\ \vdots \\ \alpha_d \end{pmatrix}, \quad p = \begin{pmatrix} \text{sign}(g_1) \\ \vdots \\ \text{sign}(g_d) \end{pmatrix},$$

where $\odot$ denotes the element-wise product. Each component $\alpha_i$ is determined by:

$$\forall i \in \{1, \cdots, d\} : \quad \alpha_i = \begin{cases} 1 & \text{if } \frac{|g_i|}{\|g\|} > \tau, \\ 0 & \text{otherwise.} \end{cases}$$

Since $\|v\| \ll \|x\|$, we can approximate the loss at the adversarial example $x + v$ using a second-order Taylor expansion:

$$L(x + v) = L(x + \alpha \odot p) \approx L(x) + g^T(\alpha \odot p) + \frac{1}{2}(\alpha \odot p)^T H(\alpha \odot p),$$

where $g = \nabla_x L(x)$ and $H = \nabla_x^2 L(x)$.

For the attack to succeed, we require:

$$L(x + \alpha \odot p) > L(x),$$

which is equivalent to:

$$g^T(\alpha \odot p) + \frac{1}{2}(\alpha \odot p)^T H(\alpha \odot p) > 0.$$

Since $H$ is symmetric, its eigendecomposition is:

$$H = Q^T \Lambda Q,$$

where $Q$ is orthogonal and $\Lambda$ is diagonal with eigenvalues $\lambda_i$. Substituting this into the inequality yields:

$$g^T(\alpha \odot p) + \frac{1}{2}(Q(\alpha \odot p))^T \Lambda Q(\alpha \odot p) = g^T(\alpha \odot p) + \frac{1}{2}h^T \Lambda h,$$

where $h = Q(\alpha \odot p)$. Component-wise, this becomes:

$$\sum_{i=1}^{d} \alpha_i p_i g_i + \frac{1}{2}\lambda_i h_i^2 = \sum_{i=1}^{d} \alpha_i |g_i| + \frac{1}{2}\lambda_i h_i^2.$$

For each term $i$ we have:

$$\alpha_i |g_i| + \frac{1}{2}\lambda_i h_i^2. \tag{13}$$

Because $\alpha_i |g_i| \geq 0$ and $h_i^2 \geq 0$, the sign of Equation 13 depends on the eigenvalue $\lambda_i$. If $\lambda_i < 0$ and

$$\lambda_i < -\frac{2\alpha_i |g_i|}{h_i^2},$$

then the $i$-th term is negative. If enough such terms are negative, the total sum becomes negative, meaning the adversarial example has lower loss than the clean sample, an *Abnormal Adversarial Example* (AAE). As noted by Lin et al. (2023), AAEs are closely linked to the onset of CO.

For components where $|g_i|/\|g\| \ll 1$, this condition is more easily satisfied, increasing the likelihood of generating AAEs. ZeroGrad mitigates this by setting $\alpha_i = 0$ whenever $|g_i|/\|g\| < \tau$, thereby avoiding perturbations in directions that could reduce the loss.

However, the magnitudes of $\lambda_i$ and $g_i$ vary substantially across datasets, models, and training stages. This variability directly affects the chosen $\tau$ hyperparameter (which closely relates to $q_{\text{val}}$ in the original paper by Golgooni et al. (2023)), which can lead to limited generalizability in diverse scenarios.

Golgooni et al. (2023) also proposed MultiGrad which tries to overcome shortcomings of ZeroGrad by only perturbing in directions where different random starts all show same gradient sign, but this is at the cost of more computational cost and memory consumption.

### C.4. PGD-2 and SORA

PertAlign computes the cosine similarity between gradients from the first two iterations of a PGD attack. Significant divergence between these gradients indicates a highly distorted and nonlinear loss surface. High alignment between consecutive PGD gradients suggests that the resulting perturbation could be approximated by a single-step attack. Conversely, when the loss surface becomes distorted, the alignment between PGD iterations decreases, and the optimization path can no longer be inferred from the initial gradient direction.

PGD variants, including PGD-2, avoid catastrophic failure by refining their results over multiple iterations. As shown in Figure 15, during catastrophic overfitting (CO), PertAlign collapses to zero, indicating that the second attack iteration moves in a direction nearly orthogonal to the first. This explains the substantial accuracy gap between single-step and multi-step methods after CO.

However, the small fixed step-size used in PGD, while beneficial in some contexts, introduces limitations. Fixed step-sizes can significantly reduce convergence speed compared to adaptive approaches. Auto-PGD (APGD) (Croce & Hein, 2020) addresses this by adapting step-sizes based on optimization progress and restarting from the best-found point when step-sizes are reduced. SORA builds upon these insights by using curvature information to estimate optimal step-sizes adaptively.

Another limitation of standard PGD is its use of a fixed $\epsilon$ budget. As noted by Ding et al. (2020), adaptive selection of $\epsilon$ for individual data points, treating it as a margin, can improve performance. SORA avoids the pitfalls of fixed-$\epsilon$ training identified by Ding et al. (2020), including the potential reduction of margins between clean samples and decision boundaries.

### C.5. FGSM-CKPT

Kim et al. (2021) observed that, upon the onset of CO, the loss landscape becomes distorted in the direction of single-step perturbations. They proposed FGSM-CKPT, which partitions the attack step-size $\alpha$ into $c$ discrete fractions. For each scaled step-size $i\alpha/c$, the perturbed input $x + i\alpha/c$ is fed to the model, and the index $i$ yielding the lowest accuracy is selected; the model is then updated using that perturbation.

Although effective in some cases, FGSM-CKPT requires multiple forward passes per training step, resulting in high computational cost. Moreover, restricting the step-size to a discrete set limits flexibility in selecting the optimal value. In this work, we analyze the loss landscape in greater detail and, leveraging observed distortion patterns, propose an adaptive mechanism that selects a more accurate step-size with effectively zero additional cost.

# D. Datasets

In this section we explore the efficacy of different methods on a number of different datasets. We provide a brief overview of the key aspects of each dataset in Table 3. The distribution of labels in each dataset can be seen in Figure 8.

*Table 3.* Datasets overview.

| Dataset | Dimensions | # Classes | # Training Samples | # Test Samples |
|---|---|---|---|---|
| CIFAR-10 | $3 \times 32 \times 32$ | 10 | 50,000 | 10,000 |
| CIFAR-100 | $3 \times 32 \times 32$ | 100 | 50,000 | 10,000 |
| TINYIMAGENET | $3 \times 64 \times 64$ | 200 | 100,000 | 10,000 |
| IMAGENET-100 | $3 \times 224 \times 224$ | 100 | 117,000 | 5,000 |
| PATHMNIST | $3 \times 28 \times 28$ | 9 | 89,996 | 7,180 |
| TISSUEMNIST | $1 \times 28 \times 28$ | 8 | 165,466 | 47,280 |

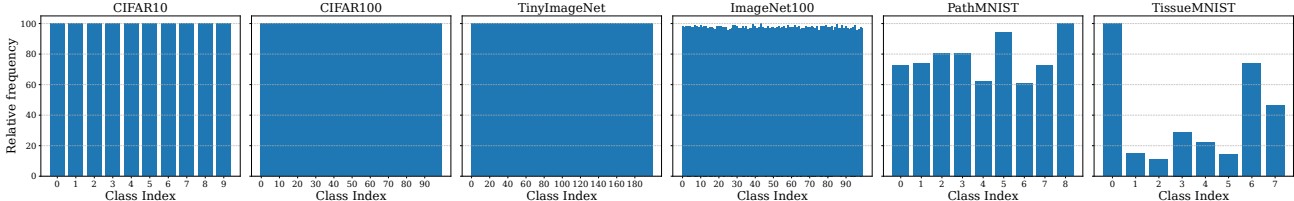

*Figure 8.* Class distributions across datasets. CIFAR-10, CIFAR-100, TINYIMAGENET, and IMAGENET-100 exhibit balanced class distributions, whereas PATHMNIST and TISSUEMNIST are imbalanced, with TISSUEMNIST showing the most pronounced imbalance.

**Dataset Selection Criteria**

In line with the previous work done on FAT, we included the CIFAR-10 and CIFAR-100 datasets which are the two most commonly used datasets in the community. Similarly, we included the TinyImageNet and ImageNet-100 datasets which are used less frequently, but they allow us to study larger datasets. Unlike previous works, we extend our datasets to the medical domain via the MEDMNIST suite (Yang et al., 2023).

The selection of PATHMNIST and TISSUEMNIST for our evaluation was guided by several principled considerations. First and foremost, we sought datasets with comparable scale to established benchmarks like CIFAR-10 and CIFAR-100, both in terms of sample size and image dimensions (Table 3). This ensures that computational requirements remain manageable while maintaining relevance to real-world applications.

Medical imaging datasets were prioritized for two key reasons:

1. Robustness in medical applications carries significant practical importance, where model reliability can directly impact diagnostic outcomes, and

2. the distribution shift between natural images (e.g. CIFAR, IMAGENET) and medical images provides a rigorous test of generalization capabilities.

The MEDMNIST suite (Yang et al., 2023) emerged as a natural candidate due to its standardized formatting and accessibility.

Our selection process was systematic rather than exhaustive. We reviewed available datasets in the MEDMNIST collection and selected PATHMNIST and TISSUEMNIST based on their *number of training samples*, *test samples*, and their *number of classes*. We explicitly did not perform extensive dataset screening or cherry-picking to favor our method; these datasets were chosen from a limited candidate pool based on the above criteria within our available time constraints.

Another dataset which has been used by previous work is the Street View House Numbers (SVHN) (Netzer et al., 2011). We excluded this dataset due to the label noise present in it. Since SVHN also has imbalanced classes, to preserve the diversity of our experiments, we relied on TISSUEMNIST which also has imbalanced classes, but to the best of our knowledge doesn't suffer from label noise. Therefore we preserved the diversity of our benchmarks without compromising their integrity.

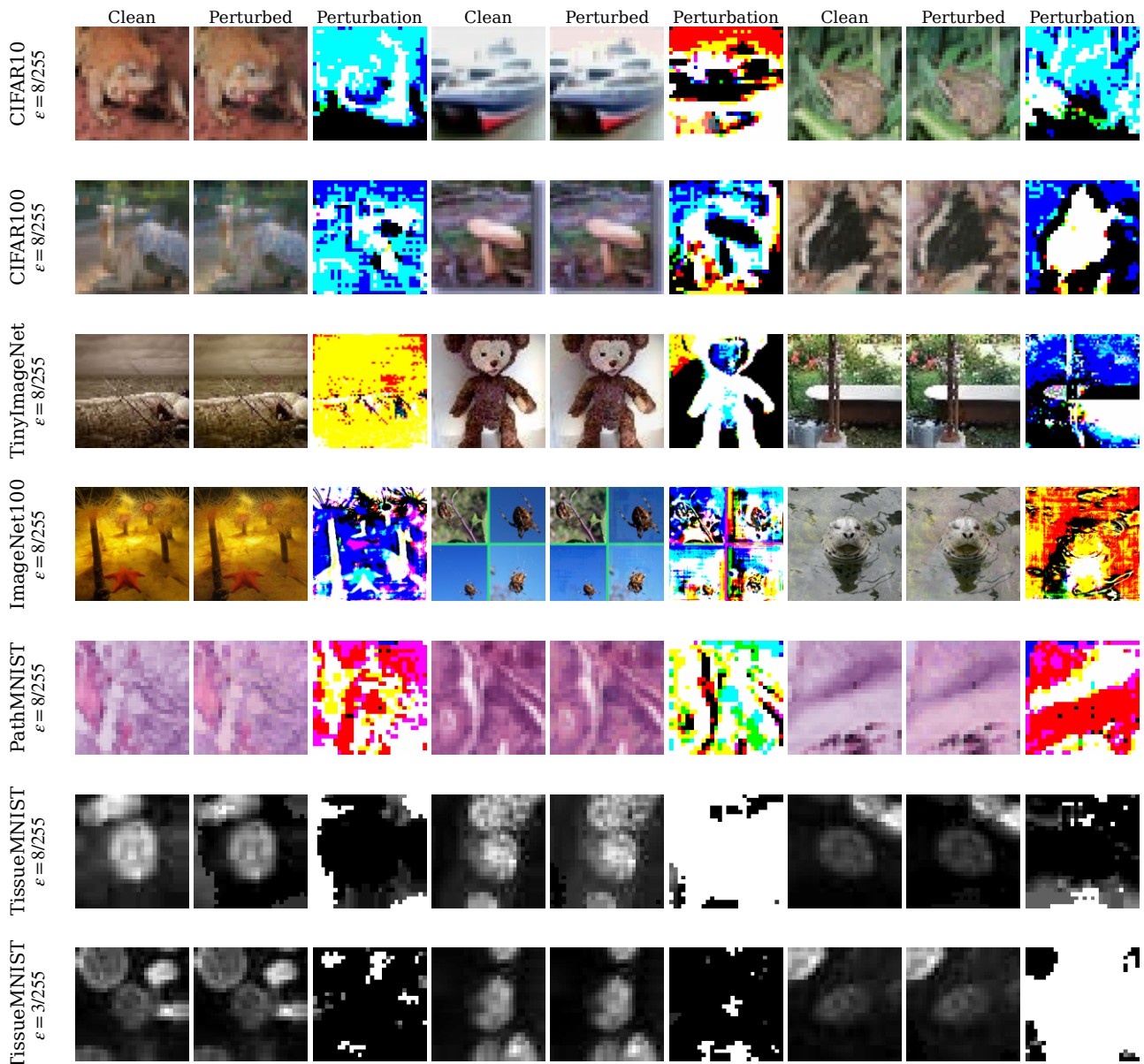

*Figure 9.* Dataset samples and their corresponding FGSM adversarial examples for a model adversarially trained with the SORA method. Each row corresponds to a dataset (CIFAR-10, CIFAR-100, TINYIMAGENET, IMAGENET-100, PATHMNIST, and TISSUEMNIST, respectively). The "Clean" columns show the original images, the "Perturbed" columns show the FGSM adversarial examples, and the "Perturbation" columns visualize the added perturbations, amplified for clarity.

## Selecting the Perturbation Budget

Szegedy et al. (2014) define adversarial examples as *imperceptible* non-random perturbations added to test images, capable of arbitrarily changing the network's prediction. In general, imperceptibly tiny perturbations of a given image do not normally change the underlying class for most computer vision problems (Szegedy et al., 2014). In order to abide by this convention of *imperceptibly*, accompanied by the desire to not accidentally change the true label of images by distorting them too much, we reduce the value of $\epsilon$ to $3/255$ for grayscale images of the TISSUEMNIST dataset. For the remaining datasets we use an $\epsilon$ value of $8/255$ similar to all previous works. Samples from each dataset alongside their corresponding adversarial perturbations can be viewed in Figure 9.

## D.1. CIFAR-10

The CIFAR-10 dataset (Krizhevsky, 2009) consists of 60,000 $32 \times 32$ color images across 10 classes, with 6,000 images per class. The dataset is divided into 50,000 training images and 10,000 test images. The test set contains exactly 1,000 randomly-selected images from each class, while the training batches contain the remaining images with some variation in class distribution per batch, though each class has exactly 5,000 training images in total.

Following the instructions of Wong et al. (2020), we use padding of size 4 and perform random crops, in addition to random horizontal flips on the training data. Both the training and test data are normalized with respect to the mean and standard deviation of the training data. For all models and architectures we use an SGD optimizer with momentum 0.9 and weight decay $5 \times 10^{-4}$. A simple cyclic learning rate schedules the learning rate linearly from 0.01, to a maximum learning rate of 0.2, and back down to 0.01.

## D.2. CIFAR-100

The CIFAR-100 dataset (Krizhevsky, 2009) contains 60,000 $32 \times 32$ color images across 100 fine-grained classes, which are grouped into 20 superclasses. Each class contains 600 images, with 500 training and 100 testing images per class. Each image is annotated with both a fine label (specific class) and a coarse label (superclass).

Following the instructions of Wong et al. (2020), we use padding of size 4 and perform random crops, in addition to random horizontal flips on the training data. Both the training and test data are normalized with respect to the mean and standard deviation of the training data. For all models and architectures we use an SGD optimizer with momentum 0.9 and weight decay $5 \times 10^{-4}$. A simple cyclic learning rate schedules the learning rate linearly from 0.01, to a maximum learning rate of 0.2, and back down to 0.01.

## D.3. TinyImageNet

The TINYIMAGENET dataset (Le & Yang, 2015) is a downsampled version of IMAGENET (Deng et al., 2009; Russakovsky et al., 2015) containing 100,000 $64 \times 64$ color images across 200 classes. Each class has 500 training images, 50 validation images, and 50 test images, providing a more computationally manageable alternative to the full ImageNet dataset while maintaining the multi-class classification challenge.

Following the instructions of Wong et al. (2020) and Lin et al. (2023), we use padding of size 8 and perform random crops, in addition to random horizontal flips on the training data. Both the training and test data are normalized with respect to the mean and standard deviation of the training data. For all models and architectures we use an SGD optimizer with momentum 0.9 and weight decay $5 \times 10^{-4}$. A simple cyclic learning rate schedules the learning rate linearly from 0.01, to a maximum learning rate of 0.2, and back down to 0.01.

## D.4. ImageNet-100

Our results on the IMAGENET introduced by Deng et al. (2009); Russakovsky et al. (2015) are as follows. The ImageNet dataset contains 14,197,122 annotated images according to the WordNet hierarchy. Since 2010 the dataset is used in the IMAGENET Large Scale Visual Recognition Challenge (ILSVRC), a benchmark in image classification and object detection. The publicly released dataset contains a set of manually annotated training images. A set of test images is also released, with the manual annotations withheld. ILSVRC annotations fall into one of two categories: (1) image-level annotation of a binary label for the presence or absence of an object class in the image, e.g., "there are cars in this image" but "there are no tigers," and (2) object-level annotation of a tight bounding box and class label around an object instance in the image, e.g., "there is a screwdriver centered at position (20,25) with width of 50 pixels and height of 30 pixels". The IMAGENET project does not own the copyright of the images, therefore only thumbnails and URLs of images are provided.

For specifying the 100 classes used in our training we used this repository from huggingface `https://huggingface.co/datasets/ilee0022/ImageNet100/`. Due to the higher image quality in the IMAGENET-100 dataset, we reduced the batch size to 32. This reduced our memory usage, allowing us to proceed with our experiments. We also used random crops, in addition to random horizontal flips on the training data. Both the training and test data are normalized with respect to the mean and standard deviation of the training data. For all models and architectures we use an SGD optimizer with momentum 0.9 and weight decay $5 \times 10^{-4}$. A simple cyclic learning rate schedules the learning rate linearly from 0.01, to a maximum learning rate of 0.2, and back down to 0.01.

### D.5. PathMNIST

The PATHMNIST dataset (Yang et al., 2023) is a medical image classification benchmark for colorectal cancer detection. It contains 107,180 $28 \times 28$ color histological images across 9 tissue classes. The dataset is split into 89,996 training images, 10,004 validation images, and 7,180 test images. Unlike the balanced natural image datasets, PATHMNIST exhibits natural class imbalance reflective of real-world medical data distributions.

We perform random rotations with 10 degrees, in addition to random horizontal flips on the training data. Both the training and test data are normalized with respect to the mean and standard deviation of the training data. For all models and architectures we use an SGD optimizer with momentum $0.9$ and weight decay $5 \times 10^{-4}$. To further stabilize training, we use a cosine annealing learning rate scheduler that starts from the initial learning rate of $0.05$ and the minimum learning rate of $10^{-3}$. This scheduler improves the results across all settings.

### D.6. TissueMNIST

The TISSUEMNIST dataset (Yang et al., 2023) is based on the broadly annotated tissue atlas of human gene expression, containing 236,386 $28 \times 28$ grayscale images of human kidney cortex cells across 8 tissue classes. The dataset is split into 165,466 training images, 23,640 validation images, and 47,280 test images. TISSUEMNIST exhibits significant class imbalance, making it particularly challenging for evaluation and representative of real-world medical imaging scenarios.

We perform random rotations with 10 degrees, in addition to random horizontal flips on the training data. Both the training and test data are normalized with respect to the mean and standard deviation of the training data. For all models and architectures we use an SGD optimizer with momentum $0.9$ and weight decay $5 \times 10^{-4}$. To further stabilize training, we use a cosine annealing learning rate scheduler that starts from the initial learning rate of $0.05$ and the minimum learning rate of $10^{-3}$. This scheduler improves the results for all results.

# E. Architectures

In this section, we describe the architectures employed in our experiments. We include classical convolutional backbones such as ResNets (He et al., 2016a), PreActResNets (He et al., 2016b), and WideResNets (Zagoruyko & Komodakis, 2017), as well as more recent variants like SENets (Hu et al., 2018) and Transformer-based models such as the Vision Transformer (Dosovitskiy et al., 2021). This diversity enables us to assess performance across architectures with distinct inductive biases.

**Architecture Selection Criteria**

In line with the previous work done on FAT, we included the PreActResNet and WideResNet architectures. These two form the basis for the majority of the work done in this field. We also include the ResNet architecture which is also used by the literature, although its usage is less frequent. The SENet architecture was added to increase architectural diversity whilst sharing similarities to the ResNet family. We also include the Vision Transformer architecture to evaluate methods on more modern attention-based architectures.

## E.1. Residual Networks

Residual Networks (ResNets) introduced by He et al. (2016a) remain a cornerstone of deep learning due to their ability to train very deep models effectively. The key innovation is the residual connection, which mitigates the vanishing gradient problem by providing identity shortcuts for direct gradient propagation. This enables deeper networks to converge more reliably, enhancing representational capacity without significant optimization challenges. In our experiments, we use ResNet-18, which offers a balance between computational efficiency and representational power.

## E.2. Pre-Activation Residual Networks

Pre-Activation Residual Networks (PreActResNets) (He et al., 2016b) refine the original ResNet architecture by placing batch normalization and ReLU activations before the convolutional layers. This modification improves gradient flow and facilitates optimization, particularly in deeper architectures. We evaluate PreActResNet-18, which maintains the overall structure of standard ResNets while benefiting from more stable training dynamics.

## E.3. Wide Residual Networks

Wide Residual Networks (WideResNets) (Zagoruyko & Komodakis, 2017) challenge the trend of increasing depth by instead widening residual blocks through additional channels. This approach enhances model capacity while reducing training time compared to deeper, narrower networks. We employ WideResNet-28-10, where 28 indicates the number of convolutional layers and 10 is the widening factor applied to channels in each residual block relative to a standard ResNet.

## E.4. Squeeze-and-Excitation Networks

Squeeze-and-Excitation Networks (SENets) (Hu et al., 2018) enhance convolutional architectures with channel-wise attention mechanisms. By adaptively recalibrating feature responses, SENets emphasize informative channels and suppress less useful ones, improving performance across various recognition tasks. We use SENet-18, which integrates squeeze-and-excitation blocks into a standard ResNet-18, providing an effective attention-enhanced backbone with minimal computational overhead.

## E.5. Vision Transformers

The Vision Transformer (ViT) (Dosovitskiy et al., 2021) represents a paradigm shift from convolutional architectures by directly applying a Transformer encoder to sequences of image patches. This design eliminates the locality bias of convolutions and instead leverages global self-attention, enabling the model to capture long-range dependencies across the entire image. We adopt the standard ViT architecture as introduced by Dosovitskiy et al. (2021), which has proven effective across diverse vision benchmarks and provides a complementary inductive bias compared to convolutional networks.

Following Lin et al. (2024), in our experiments we employ the ViT-Small variant, which differ primarily in model size, hidden dimension, and number of attention heads. ViT-Small offers a more lightweight configuration suitable for efficiency-focused settings. Our results are available in Appendix L. Wu et al. (2022); Salmani et al. (2023) explore FAT on ViTs with pre-training.

# F. Ablations

## F.1. Batch Size

We run our SORA algorithm with different batch sizes on the CIFAR-10 dataset. You can see the results for our method and N-FGSM in Table 4.

*Table 4.* **Ablation on batch size.** The values in each cell correspond to clean and PGD-10 accuracies.

| Method | Batch Size 32 | Batch Size 64 | Batch Size 128 | Batch Size 256 | Batch Size 512 |
|---|---|---|---|---|---|
| N-FGSM | $73.40 \pm 0.08$ | $77.15 \pm 0.29$ | $79.05 \pm 0.21$ | $79.95 \pm 0.15$ | $78.69 \pm 0.21$ |
| | $46.02 \pm 0.16$ | $48.15 \pm 0.42$ | $48.73 \pm 0.44$ | $47.84 \pm 0.37$ | $46.91 \pm 0.09$ |
| SORA (Ours) | $74.74 \pm 0.36$ | $78.60 \pm 0.44$ | $79.90 \pm 0.05$ | $80.80 \pm 0.16$ | $79.86 \pm 0.17$ |
| | $46.38 \pm 0.18$ | $48.28 \pm 0.81$ | $48.64 \pm 0.44$ | $47.85 \pm 0.16$ | $47.00 \pm 0.08$ |

## F.2. SORA Hyperparameter Search

SORA has three hyperparameters (i.e. $\beta$, $\alpha_0$, and $\alpha_{\max}$). Since SORA uses per-channel randomization, the average maximum step-size is approximately half of $\alpha_{\max}$. Thus we set $\alpha_{\max} = 2\epsilon$ so that the expected maximum size is around $\epsilon$ which is our attack budget.

$\beta$ and $\alpha_0$ parameters control the second-order adaptive component of SORA's step-size calculation. $\beta$ specifically, controls the speed at which we modify our curvature estimation. During our experiments on CIFAR-10/100, SORA consistently used maximum attack step-size, $\alpha_{\max}$, indicating that these datasets were not challenging enough to lower the step-size. Thus for calibrating these two parameters we used PATHMNIST.

*Table 5.* SORA: Hyperparameter sweep on CIFAR-10 (clean accuracy)

| | $\alpha_{\max} = 1.5\epsilon$ | | | $\alpha_{\max} = 2.0\epsilon$ | | | $\alpha_{\max} = 2.5\epsilon$ | | |
|---|---|---|---|---|---|---|---|---|---|
| $\beta$ | $\alpha_0 = 0.01$ | $\alpha_0 = 0.02$ | $\alpha_0 = 0.05$ | $\alpha_0 = 0.01$ | $\alpha_0 = 0.02$ | $\alpha_0 = 0.05$ | $\alpha_0 = 0.01$ | $\alpha_0 = 0.02$ | $\alpha_0 = 0.05$ |
| 0.02 | 83.54 | 83.91 | 83.75 | 80.20 | 79.78 | 80.22 | 75.76 | 75.36 | 75.44 |
| 0.05 | 83.92 | 83.97 | 83.73 | 79.84 | 79.84 | 79.85 | 76.00 | 75.87 | 75.90 |
| 0.10 | 84.13 | 83.68 | 83.83 | 80.17 | 79.79 | 80.08 | 75.99 | 75.81 | 75.91 |

*Table 6.* SORA: Hyperparameter sweep on CIFAR-10 (PGD-10 accuracy)

| | $\alpha_{\max} = 1.5\epsilon$ | | | $\alpha_{\max} = 2.0\epsilon$ | | | $\alpha_{\max} = 2.5\epsilon$ | | |
|---|---|---|---|---|---|---|---|---|---|
| $\beta$ | $\alpha_0 = 0.01$ | $\alpha_0 = 0.02$ | $\alpha_0 = 0.05$ | $\alpha_0 = 0.01$ | $\alpha_0 = 0.02$ | $\alpha_0 = 0.05$ | $\alpha_0 = 0.01$ | $\alpha_0 = 0.02$ | $\alpha_0 = 0.05$ |
| 0.02 | 45.60 | 44.87 | 45.54 | 48.48 | 48.30 | 48.36 | 49.49 | 49.31 | 49.15 |
| 0.05 | 45.17 | 45.70 | 45.15 | 48.27 | 48.70 | 48.51 | 49.30 | 49.36 | 49.58 |
| 0.10 | 45.18 | 45.50 | 45.50 | 48.22 | 48.85 | 48.62 | 49.51 | 49.59 | 49.55 |

*Table 7.* SORA: Hyperparameter sweep on CIFAR-100 (clean accuracy)

| | $\alpha_{\max} = 1.5\epsilon$ | | | $\alpha_{\max} = 2.0\epsilon$ | | | $\alpha_{\max} = 2.5\epsilon$ | | |
|---|---|---|---|---|---|---|---|---|---|
| $\beta$ | $\alpha_0 = 0.01$ | $\alpha_0 = 0.02$ | $\alpha_0 = 0.05$ | $\alpha_0 = 0.01$ | $\alpha_0 = 0.02$ | $\alpha_0 = 0.05$ | $\alpha_0 = 0.01$ | $\alpha_0 = 0.02$ | $\alpha_0 = 0.05$ |
| 0.02 | 58.54 | 58.53 | 57.85 | 54.26 | 54.46 | 53.80 | 51.69 | 50.82 | 50.85 |
| 0.05 | 58.03 | 58.41 | 58.06 | 53.93 | 54.06 | 53.98 | 52.56 | 50.98 | 50.33 |
| 0.10 | 58.47 | 57.84 | 58.21 | 54.10 | 53.94 | 53.94 | 52.65 | 51.26 | 50.87 |

*Table 8.* SORA: Hyperparameter sweep on CIFAR-100 (PGD-10 accuracy)

| | $\alpha_{\max} = 1.5\epsilon$ | | | $\alpha_{\max} = 2.0\epsilon$ | | | $\alpha_{\max} = 2.5\epsilon$ | | |
| $\beta$ | $\alpha_0 = 0.01$ | $\alpha_0 = 0.02$ | $\alpha_0 = 0.05$ | $\alpha_0 = 0.01$ | $\alpha_0 = 0.02$ | $\alpha_0 = 0.05$ | $\alpha_0 = 0.01$ | $\alpha_0 = 0.02$ | $\alpha_0 = 0.05$ |
|---|---|---|---|---|---|---|---|---|---|
| 0.02 | 24.58 | 24.35 | 24.33 | 26.34 | 26.21 | 26.26 | 26.89 | 27.05 | 27.40 |
| 0.05 | 24.53 | 24.63 | 24.50 | 26.51 | 26.29 | 26.20 | 27.14 | 27.05 | 27.47 |
| 0.10 | 24.48 | 24.81 | 24.30 | 26.36 | 26.41 | 26.14 | 26.76 | 27.00 | 27.38 |

*Table 9.* SORA: Hyperparameter sweep on PATHMNIST (clean accuracy)

| | $\alpha_{\max} = 1.5\epsilon$ | | | $\alpha_{\max} = 2.0\epsilon$ | | | $\alpha_{\max} = 2.5\epsilon$ | | |
| $\beta$ | $\alpha_0 = 0.01$ | $\alpha_0 = 0.02$ | $\alpha_0 = 0.05$ | $\alpha_0 = 0.01$ | $\alpha_0 = 0.02$ | $\alpha_0 = 0.05$ | $\alpha_0 = 0.01$ | $\alpha_0 = 0.02$ | $\alpha_0 = 0.05$ |
|---|---|---|---|---|---|---|---|---|---|
| 0.02 | 84.99 | 85.68 | 86.70 | 84.68 | 87.26 | 87.19 | 85.04 | 85.50 | 86.04 |
| 0.05 | 85.15 | 85.58 | 84.83 | 85.06 | 84.69 | 90.42 | 85.03 | 84.87 | 75.07 |
| 0.10 | 85.33 | 85.36 | 84.33 | 86.48 | 84.94 | 89.42 | 86.16 | 86.48 | 84.35 |

*Table 10.* SORA: Hyperparameter sweep on PATHMNIST (PGD-10 accuracy)

| | $\alpha_{\max} = 1.5\epsilon$ | | | $\alpha_{\max} = 2.0\epsilon$ | | | $\alpha_{\max} = 2.5\epsilon$ | | |
| $\beta$ | $\alpha_0 = 0.01$ | $\alpha_0 = 0.02$ | $\alpha_0 = 0.05$ | $\alpha_0 = 0.01$ | $\alpha_0 = 0.02$ | $\alpha_0 = 0.05$ | $\alpha_0 = 0.01$ | $\alpha_0 = 0.02$ | $\alpha_0 = 0.05$ |
|---|---|---|---|---|---|---|---|---|---|
| 0.02 | 44.21 | 45.06 | 40.38 | 42.98 | 35.84 | 23.72 | 44.65 | 39.04 | 36.49 |
| 0.05 | 42.98 | 43.59 | 46.85 | 44.01 | 45.56 | 19.90 | 44.00 | 44.50 | 25.57 |
| 0.10 | 44.22 | 43.04 | 45.22 | 41.87 | 44.47 | 21.78 | 37.72 | 43.48 | 45.35 |

Since $\alpha_{\max}$ acts as an upper bound for $\alpha^*$, setting $\alpha_{\max} \approx 2\epsilon$ will yield on average step-sizes equal to the perturbation budget $\epsilon$ used by other baselines when the PertAlign metric is high enough. While we use $\alpha_{\max}$ to clip step-sizes, when the PertAlign metric is low enough to trigger the adaptive step-size mechanism, a scaling factor $\alpha_0$ is also utilized. Tables 5, 6, 7, and 8 therefore exhibit near identical performance as we change the value of $\alpha_0$ for CIFAR-10/100 datasets, but for the challenging PATHMNIST we can clearly see the effects of using different $\alpha_0$ values, especially for large $\alpha_{\max}$.

### F.3. Different Epsilons at Training

We trained our SORA algorithm with different values of $\epsilon$ during training on the CIFAR-10 dataset. The results for this experiment are reported in Table 11. Since SORA uses adaptive step-sizes, for smaller perturbation budgets it can fully exploit the allotted space. For larger values of epsilon it manages to detect the most effective step-size to avoid CO and improve robustness.

*Table 11.* **Ablation on training with different epsilons.** The values in each cell correspond to clean and PGD-10 accuracies.

| Method | $\epsilon = 4/255$ | $\epsilon = 8/255$ | $\epsilon = 12/255$ | $\epsilon = 16/255$ |
|---|---|---|---|---|
| ATAS | $90.63 \pm 0.26$ $61.53 \pm 0.17$ | $87.25 \pm 0.13$ $41.91 \pm 0.38$ | $86.89 \pm 1.83$ $12.08 \pm 9.60$ | $83.89 \pm 1.45$ $8.86 \pm 6.29$ |
| MultiGrad | $87.56 \pm 0.07$ $66.49 \pm 0.22$ | $80.62 \pm 0.22$ $43.33 \pm 0.25$ | $73.50 \pm 0.36$ $36.65 \pm 0.63$ | $74.87 \pm 3.70$ $0.00 \pm 0.00$ |
| N-FGSM | $87.40 \pm 0.18$ $66.44 \pm 0.20$ | $79.05 \pm 0.21$ $48.73 \pm 0.44$ | $69.36 \pm 0.35$ $37.54 \pm 0.63$ | $61.20 \pm 0.57$ $29.23 \pm 0.59$ |
| SORA (Ours) | $87.85 \pm 0.04$ $66.86 \pm 0.30$ | $79.90 \pm 0.05$ $48.64 \pm 0.44$ | $72.18 \pm 0.46$ $37.32 \pm 0.42$ | $67.66 \pm 0.21$ $27.12 \pm 0.36$ |

## F.4. Different Epsilons at Test Time

We report the accuracy of different methods against different values of $\epsilon$ for FGSM and PGD attacks in Table 12. Since FGSM AT suffers from CO it dominates the other methdos in terms of FGSM accuracy but is not actually robust as indicated by its PGD accuracy. Other methods also struggle to match the performance of SORA across all values of $\epsilon$. Methods like ELLE and GradAlign perform better than our SORA for smaller $\epsilon$ values but their performance deteriorates rapidly when we move to larger values. SORA retains its accuracy and adapts better to attacks with different perturbation budgets.

*Table 12.* Training results for PreActResNet-18 on the CIFAR-10.

| Method | $\epsilon = 3/255$ | | $\epsilon = 5/255$ | | $\epsilon = 9/255$ | | $\epsilon = 13/255$ | | $\epsilon = 17/255$ | |
| --- | --- | --- | --- | --- | --- | --- | --- | --- | --- | --- |
| | FGSM | PGD | FGSM | PGD | FGSM | PGD | FGSM | PGD | FGSM | PGD |
| FGSM | 38.17 | 0.19 | 91.33 | 0.04 | 58.07 | 0.04 | 49.33 | 0.07 | 41.18 | 0.04 |
| FGSM-RS | 72.47 | 71.76 | 64.77 | 62.72 | 49.29 | 41.89 | 38.17 | 24.40 | 30.17 | 10.64 |
| GradAlign | 72.02 | 71.61 | 63.84 | 61.64 | 49.74 | 42.04 | 37.91 | 23.44 | 29.65 | 10.49 |
| ZeroGrad | 71.35 | 70.72 | 63.88 | 61.98 | 50.37 | 43.75 | 40.40 | 25.82 | 31.73 | 12.50 |
| MultiGrad | 70.50 | 70.05 | 63.24 | 61.53 | 50.63 | 43.64 | 39.96 | 26.56 | 31.36 | 12.72 |
| N-FGSM | 69.31 | 69.08 | 62.83 | 61.35 | 50.26 | 44.31 | 39.73 | 27.94 | 31.36 | 14.21 |
| ATAS | 75.37 | 74.26 | 65.40 | 60.83 | 47.92 | 36.61 | 35.16 | 16.59 | 26.19 | 6.21 |
| AAER | 69.49 | 69.35 | 63.02 | 61.42 | 50.60 | 44.64 | 40.10 | 28.31 | 31.70 | 14.06 |
| ELLE | 72.14 | 71.35 | 64.69 | 61.90 | 48.51 | 39.69 | 36.05 | 20.09 | 26.08 | 7.96 |
| SORA (Ours) | 69.41 | 68.97 | 63.95 | 62.72 | 51.78 | 43.97 | 40.40 | 28.45 | 32.92 | 14.17 |
| PGD-10 | 70.87 | 70.50 | 64.14 | 63.10 | 50.86 | 45.94 | 40.66 | 30.32 | 31.47 | 16.96 |
| TRADES | 69.46 | 69.05 | 62.50 | 61.16 | 50.60 | 45.87 | 40.36 | 32.59 | 32.51 | 20.05 |

## F.5. Epsilon Overfitting

We demonstrate the occurrence of Epsilon Overfitting (EO) under varying training perturbation magnitudes and across different datasets.

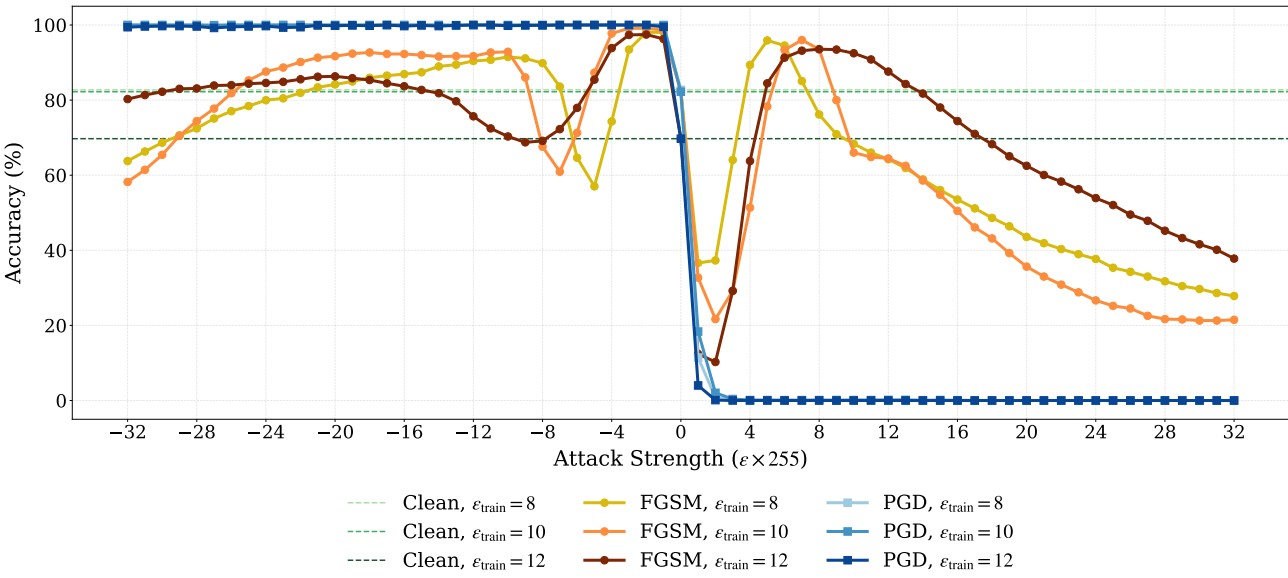

*Figure 10.* Overlay of FGSM accuracies showing the effect of different training $\epsilon$ values on the EO peak location and sharpness.

Varying the training value of $\epsilon$ influences the peak of FGSM accuracy following CO Figure 10. Larger $\epsilon$ values shift these peaks toward higher perturbation magnitudes, whereas smaller values produce sharper peaks.

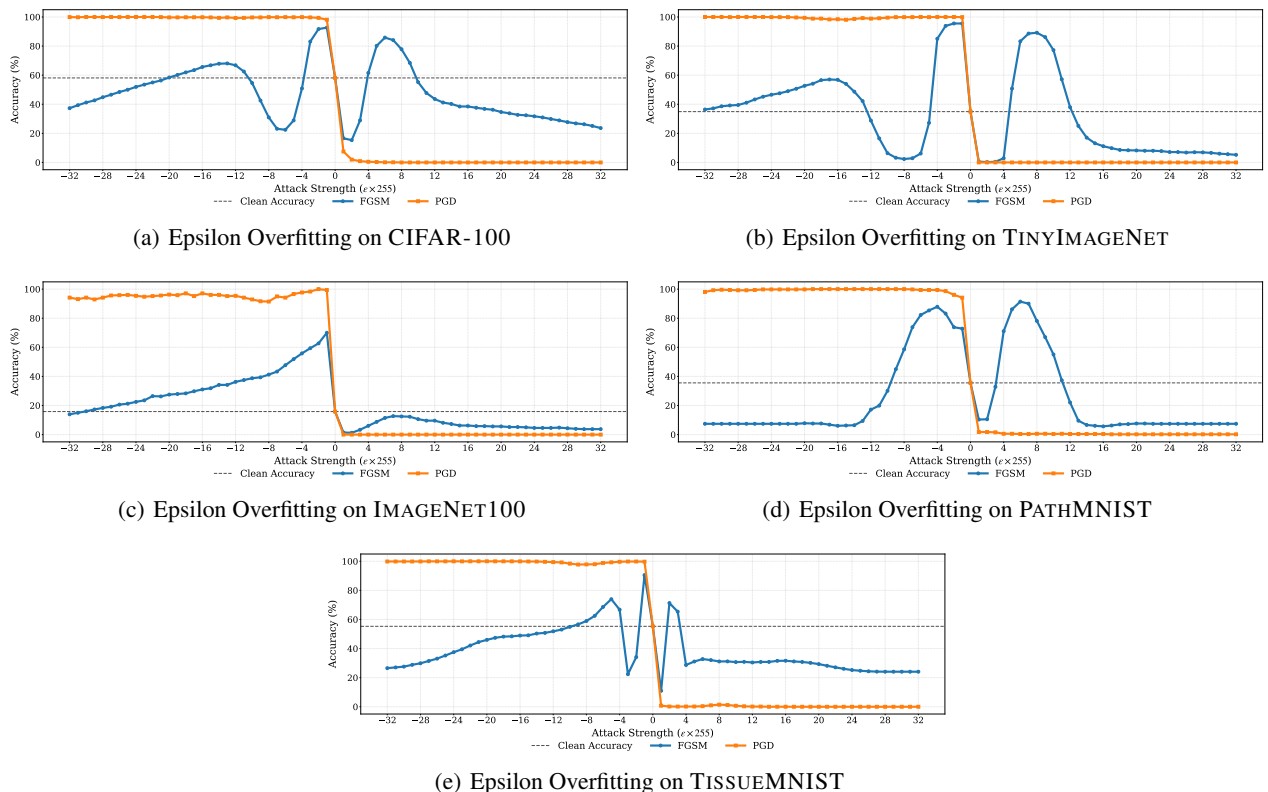

(a) Epsilon Overfitting on CIFAR-100

(b) Epsilon Overfitting on TINYIMAGENET

(c) Epsilon Overfitting on IMAGENET100

(d) Epsilon Overfitting on PATHMNIST

(e) Epsilon Overfitting on TISSUEMNIST

*Figure 11.* Examples of EO occurrence across datasets with models exhibiting CO.

EO is also evident across multiple datasets, as shown in Figure 11.

### F.6. Monitoring $\alpha^*$ in Training

Similar to Figure 5, we monitor the theoretically optimal step-size $\alpha^*$ from Lemma 5.1 across different datasets and architectures, as shown in Figure 12.

In each setting, SORA adaptively adjusts its step-size distribution using $\alpha^*$ to generate stronger adversarial examples while avoiding EO and CO. For datasets such as CIFAR-10 and CIFAR-100, SORA can select the maximum allowable step-size more consistently and confidently without the occurrence of CO (Figure 12(g), Figure 12(a)). In contrast, for datasets such as IMAGENET-100 and PATHMNIST, it must readjust the step-size more frequently and rapidly (Figure 12(c), Figure 12(h)). The behavior of $\alpha^*$ is also influenced by the choice of architecture, even when the dataset is fixed, as illustrated by the difference between Figure 12(b) and Figure 12(e).

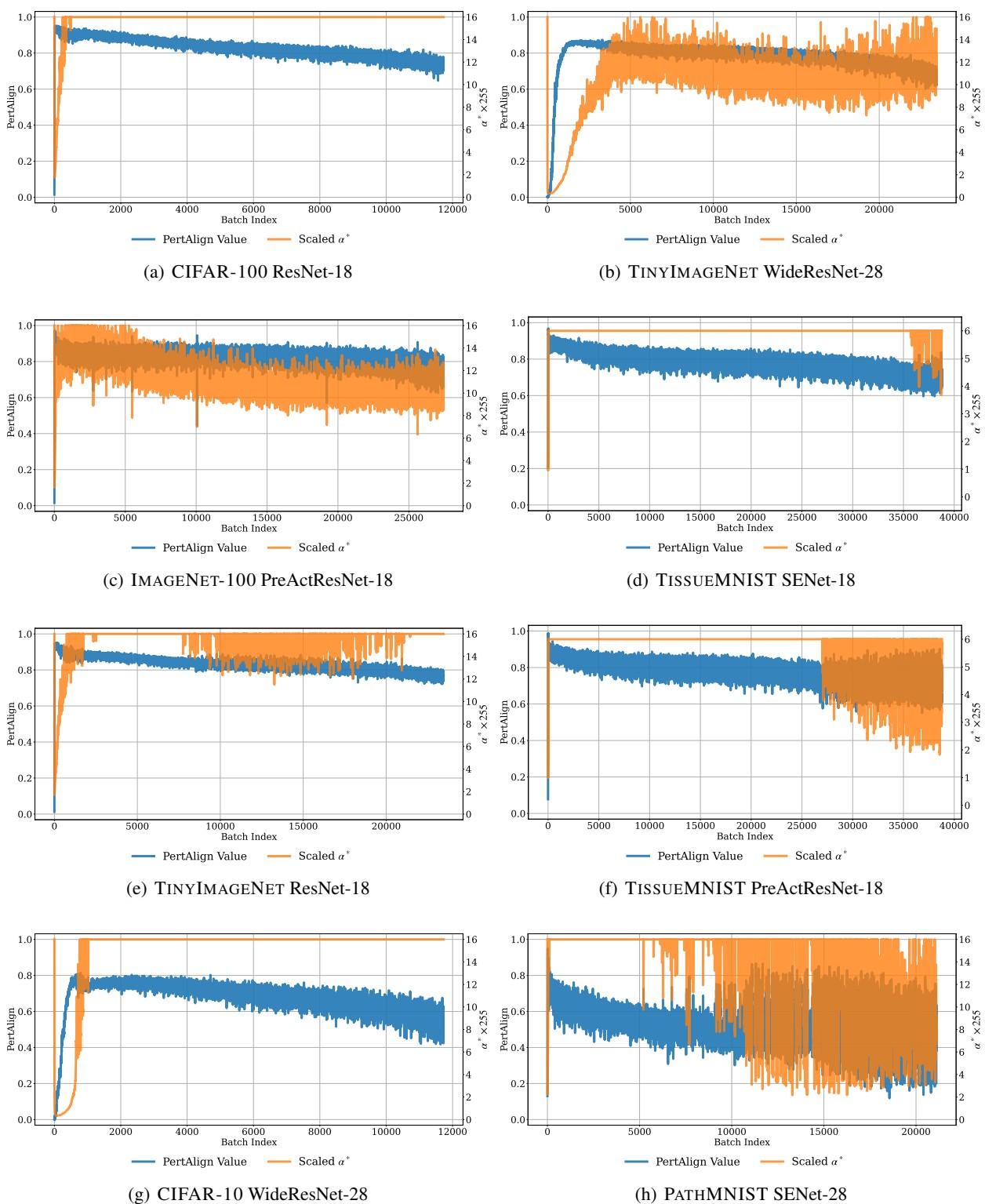

*Figure 12.* Tracking SORA's $\alpha^*$ across datasets and models.

# G. Loss Landscape Visualization

In this section, we visualize the loss landscapes for the CIFAR-10, PATHMNIST, and TISSUEMNIST datasets under three scenarios: benign training, CO, and robust training, as shown in Figure 13. Each unit in the $xy$ plane corresponds to one FGSM perturbation ($8/255 \times \text{sign}(g)$ for CIFAR-10 and PATHMNIST and $3/255 \times \text{sign}(g)$ for TISSUEMNIST).

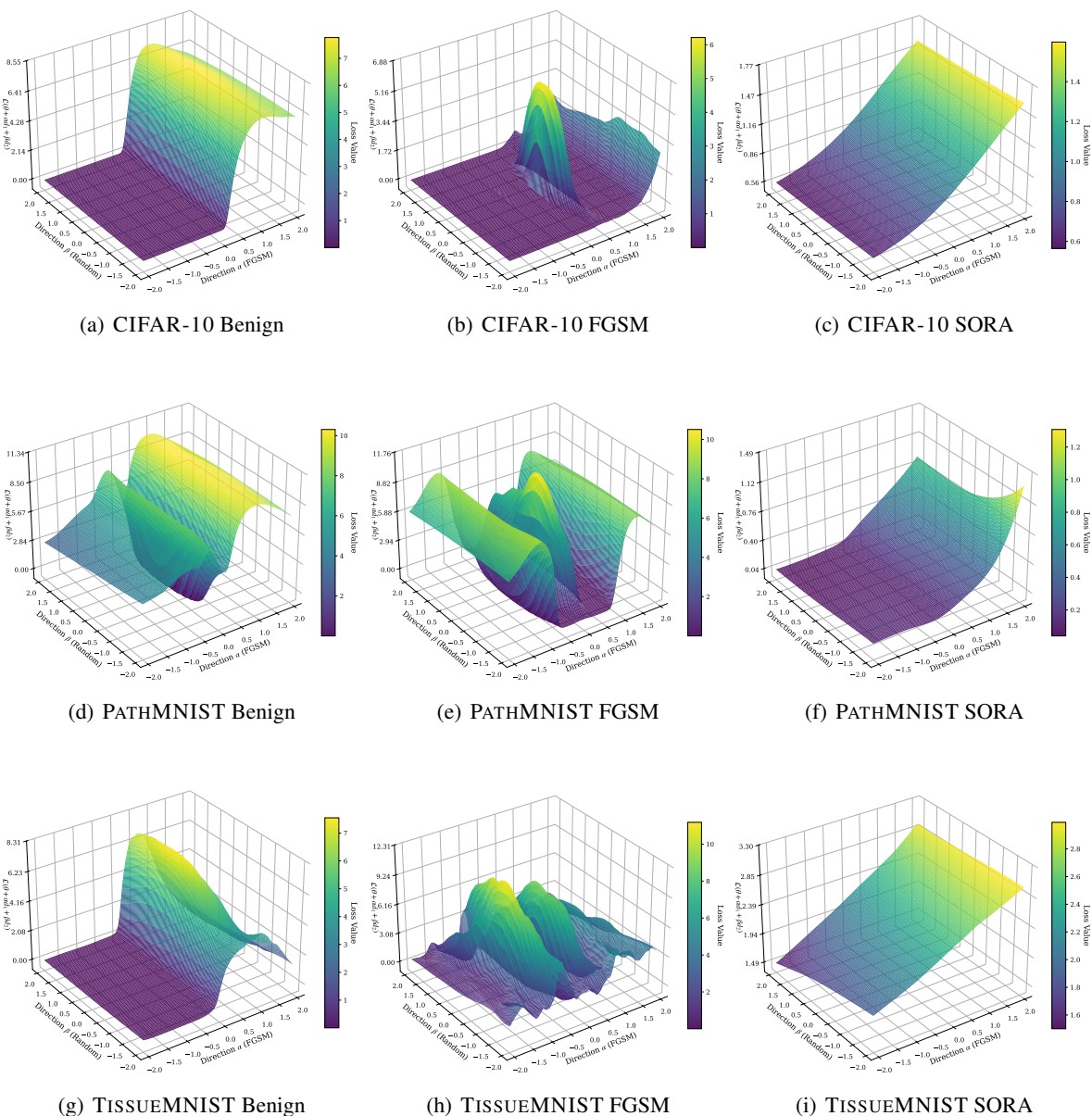

(a) CIFAR-10 Benign     (b) CIFAR-10 FGSM     (c) CIFAR-10 SORA

(d) PATHMNIST Benign     (e) PATHMNIST FGSM     (f) PATHMNIST SORA

(g) TISSUEMNIST Benign     (h) TISSUEMNIST FGSM     (i) TISSUEMNIST SORA

*Figure 13.* Loss landscape visualization for different datasets and training methods, trained using PreActResNet-18.

As shown in Figures 13(a), 13(d), and 13(g), the distortion of the loss around the origin ($\alpha = 0$ and $\beta = 0$), which arises naturally from standard training, does not exhibit a consistent or distinctive pattern. In contrast, the distortion caused by CO shows clear similarities across different datasets (Figures 13(b), 13(e), and 13(h)). In all cases, we observe a bump in the loss surface near the origin, which then flattens as we continue moving in the FGSM direction, demonstrating EO.

This characteristic distortion motivates the use of a second-order approximation to measure the distortion more efficiently (Section 4) and to find stronger adversarial examples (Section 5).

## H. Baseline Brittleness and Hyperparameter Sensitivity

We strongly believe that an effective FAT method should satisfy the qualities mentioned in Section 1. This means that a good solution to CO must have dataset and architecture agnostic hyperparameters in the sense that keeping the same hyperparameters across all settings performs reasonably well. If a solution requires multiple runs to find hyperparameters that work, one can no longer consider it fast. In fact, instead of training a single-step method for a number of different times to find decent hyperparameters, one could use multi-step methods to avoid the challenges of single-step adversarial training altogether. Therefore we insist that effective solutions must retain their performance on a fixed set of hyperparameters across a wide range of datasets and architectures to be classified as **fast**.

Several previous work however, have overlooked this principle. Nevertheless our method can still outperform these baselines **with a fixed set of hyperparameters** even if the baselines are allowed to **perform an exhaustive grid search of hyperparameters**. In this Appendix we illustrate the shortcomings and failures of several baselines in the extreme case of them being allowed to perform a grid search of hyperparameters.

For example Tables 13 and 14 illustrate how baselines like ZeroGrad and GradAlign fail across all hyperparameters on PATHMNIST, indicating limitations much deeper than a lack of proper hyperparameters. Specifically, in the case of ZeroGrad where $q_{\text{value}} \in [0, 1]$ we can infer that no viable hyperparameter can result in a robust model. Note that the highlighted row in each table is the default hyperparameter reported by the corresponding authors.

*Table 13.* GradAlign sweep: Effect of $\lambda$ on PATHMNIST performance

| $\lambda$ | Clean (%) | PGD-10 (%) |
|-----------|-----------|------------|
| 0.2 | 45.71 | 0.78 |
| 0.4 | 11.39 | 0.19 |
| 0.6 | 9.87 | 0.54 |
| 0.8 | 73.26 | 8.65 |
| 1.0 | 35.58 | 1.81 |

*Table 14.* ZeroGrad sweep: effect of $q_{\text{val}}$ on PATHMNIST performance

| $q_{\text{val}}$ | Clean (%) | PGD-10 (%) |
|------------------|-----------|------------|
| 0.35 | 49.47 | 0.86 |
| 0.45 | 67.22 | 5.69 |
| 0.60 | 78.69 | 8.18 |
| 0.70 | 88.13 | 10.43 |
| 0.90 | 88.32 | 16.42 |

Similarly AAER also fails to create a robust model even if we extend the hyperparameter search over multiple values as can be seen from Table 15. In many cases, the models simply manage to identify the class with the highest frequency in the training set (and by extension, the test set) and simply return that class in all cases exhibiting a form of shortcut learning. In other cases, AAER still fails to produce competitive results. The high number of hyperparameters further restricts the possibility of arriving at a reasonable solution.

Much like AAER, N-FGSM fails to avoid CO in many cases as well as depicted in Table 16. In addition to the fact that N-FGSM also doesn't produce competitive results, in the cases that it does manage to perform relatively better, we see a sharp decline over the original CIFAR-10 dataset. This means that not only N-FGSM fails to be competitive, it only manages to avoid CO at the cost of significant deterioration of other datasets. Unlike SORA, where the tune-able hyperparameters are stable across different settings, N-FGSM's hyperparameters which are its noise magnitude $k$ and step-size $\alpha$ don't generalize as well.

Finally, in the case of ELLE, although the resulting robust accuracies are competitive with respect to SORA, the competitive regions in terms of CIFAR-10 and PATHMNIST are mutually exclusive. From Table 17 it can be seen that the best results on CIFAR-10 are obtained on $\lambda = 1000$ and as we increase the strength of the regularizer $\lambda$ by increasing its value, both the clean and robust accuracies drop on this dataset. On the PATHMNIST dataset we observe a different pattern where by increasing $\lambda$ we can find the best models at $\lambda = 4000$. This discrepancy in terms of where we see the best results, coupled

*Table 15.* AAER sweep: Effect of $\lambda_1$, $\lambda_2$, and $\lambda_3$ on PATHMNIST performance

| $\lambda_1$ | $\lambda_2$ | $\lambda_3$ | Clean (%) | PGD-10 (%) |
|---|---|---|---|---|
| 1 | 1.5 | 0.15 | 74.86 | 18.84 |
| 1 | 5 | 0.55 | 82.26 | 32.58 |
| 1 | 8.5 | 0.55 | 82.26 | 32.58 |
| 1 | 2.75 | 0.55 | 82.26 | 32.58 |
| 1 | 5 | 1.5 | 18.64 | 18.64 |
| 1 | 8.5 | 1.5 | 18.64 | 18.64 |
| 1 | 2.75 | 1.5 | 18.64 | 18.64 |
| 1 | 5 | 0.75 | 18.64 | 18.64 |
| 1 | 8.5 | 0.75 | 18.64 | 18.64 |
| 1 | 2.75 | 0.75 | 18.64 | 18.64 |

*Table 16.* NFGSM sweep: Effect of $\alpha$ and $k$ on PATHMNIST and CIFAR-10 performance

| $k$ | $\alpha$ | Dataset | Clean (%) | PGD-10 (%) |
|---|---|---|---|---|
| $\epsilon$ | $\epsilon$ | PATHMNIST | 86.04 | 0.82 |
| $3\epsilon$ | $\epsilon$ | PATHMNIST | 62.71 | 20.59 |
| $3\epsilon$ | $2\epsilon$ | PATHMNIST | 66.78 | 4.45 |
| $2\epsilon$ | $2\epsilon$ | PATHMNIST | 35.94 | 1.15 |
| $2\epsilon$ | $\epsilon$ | CIFAR-10 | 79.05 | 48.73 |
| $2\epsilon$ | $\epsilon$ | PATHMNIST | 74.86 | 18.84 |
| $2\epsilon$ | $0.75\epsilon$ | PATHMNIST | 59.42 | 27.61 |
| $2\epsilon$ | $0.5\epsilon$ | PATHMNIST | 85.16 | 37.08 |
| $2\epsilon$ | $0.5\epsilon$ | CIFAR-10 | 86.57 | 39.84 |

*Table 17.* ELLE sweep: Effect of regularization strength ($\lambda$) on PATHMNIST and CIFAR-10

| $\lambda$ | Dataset | Clean (%) | PGD-10 (%) |
|---|---|---|---|
| 1000 | PATHMNIST | 79.80 | 40.52 |
| | CIFAR-10 | 83.74 | 45.10 |
| 4000 | PATHMNIST | 83.52 | 43.08 |
| | CIFAR-10 | 81.12 | 44.27 |
| 8000 | PATHMNIST | 79.55 | 42.77 |
| | CIFAR-10 | 79.10 | 42.63 |
| 12000 | PATHMNIST | 78.82 | 42.87 |
| | CIFAR-10 | 76.55 | 41.35 |
| 16000 | PATHMNIST | 77.70 | 44.21 |
| | CIFAR-10 | 74.90 | 40.70 |
| 20000 | PATHMNIST | 70.04 | 38.82 |
| | CIFAR-10 | 73.52 | 39.52 |

with the fact that ELLE itself is computationally expensive, further discourages its usage in favor of methods that are faster and attain higher accuracies such as SORA, or multi-step methods such as standard adversarial training if computational resources are not limited. For example, ELLE takes twice longer than SORA to train and consumes 3 times the memory (see Figures 7 and 16).

# I. Euclidean Norm Attacks

In Appendix A we investigated our proposed PertAlign metric to develop theoretical justifications for extracting the optimal step-sizes. Our theoretical analysis made no assumptions on the direction of the optimization step $v$, allowing for a wide variety of choices. Since $\ell_\infty$-norm perturbations are a great area of interest in adversarial robustness (Goodfellow et al., 2015), we dedicated Algorithm 1 to this norm, with the optimal step-size in the direction of the sign of the gradient $v = \text{sign}(g)$ derived in Proposition A.7. Another natural choice however, would be the the direction of the steepest ascent (i.e. gradient of the input loss $v = g$). Unlike many previous works such as ZeroGrad and MultiGrad (Golgooni et al., 2023) where the proposed method only works for $\ell_\infty$-norm perturbations, our method could be adjusted to work with any direction from the general class of $\ell_p$-norm perturbations.

While our theoretical results allow for any direction or any $\ell_p$-norm, in this appendix, we focus on the specific case of $v = g$ and show how the theory allows us to modify our method to the case of $\ell_2$-norm perturbations. The $\ell_2$ version of our method is provided in Algorithm 2. The theoretical derivation of the optimal step-size for $\ell_2$ is presented in Proposition A.6 which are depicted in lines 10 and 11. Also following previous works which employ random starts for $\ell_2$-norms, we use Gaussian noise instead of uniform noise (Madry et al., 2018). These changes are depicted in lines 3 and 4.

---

**Algorithm 2** **S**econd-**Or**der **A**daptive Method for $\ell_2$ (SORA)

---

**Require:** # epochs $T$, # batches $M$, radius $\epsilon$, step-size $\alpha$, exponential average coefficient $\beta$.

1:   $v \leftarrow 0.99$                                                                    {Initialize moving linearity coefficient}

2:   **for** Epoch $t = 1, \ldots, T$ **do**

3:       **for** Batch $i = 1, \ldots, M$ **do**

4:           $\boldsymbol{\eta} \sim \mathcal{N}(\boldsymbol{0}, I_d)$                                                      {Random start}

5:           $\boldsymbol{\eta} = \frac{\epsilon}{\|\boldsymbol{\eta}\|_2} \boldsymbol{\eta}$                                                 {Noise rescaling}

6:           $\boldsymbol{g} \leftarrow \nabla_{\boldsymbol{x}_i} \mathcal{L}(f_{\boldsymbol{\theta}}(\boldsymbol{x}_i + \boldsymbol{\eta}), y_i)$

7:           $\boldsymbol{\alpha}_i \sim \mathcal{U}(0, \alpha^*)^d$                                        {Element-wise step sampling}

8:           $\boldsymbol{x}_i' \leftarrow \boldsymbol{x}_i + \boldsymbol{\eta} + \boldsymbol{\alpha}_i \odot \boldsymbol{g}$

9:           $\boldsymbol{x}_i' = \Pi_{[0,1]}(\boldsymbol{x}_i')$                                     {Project onto the valid pixel range}

10:         $\boldsymbol{g}', \boldsymbol{g_\theta} \leftarrow \nabla_{[\boldsymbol{x}_i', \theta]} \mathcal{L}(f_{\boldsymbol{\theta}}(\boldsymbol{x}_i'), y_i)$                        {Backpropagation}

11:         $\boldsymbol{\theta} \leftarrow \text{optimizer}(\boldsymbol{\theta}, \boldsymbol{g_\theta})$            {Standard parameters update, (e.g. SGD)}

12:       Calculate optimal step-size for next batch:

$$\alpha^* = \begin{cases} \min\left(\alpha_{\max}, \dfrac{\alpha_0}{1-v}\right), & v < 1, \\ \alpha_{\max}, & \text{otherwise.} \end{cases}$$

13:       $v \leftarrow (1-\beta) \cdot v + \beta \cdot \dfrac{\|g'\|}{\|g\|} \cdot \text{PertAlign}$                      {Update moving linearity coefficient}

---

Table 18 shows our results on the $\ell_2$ variant of our method, trained for 30 epochs and with $\epsilon = 0.5$.

*Table 18.* SORA $\ell_2$ performance on CIFAR-10 and CIFAR-100

| Dataset | Architecture | Clean (%) | PGD-10 (%) |
|---------|--------------|-----------|------------|
| CIFAR-10 | PreActResNet-18 | 81.94 | 53.72 |
| | ResNet-18 | 81.50 | 53.06 |
| | SENet-18 | 82.53 | 53.52 |
| CIFAR-100 | PreActResNet-18 | 56.75 | 30.19 |
| | ResNet-18 | 56.62 | 30.18 |
| | SENet-18 | 56.12 | 29.86 |

## J. Reliability of PertAlign

To demonstrate the reliability of PertAlign, we first compare how quickly each metric signals the occurrence of CO in Section J.1. We then trace PertAlign during the training of various single-step methods across different datasets and architectures in Section J.2 to establish its applicability.

### J.1. Predictability

We tracked PertAlign, GradAlign, the AAE rate, scaled ELLE, and the scaled KL divergence (used by TRADES) during SORA (Figure 14(a)) and FGSM (Figure 14(b)) training.

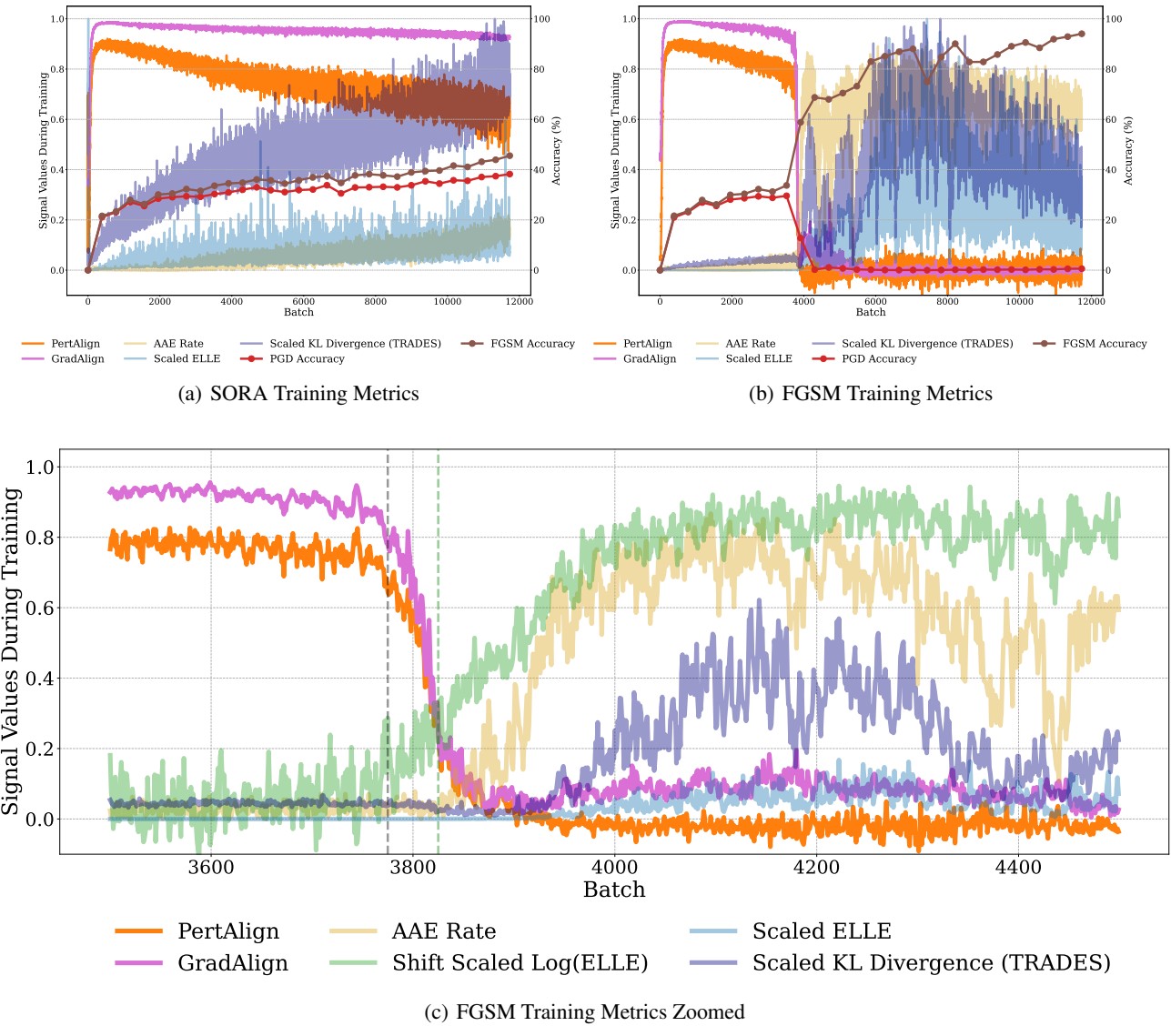

(a) SORA Training Metrics

(b) FGSM Training Metrics

(c) FGSM Training Metrics Zoomed

*Figure 14.* Tracking different metrics during FGSM and SORA AT.

During robust training where CO does not occur (Figure 14(a)), we observe that the metrics behave smoothly. Conversely, in FGSM training where CO does occur (Figure 14(b)), PertAlign and GradAlign suddenly drop to zero, while the number of Abnormal Adversarial Examples (AAEs), ELLE, and the KL divergence between the model's output distributions on clean and adversarial examples rise.

Figure 14(c) details the onset of CO from Figure 14(b), illustrating how each metric transitions as CO occurs. Notably,

PertAlign and GradAlign begin reacting at around batch 3775, whereas FGSM and PGD accuracies diverge visibly at around batch 3825, thus effectively predicting CO. We can see that ELLE, which measures the convexity of the loss landscape in the direction of the perturbation using sampling, also starts increasing at the same time. However, because ELLE does not have a bounded value range, we must manually shift and scale the logarithm of the ELLE metric to clearly observe its transition and it is important to note that ELLE achieves its peak value much later (Figure 14(b)).

It is important to note that while GradAlign and ELLE also react before the occurrence of CO, they require additional forward or backward passes. This significantly increases their computational overhead, whereas PertAlign is essentially free to compute. On the other hand, the number of AAEs and the KL divergence begin to react later, only after the FGSM and PGD accuracies have already diverged.

## J.2. Generalizability

To demonstrate the predictive power of the PertAlign metric, we present training traces from various architectures, methods, and datasets. The correlation between Catastrophic Overfitting (CO) and concurrent drops in PGD robustness and PertAlign is clearly evident. For instance, Figure 15(g) shows PertAlign dropping, recovering, and dropping again as the model undergoes CO, temporarily recovers, and then succumbs to CO once more.

More formally, the relationship between Catastrophic Overfitting and PertAlign can be explained by considering the loss landscape. During normal adversarial training, moving along the gradient ascent direction should increase the loss. However, during CO, we observe a decline in loss values after a certain perturbation threshold, indicating that the loss begins to decrease beyond this point. This behavior *requires* high curvature in the gradient direction; otherwise, the loss would increase monotonically along the gradient path.

From an optimization perspective, multi-step gradient ascent with appropriate step-sizes could mitigate this issue by carefully navigating the loss landscape. However, in single-step adversarial attacks, as in fast adversarial training, the step-size becomes the critical parameter. SORA addresses this by using the PertAlign metric to estimate the local curvature and adaptively adjust the step-size, avoiding overshooting that leads to CO.

Overshooting along the gradient direction is problematic not only because it generates weaker or even abnormal adversarial examples, but also because training on these examples can distort the loss surface further, as illustrated in Figure 4. This distortion may reduce margins, defined as the distance from inputs to the decision boundary, contrary to the goal of improving robust accuracy (Ding et al., 2020; Sriramanan et al., 2020).

Methods like that of Li et al. (2021) rely on PGD accuracy to change the attack in AT. As the PGD accuracy drops, we can conclude that CO has occurred. This in turn suggests that by going back to using the PGD attack we can make our method robust once more. While the iterations where PGD is the attack take longer, the most time consuming aspect of this method is the evaluation step using PGD. Since PertAlign can reliably predict CO without incurring additional costs, the approach by Li et al. (2021) can be implemented much more efficiently.

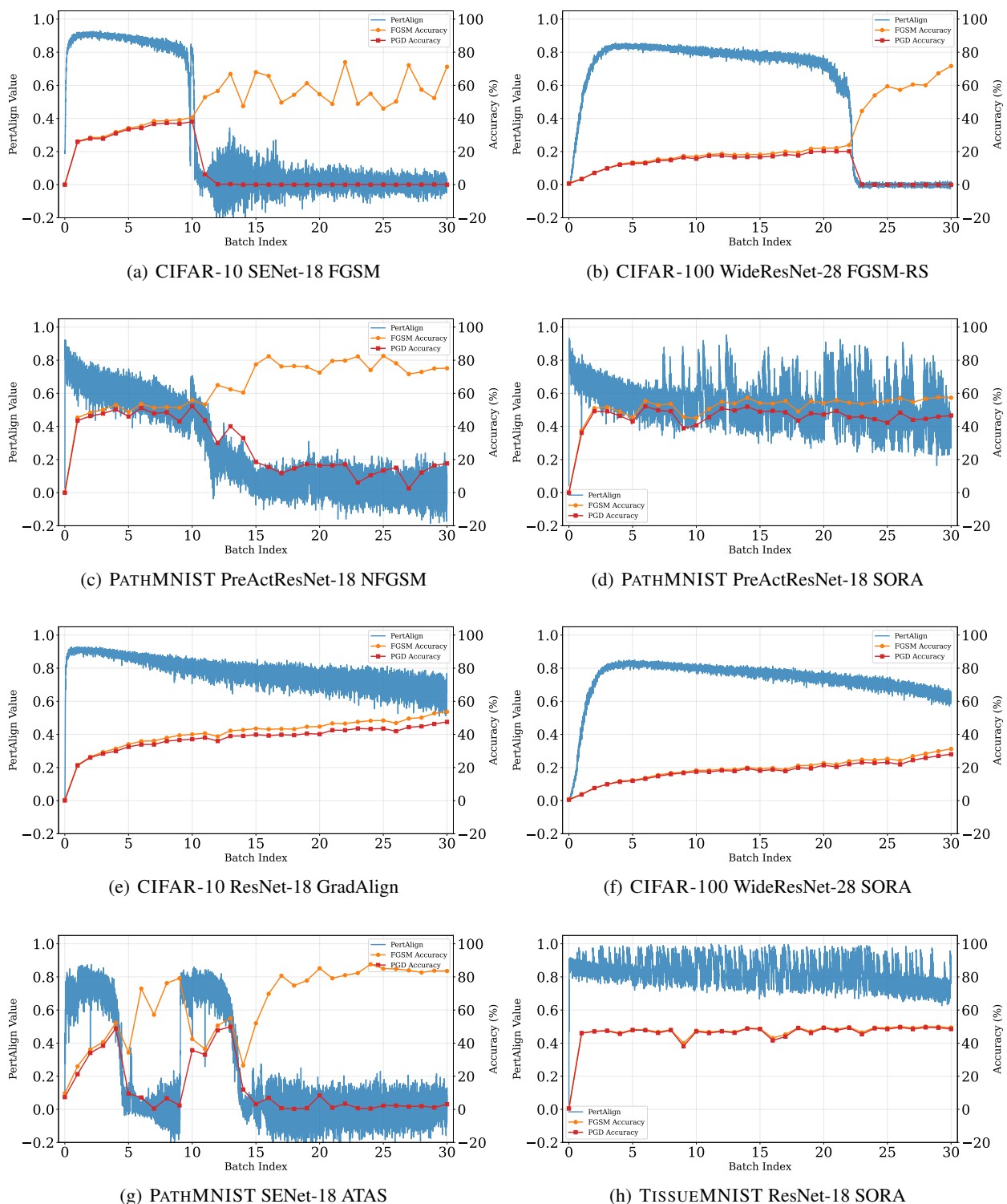

*Figure 15.* Tracking PertAlign across datasets, models, and FAT methods.

# K. Extended Results

We provide detailed results of our experiments on methods, datasets and architectures from Appendices B, D and E in Appendix K.3. The key results have summarized in Appendix K.1. The computational costs of our method and baselines have been discussed in Appendix K.2.

## K.1. Summary of Results

Figure 6 presents an overview of these results. Each column in Figure 6 is presented as a table in Appendix K.3. The relative accuracy reported in Figure 6 is obtained via dividing accuracies for each setting by the highest accuracy in that setting. This allows us to analyze the results across a wide range of settings in a single figure.

Our SORA, manages to attain excellent performance across all dataset–architecture pairs whereas other baselines fail to be competitive on some datasets. Baselines such as FGSM and FGSM-RS suffer from CO in most settings. Other baselines such as GradAlign, ZeroGrad, and ATAS also suffer from CO, especially on previously unseen datasets, albeit less frequently. Methods such as N-FGSM and AAER which are competitive on CIFAR-10/100 and well known datasets, struggle on more challenging datasets such as PATHMNIST. Computationally expensive methods like MultiGrad and ELLE fair better in terms of generalization, however they also fail to match the accuracy of our SORA. Additionally, some of the more computationally demanding methods like ATAS and ELLE face more serious problems which are discussed in Appendix K.2.

Therefore, we conclude that our SORA is the **only** method which can provide optimal performance across all settings, even though we impose additional constrains on it in terms of hyperparameters (see Appendix M.3). This makes SORA the only method that consistently achieves high robust accuracies, at a low cost and across datasets.

Moreover, as can be seen from Table 1, SORA not only outperforms the efficient baselines (see Appendix K.2) in terms of robust accuracy, but in terms of clean accuracy as well. This demonstrates that even on larger datasets such as IMAGENET-100 our method generalizes better than other methods despite the fact we don't tune our hyperparameters on this dataset (see Appendix M.3).

## K.2. Computational Costs

We have summarized some of the computational costs associated with different methods in Table 19. This table is concerned with the number of forward and backward passes required in each method. Other contributing factors, such as batch size and computational graphs have been overlooked. However, Figures 7 and 16 report the time and memory complexity of different methods empirically.

ATAS exhibits significantly higher memory consumption compared to other methods, as illustrated by its resource usage on CIFAR-10 with PreActResNet-18 in Figure 16(b). This substantial memory requirement prevented us from running ATAS on TINYIMAGENET using our NVIDIA GeForce RTX 4090 GPU due to hardware limitations.

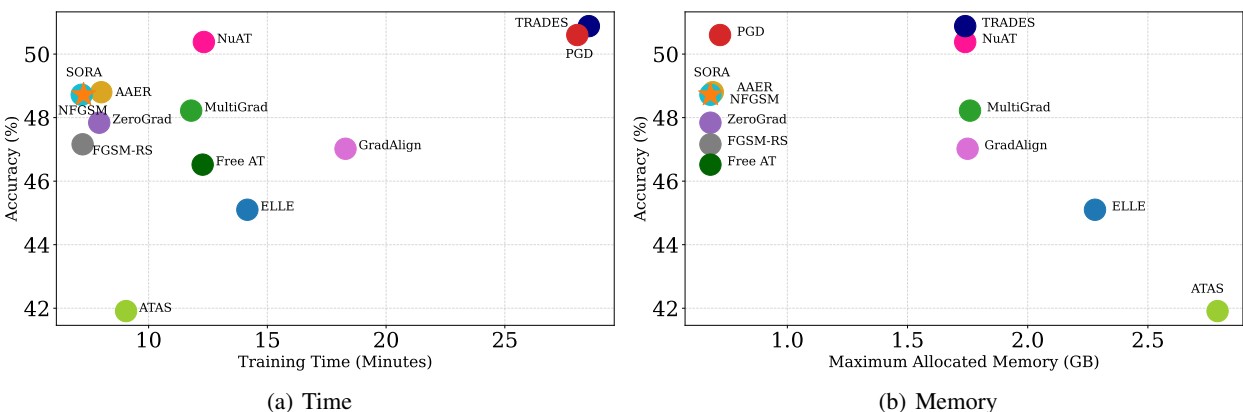

(a) Time          (b) Memory

*Figure 16.* Training time vs. memory usage on CIFAR-10 with PreActResNet-18, trained for 30 epochs, measured on an NVIDIA GeForce RTX 4090 GPU. The ★ marks SORA.

*Table 19.* Computational cost comparison of AT methods. Forward/backward passes are reported per batch per training epoch. $\mathcal{O}$(batch) and $\mathcal{O}$(dataset) denote linear scaling with batch size and dataset size respectively. PGD assumes a stateless implementation (standard practice). Free AT trains on fewer epochs to reduce the effects of the higher per-epoch costs.

| Method | Type | # Forward Passes | # Backward Passes | Memory Scaling |
|---|---|---|---|---|
| Benign | Standard | 1 | 1 | $\mathcal{O}$(batch) |
| FGSM | Single-Step | 2 | 2 | $\mathcal{O}$(batch) |
| Free AT | Single-Step (Replay) | $m$ | $m$ | $\mathcal{O}$(batch) |
| FGSM-RS | Single-Step | 2 | 2 | $\mathcal{O}$(batch) |
| GradAlign | Single-Step + Reg. | 3 | 2 | $\approx \mathcal{O}(3 \times \text{batch})$ |
| NuAT | Single-Step + Reg. | 4 | 2 | $\approx \mathcal{O}(3 \times \text{batch})$ |
| ZeroGrad | Single-Step | 2 | 2 | $\mathcal{O}$(batch) |
| MultiGrad | Single-Step (Ensemble) | 2 | 2 | $\mathcal{O}(N \times \text{batch})$ |
| N-FGSM | Single-Step | 2 | 2 | $\mathcal{O}$(batch) |
| ATAS | Single-Step (Adaptive) | 2 | 2 | $\mathcal{O}(\text{batch} + \text{dataset})$ |
| AAER | Single-Step + Reg. | 2 | 2 | $\mathcal{O}$(batch) |
| ELLE | Single-Step + Reg. (Ensemble) | 3 | 2 | $\mathcal{O}(3 \times \text{batch})$ |
| SORA | Single-Step (Adaptive) | 2 | 2 | $\mathcal{O}$(batch) |
| PGD-2 | Multi-Step | 3 | 3 | $\mathcal{O}$(batch) |
| PGD-10 | Multi-Step | 11 | 11 | $\mathcal{O}$(batch) |
| TRADES | Multi-Step + Reg. | 12 | 11 | $\approx \mathcal{O}(3 \times \text{batch})$ |

Rocamora et al. (2024) recommend $\lambda$ values in the range $[4000, 20000]$ to avoid Catastrophic Overfitting (CO) and achieve optimal performance. However, we found that for WideResNet architectures, this parameter range, combined with the typically large values of their regularizer (e.g., $10^{20}$), frequently leads to numerical overflow. This instability causes the loss regularizer term to become `NaN`, ultimately disrupting the training process.

### K.3. Comprehensive Results

In this section, we report the experimental results for every dataset–architecture pair used in our study.

*Table 20.* Results for CIFAR-10 on the PreActResNet-18 architecture.

| Method | Clean | FGSM | PGD-10 | AA |
|---|---|---|---|---|
| Benign | 92.94 ± 0.33 | 6.72 ± 0.43 | 0.00 ± 0.00 | 0.00 ± 0.00 |
| FGSM | 83.80 ± 0.18 | 69.65 ± 12.57 | 0.01 ± 0.02 | 0.00 ± 0.00 |
| Free AT | 77.53 ± 0.09 | 50.34 ± 0.12 | 46.52 ± 0.09 | 42.23 ± 0.13 |
| FGSM-RS | 82.61 ± 0.03 | 53.32 ± 0.36 | 47.16 ± 0.35 | 42.48 ± 0.23 |
| GradAlign | 82.40 ± 0.17 | 53.11 ± 0.35 | 47.02 ± 0.48 | 42.49 ± 0.39 |
| NuAT | 75.09 ± 0.19 | 53.10 ± 0.17 | 50.28 ± 0.21 | 45.90 ± 0.09 |
| ZeroGrad | 80.92 ± 0.04 | 53.52 ± 0.27 | 47.84 ± 0.38 | 43.05 ± 0.23 |
| MultiGrad | 80.64 ± 0.22 | 53.76 ± 0.24 | 48.22 ± 0.27 | 43.33 ± 0.25 |
| N-FGSM | 79.05 ± 0.21 | 53.32 ± 0.22 | 48.73 ± 0.44 | 43.95 ± 0.23 |
| ATAS | 87.25 ± 0.13 | 51.97 ± 0.13 | 41.91 ± 0.38 | 37.25 ± 0.40 |
| AAER | 79.38 ± 0.12 | 53.38 ± 0.40 | 48.59 ± 0.28 | 43.96 ± 0.26 |
| ELLE | 83.74 ± 0.51 | 52.59 ± 0.18 | 45.10 ± 0.35 | 40.63 ± 0.25 |
| SORA (Ours) | 79.90 ± 0.05 | 53.67 ± 0.31 | 48.64 ± 0.44 | 43.83 ± 0.31 |
| PGD-2 | 83.78 ± 0.10 | 53.33 ± 0.38 | 46.86 ± 0.42 | 42.30 ± 0.32 |
| PGD-10 | 79.37 ± 0.12 | 54.38 ± 0.13 | 50.60 ± 0.10 | 46.15 ± 0.19 |
| TRADES | 78.45 ± 0.42 | 54.16 ± 0.13 | 50.88 ± 0.10 | 46.50 ± 0.10 |

Table 21. Results for CIFAR-10 on the ResNet-18 architecture.

| Method | Clean | FGSM | PGD-10 | AA |
|---|---|---|---|---|
| Benign | 92.99 ± 0.08 | 8.65 ± 0.51 | 0.00 ± 0.00 | 0.00 ± 0.00 |
| FGSM | 85.21 ± 0.49 | 85.99 ± 4.21 | 0.06 ± 0.06 | 0.00 ± 0.00 |
| Free AT | 77.61 ± 0.36 | 49.97 ± 0.24 | 46.34 ± 0.53 | 42.10 ± 0.39 |
| FGSM-RS | 78.90 ± 5.28 | 66.18 ± 13.80 | 15.95 ± 21.98 | 14.15 ± 20.01 |
| GradAlign | 82.40 ± 0.22 | 53.70 ± 0.15 | 47.29 ± 0.21 | 42.49 ± 0.05 |
| NuAT | 84.32 ± 6.57 | 53.88 ± 1.26 | 35.53 ± 10.16 | 18.60 ± 19.21 |
| ZeroGrad | 80.74 ± 0.15 | 54.11 ± 0.12 | 47.94 ± 0.54 | 42.61 ± 0.94 |
| MultiGrad | 80.66 ± 0.14 | 53.99 ± 0.05 | 48.45 ± 0.29 | 43.33 ± 0.14 |
| N-FGSM | 78.91 ± 0.30 | 53.52 ± 0.30 | 48.76 ± 0.43 | 43.96 ± 0.31 |
| ATAS | 87.23 ± 0.24 | 52.44 ± 0.22 | 42.43 ± 0.18 | 38.06 ± 0.31 |
| AAER | 33.03 ± 32.56 | 24.71 ± 20.80 | 23.13 ± 18.57 | 43.96 ± 0.32 |
| ELLE | 83.30 ± 0.13 | 52.21 ± 0.23 | 45.02 ± 0.30 | 40.33 ± 0.13 |
| SORA (Ours) | 79.68 ± 0.20 | 54.02 ± 0.14 | 48.96 ± 0.35 | 43.97 ± 0.29 |
| PGD-2 | 83.78 ± 0.11 | 53.48 ± 0.40 | 46.70 ± 0.36 | 42.24 ± 0.40 |
| PGD-10 | 79.17 ± 0.03 | 54.62 ± 0.14 | 50.99 ± 0.23 | 46.46 ± 0.17 |
| TRADES | 78.19 ± 0.16 | 53.89 ± 0.09 | 50.66 ± 0.11 | 46.42 ± 0.16 |

Table 22. Results for CIFAR-10 on the WideResNet-28 architecture.

| Method | Clean | FGSM | PGD-10 | AA |
|---|---|---|---|---|
| Benign | 93.99 ± 0.16 | 10.13 ± 0.72 | 0.00 ± 0.00 | 0.00 ± 0.00 |
| FGSM | 86.07 ± 0.55 | 78.82 ± 2.65 | 0.09 ± 0.03 | 0.00 ± 0.00 |
| Free AT | 80.62 ± 0.37 | 52.52 ± 0.16 | 48.30 ± 0.12 | 44.28 ± 0.03 |
| FGSM-RS | 84.12 ± 2.40 | 92.10 ± 3.47 | 0.06 ± 0.07 | 0.00 ± 0.00 |
| GradAlign | 85.72 ± 0.42 | 55.42 ± 0.25 | 47.92 ± 0.10 | 43.63 ± 0.34 |
| NuAT | 86.53 ± 1.91 | 56.08 ± 5.33 | 29.37 ± 2.82 | 7.13 ± 3.58 |
| ZeroGrad | 81.66 ± 4.07 | 87.39 ± 6.19 | 0.01 ± 0.02 | 0.00 ± 0.00 |
| MultiGrad | 84.92 ± 0.30 | 55.98 ± 0.09 | 48.60 ± 0.21 | 43.85 ± 0.25 |
| N-FGSM | 82.54 ± 0.35 | 55.26 ± 0.13 | 49.48 ± 0.08 | 45.26 ± 0.19 |
| ATAS | 89.07 ± 0.12 | 54.58 ± 0.43 | 44.11 ± 0.45 | 39.52 ± 0.34 |
| AAER | 82.67 ± 0.54 | 55.24 ± 0.10 | 49.42 ± 0.15 | 45.27 ± 0.19 |
| ELLE | 54.71 ± 32.75 | 41.62 ± 22.43 | 18.61 ± 19.53 | 16.89 ± 17.31 |
| SORA (Ours) | 83.44 ± 0.27 | 55.74 ± 0.19 | 49.46 ± 0.07 | 44.96 ± 0.33 |
| PGD-2 | 87.27 ± 0.15 | 55.20 ± 0.21 | 46.97 ± 0.24 | 42.69 ± 0.41 |
| PGD-10 | 83.34 ± 0.67 | 57.04 ± 0.46 | 51.89 ± 0.56 | 47.17 ± 0.61 |
| TRADES | 80.43 ± 0.44 | 55.80 ± 0.03 | 52.14 ± 0.30 | 47.77 ± 0.16 |

Table 23. Results for CIFAR-10 on the SENet-18 architecture.

| Method | Clean | FGSM | PGD-10 | AA |
|---|---|---|---|---|
| Benign | 93.03 ± 0.09 | 8.90 ± 0.52 | 0.00 ± 0.00 | 0.00 ± 0.00 |
| FGSM | 84.13 ± 1.12 | 80.13 ± 19.39 | 0.07 ± 0.06 | 0.00 ± 0.00 |
| Free AT | 77.43 ± 0.09 | 50.10 ± 0.43 | 46.32 ± 0.68 | 42.09 ± 0.41 |
| FGSM-RS | 77.91 ± 6.72 | 59.19 ± 8.19 | 31.00 ± 21.87 | 27.49 ± 19.44 |
| GradAlign | 82.75 ± 0.39 | 53.75 ± 0.09 | 47.26 ± 0.24 | 42.41 ± 0.21 |
| NuAT | 86.39 ± 2.56 | 56.69 ± 5.08 | 30.74 ± 5.14 | 10.53 ± 7.74 |
| ZeroGrad | 81.83 ± 3.59 | 75.40 ± 17.00 | 15.96 ± 22.57 | 14.32 ± 20.26 |
| MultiGrad | 81.30 ± 0.51 | 54.20 ± 0.38 | 48.62 ± 0.19 | 43.61 ± 0.25 |
| N-FGSM | 79.44 ± 0.39 | 53.63 ± 0.22 | 48.91 ± 0.12 | 43.93 ± 0.23 |
| ATAS | 87.38 ± 0.07 | 52.37 ± 0.39 | 41.80 ± 0.15 | 37.25 ± 0.16 |
| AAER | 79.42 ± 0.34 | 53.49 ± 0.30 | 48.81 ± 0.10 | 43.93 ± 0.22 |
| ELLE | 83.68 ± 0.28 | 52.64 ± 0.13 | 45.44 ± 0.19 | 40.86 ± 0.21 |
| SORA (Ours) | 80.17 ± 0.46 | 53.80 ± 0.10 | 48.95 ± 0.08 | 43.82 ± 0.27 |
| PGD-2 | 84.28 ± 0.31 | 53.85 ± 0.05 | 47.28 ± 0.12 | 42.72 ± 0.16 |
| PGD-10 | 79.62 ± 0.09 | 54.93 ± 0.13 | 51.27 ± 0.18 | 46.42 ± 0.47 |
| TRADES | 78.66 ± 0.41 | 54.01 ± 0.10 | 50.90 ± 0.27 | 46.54 ± 0.34 |

Table 24. Results for CIFAR-100 on the PreActResNet-18 architecture.

| Method | Clean | FGSM | PGD-10 | AA |
|---|---|---|---|---|
| Benign | 73.89 ± 0.15 | 4.81 ± 0.53 | 0.01 ± 0.01 | 0.00 ± 0.00 |
| FGSM | 55.97 ± 2.25 | 63.93 ± 18.78 | 0.16 ± 0.13 | 0.00 ± 0.00 |
| Free AT | 50.28 ± 0.40 | 25.86 ± 0.49 | 23.79 ± 0.49 | 19.54 ± 0.31 |
| FGSM-RS | 52.70 ± 0.52 | 55.02 ± 2.45 | 0.00 ± 0.00 | 0.00 ± 0.00 |
| GradAlign | 57.50 ± 0.85 | 28.62 ± 0.11 | 25.20 ± 0.30 | 20.67 ± 0.18 |
| NuAT | 49.66 ± 0.41 | 29.23 ± 0.29 | 27.81 ± 0.33 | 22.13 ± 0.33 |
| ZeroGrad | 56.28 ± 0.47 | 28.37 ± 0.04 | 25.32 ± 0.13 | 20.81 ± 0.28 |
| MultiGrad | 55.53 ± 0.55 | 28.98 ± 0.27 | 26.09 ± 0.21 | 21.45 ± 0.05 |
| N-FGSM | 53.43 ± 0.40 | 28.82 ± 0.12 | 26.25 ± 0.18 | 21.75 ± 0.18 |
| ATAS | 63.57 ± 0.17 | 27.34 ± 0.30 | 21.10 ± 0.19 | 17.17 ± 0.16 |
| AAER | 53.72 ± 0.54 | 28.79 ± 0.20 | 26.22 ± 0.26 | 21.72 ± 0.17 |
| ELLE | 57.94 ± 0.47 | 27.74 ± 0.46 | 23.79 ± 0.28 | 19.70 ± 0.05 |
| SORA (Ours) | 54.74 ± 0.36 | 28.89 ± 0.21 | 26.12 ± 0.07 | 21.56 ± 0.24 |
| PGD-2 | 58.65 ± 0.68 | 28.60 ± 0.14 | 24.96 ± 0.19 | 20.72 ± 0.15 |
| PGD-10 | 53.57 ± 0.69 | 29.66 ± 0.11 | 27.64 ± 0.06 | 22.88 ± 0.06 |
| TRADES | 55.42 ± 0.58 | 29.86 ± 0.25 | 28.05 ± 0.23 | 22.77 ± 0.08 |

*Table 25.* Results for CIFAR-100 on the ResNet-18 architecture.

| Method | Clean | FGSM | PGD-10 | AA |
|---|---|---|---|---|
| Benign | 74.08 ± 0.15 | 5.72 ± 0.62 | 0.01 ± 0.01 | 0.00 ± 0.00 |
| FGSM | 53.51 ± 3.06 | 47.79 ± 14.19 | 0.01 ± 0.00 | 0.00 ± 0.00 |
| Free AT | 49.97 ± 0.22 | 25.81 ± 0.16 | 23.71 ± 0.06 | 19.47 ± 0.14 |
| FGSM-RS | 50.51 ± 5.11 | 66.70 ± 9.54 | 0.00 ± 0.00 | 0.00 ± 0.00 |
| GradAlign | 57.24 ± 0.41 | 28.85 ± 0.01 | 25.37 ± 0.13 | 20.97 ± 0.17 |
| NuAT | 49.28 ± 0.43 | 29.28 ± 0.35 | 27.84 ± 0.28 | 22.19 ± 0.08 |
| ZeroGrad | 56.22 ± 0.51 | 28.60 ± 0.17 | 25.49 ± 0.15 | 20.96 ± 0.14 |
| MultiGrad | 55.48 ± 0.49 | 29.17 ± 0.13 | 26.17 ± 0.03 | 21.61 ± 0.12 |
| N-FGSM | 53.26 ± 0.66 | 28.59 ± 0.10 | 26.21 ± 0.05 | 21.81 ± 0.06 |
| ATAS | 63.39 ± 0.16 | 27.71 ± 0.64 | 21.68 ± 0.31 | 17.50 ± 0.26 |
| AAER | 18.23 ± 24.37 | 10.16 ± 12.96 | 9.38 ± 11.86 | 14.90 ± 9.83 |
| ELLE | 57.53 ± 0.41 | 27.68 ± 0.25 | 23.94 ± 0.48 | 19.69 ± 0.39 |
| SORA (Ours) | 54.50 ± 0.55 | 29.09 ± 0.20 | 26.24 ± 0.15 | 21.70 ± 0.18 |
| PGD-2 | 58.41 ± 0.45 | 28.39 ± 0.31 | 25.01 ± 0.31 | 20.65 ± 0.29 |
| PGD-10 | 53.39 ± 0.40 | 29.68 ± 0.29 | 27.61 ± 0.41 | 22.93 ± 0.19 |
| TRADES | 55.00 ± 0.59 | 30.13 ± 0.19 | 28.16 ± 0.23 | 22.79 ± 0.07 |

*Table 26.* Results for CIFAR-100 on the WideResNet-28 architecture.

| Method | Clean | FGSM | PGD-10 | AA |
|---|---|---|---|---|
| Benign | 76.78 ± 0.26 | 7.24 ± 0.15 | 0.01 ± 0.01 | 0.00 ± 0.00 |
| FGSM | 57.83 ± 1.12 | 49.82 ± 6.14 | 0.02 ± 0.01 | 0.00 ± 0.00 |
| Free AT | 53.23 ± 0.40 | 27.55 ± 0.23 | 25.35 ± 0.29 | 20.99 ± 0.11 |
| FGSM-RS | 48.19 ± 1.36 | 70.83 ± 0.84 | 0.00 ± 0.00 | 0.00 ± 0.00 |
| GradAlign | 60.43 ± 1.46 | 41.21 ± 8.98 | 9.96 ± 12.31 | 7.74 ± 10.83 |
| NuAT | 20.99 ± 28.27 | 15.57 ± 20.60 | 7.71 ± 9.49 | 2.54 ± 2.17 |
| ZeroGrad | 50.57 ± 5.73 | 69.10 ± 4.99 | 1.07 ± 1.50 | 0.09 ± 0.13 |
| MultiGrad | 60.07 ± 0.26 | 31.02 ± 0.03 | 27.37 ± 0.08 | 23.08 ± 0.06 |
| N-FGSM | 57.28 ± 0.19 | 30.50 ± 0.34 | 27.82 ± 0.38 | 23.74 ± 0.29 |
| ATAS | 68.18 ± 0.88 | 40.85 ± 3.88 | 16.07 ± 2.36 | 7.21 ± 3.27 |
| AAER | 57.10 ± 0.42 | 30.54 ± 0.32 | 27.92 ± 0.31 | 23.75 ± 0.29 |
| ELLE | 2.17 ± 1.66 | 2.19 ± 1.69 | 0.67 ± 0.47 | 0.67 ± 0.47 |
| SORA (Ours) | 58.58 ± 0.30 | 31.23 ± 0.12 | 27.93 ± 0.18 | 23.77 ± 0.14 |
| PGD-2 | 62.96 ± 0.25 | 30.62 ± 0.15 | 26.23 ± 0.16 | 22.29 ± 0.06 |
| PGD-10 | 57.97 ± 0.24 | 32.26 ± 0.20 | 29.57 ± 0.32 | 25.20 ± 0.28 |
| TRADES | 57.76 ± 0.02 | 32.17 ± 0.31 | 30.15 ± 0.35 | 24.85 ± 0.24 |

*Table 27.* Results for CIFAR-100 on the SENet-18 architecture.

| Method | Clean | FGSM | PGD-10 | AA |
|--------|-------|------|--------|-----|
| Benign | 73.66 ± 0.42 | 6.46 ± 0.38 | 0.09 ± 0.03 | 0.00 ± 0.00 |
| FGSM | 53.47 ± 2.71 | 40.70 ± 26.03 | 0.14 ± 0.08 | 0.00 ± 0.00 |
| Free AT | 48.56 ± 0.18 | 25.53 ± 0.34 | 23.65 ± 0.24 | 19.30 ± 0.04 |
| FGSM-RS | 49.62 ± 9.38 | 44.65 ± 21.85 | 7.87 ± 11.11 | 5.71 ± 8.08 |
| GradAlign | 56.44 ± 0.51 | 28.34 ± 0.30 | 25.06 ± 0.14 | 20.71 ± 0.21 |
| NuAT | 47.58 ± 0.24 | 28.41 ± 0.43 | 26.96 ± 0.25 | 21.44 ± 0.17 |
| ZeroGrad | 55.24 ± 0.39 | 28.21 ± 0.34 | 25.11 ± 0.11 | 20.64 ± 0.05 |
| MultiGrad | 54.36 ± 0.24 | 28.79 ± 0.34 | 25.86 ± 0.22 | 21.35 ± 0.07 |
| N-FGSM | 52.30 ± 0.24 | 28.27 ± 0.39 | 25.85 ± 0.16 | 21.55 ± 0.02 |
| ATAS | 62.48 ± 0.23 | 27.12 ± 0.13 | 21.31 ± 0.21 | 17.70 ± 0.21 |
| AAER | 52.23 ± 0.16 | 28.50 ± 0.53 | 26.12 ± 0.38 | 21.59 ± 0.17 |
| ELLE | 57.04 ± 0.13 | 27.35 ± 0.15 | 23.70 ± 0.18 | 19.41 ± 0.06 |
| SORA (Ours) | 53.61 ± 0.29 | 28.95 ± 0.40 | 26.44 ± 0.24 | 21.76 ± 0.11 |
| PGD-2 | 57.69 ± 0.32 | 28.43 ± 0.36 | 24.84 ± 0.28 | 20.55 ± 0.14 |
| PGD-10 | 52.52 ± 0.39 | 29.11 ± 0.27 | 26.94 ± 0.13 | 22.40 ± 0.04 |
| TRADES | 54.11 ± 0.37 | 29.41 ± 0.13 | 27.49 ± 0.06 | 22.43 ± 0.08 |

*Table 28.* Results for TINYIMAGENET on the PreActResNet-18 architecture.

| Method | Clean | FGSM | PGD-10 | AA |
|--------|-------|------|--------|-----|
| Benign | 61.78 ± 0.23 | 0.56 ± 0.06 | 0.01 ± 0.00 | 0.00 ± 0.00 |
| FGSM | 36.61 ± 0.10 | 88.21 ± 3.54 | 0.01 ± 0.01 | 0.00 ± 0.00 |
| FGSM-RS | 47.46 ± 0.11 | 22.31 ± 0.11 | 19.67 ± 0.22 | 14.95 ± 0.39 |
| GradAlign | 47.61 ± 0.05 | 22.03 ± 0.32 | 19.46 ± 0.21 | 14.79 ± 0.21 |
| NuAT | 34.80 ± 3.91 | 24.72 ± 6.82 | 16.22 ± 4.09 | 10.32 ± 5.97 |
| ZeroGrad | 46.08 ± 0.13 | 21.65 ± 0.14 | 19.31 ± 0.07 | 14.86 ± 0.12 |
| MultiGrad | 45.42 ± 0.11 | 22.27 ± 0.15 | 19.98 ± 0.16 | 15.26 ± 0.07 |
| N-FGSM | 44.66 ± 0.20 | 22.14 ± 0.28 | 20.18 ± 0.18 | 15.77 ± 0.10 |
| AAER | 44.72 ± 0.25 | 22.02 ± 0.12 | 19.94 ± 0.25 | 15.71 ± 0.13 |
| ELLE | 48.77 ± 0.04 | 21.61 ± 0.09 | 18.70 ± 0.04 | 14.29 ± 0.32 |
| SORA (Ours) | 45.30 ± 0.19 | 22.58 ± 0.16 | 20.07 ± 0.22 | 15.38 ± 0.12 |
| PGD-2 | 48.64 ± 0.24 | 21.72 ± 0.24 | 19.09 ± 0.38 | 14.79 ± 0.13 |

*Table 29.* Results for TINYIMAGENET on the ResNet-18 architecture.

| Method | Clean | FGSM | PGD-10 | AA |
|--------|-------|------|--------|-----|
| Benign | 63.00 ± 0.56 | 1.05 ± 0.06 | 0.03 ± 0.02 | 0.00 ± 0.00 |
| FGSM | 39.10 ± 0.60 | 38.16 ± 14.90 | 5.75 ± 8.13 | 4.30 ± 6.08 |
| FGSM-RS | 45.92 ± 4.16 | 17.59 ± 7.37 | 0.05 ± 0.07 | 0.00 ± 0.00 |
| GradAlign | 48.19 ± 0.23 | 22.30 ± 0.12 | 19.47 ± 0.19 | 15.12 ± 0.09 |
| NuAT | 45.35 ± 3.38 | 27.09 ± 3.66 | 11.22 ± 1.08 | 2.39 ± 0.25 |
| ZeroGrad | 46.66 ± 0.13 | 22.17 ± 0.15 | 19.44 ± 0.13 | 15.09 ± 0.21 |
| MultiGrad | 46.24 ± 0.23 | 22.69 ± 0.15 | 20.26 ± 0.18 | 15.78 ± 0.08 |
| N-FGSM | 45.19 ± 0.08 | 22.36 ± 0.20 | 20.16 ± 0.29 | 15.68 ± 0.25 |
| AAER | 45.11 ± 0.39 | 22.45 ± 0.20 | 20.29 ± 0.18 | 15.85 ± 0.13 |
| ELLE | 49.43 ± 0.17 | 21.67 ± 0.25 | 18.56 ± 0.18 | 14.40 ± 0.08 |
| SORA (Ours) | 45.83 ± 0.05 | 22.86 ± 0.14 | 20.45 ± 0.08 | 15.72 ± 0.12 |
| PGD-2 | 49.40 ± 0.16 | 22.34 ± 0.36 | 19.43 ± 0.11 | 15.13 ± 0.13 |

*Table 30.* Results for TINYIMAGENET on the WideResNet-28 architecture.

| Method | Clean | FGSM | PGD-10 | AA |
|---|---|---|---|---|
| FGSM | 33.80 ± 5.73 | 64.24 ± 14.68 | 0.01 ± 0.01 | 0.00 ± 0.00 |
| FGSM-RS | 29.96 ± 7.26 | 64.75 ± 5.48 | 0.00 ± 0.00 | 0.00 ± 0.00 |
| ZeroGrad | 43.13 ± 9.18 | 59.34 ± 11.06 | 0.00 ± 0.00 | 0.00 ± 0.00 |
| N-FGSM | 48.47 ± 0.09 | 23.83 ± 0.21 | 21.21 ± 0.18 | 16.58 ± 0.18 |
| AAER | 42.23 ± 7.57 | 21.31 ± 3.20 | 19.43 ± 2.64 | 16.56 ± 0.14 |
| SORA (Ours) | 54.59 ± 0.38 | 22.84 ± 0.43 | 18.73 ± 0.65 | 14.31 ± 0.58 |

*Table 31.* Results for TINYIMAGENET on the SENet-18 architecture.

| Method | Clean | FGSM | PGD-10 | AA |
|---|---|---|---|---|
| Benign | 61.78 ± 0.32 | 0.92 ± 0.08 | 0.06 ± 0.01 | 0.00 ± 0.00 |
| FGSM | 42.82 ± 0.69 | 20.81 ± 0.46 | 18.94 ± 0.30 | 14.21 ± 0.40 |
| FGSM-RS | 46.41 ± 0.22 | 21.57 ± 0.12 | 19.02 ± 0.04 | 14.37 ± 0.09 |
| GradAlign | 47.00 ± 0.14 | 21.91 ± 0.25 | 19.51 ± 0.34 | 14.55 ± 0.31 |
| NuAT | 26.88 ± 0.37 | 16.91 ± 0.06 | 16.44 ± 0.14 | 11.95 ± 0.05 |
| ZeroGrad | 45.24 ± 0.45 | 21.36 ± 0.10 | 19.19 ± 0.01 | 14.55 ± 0.25 |
| MultiGrad | 44.87 ± 0.39 | 22.04 ± 0.22 | 19.86 ± 0.16 | 15.03 ± 0.17 |
| N-FGSM | 43.40 ± 0.57 | 21.88 ± 0.12 | 20.17 ± 0.19 | 15.25 ± 0.16 |
| AAER | 43.59 ± 0.17 | 21.79 ± 0.21 | 20.04 ± 0.36 | 15.26 ± 0.15 |
| ELLE | 48.42 ± 0.14 | 21.26 ± 0.12 | 18.41 ± 0.13 | 13.87 ± 0.19 |
| SORA (Ours) | 44.11 ± 0.16 | 22.21 ± 0.35 | 20.39 ± 0.36 | 15.20 ± 0.36 |
| PGD-2 | 48.07 ± 0.16 | 21.73 ± 0.22 | 19.21 ± 0.23 | 14.66 ± 0.17 |

*Table 32.* Results for PATHMNIST on the PreActResNet-18 architecture.

| Method | Clean | FGSM | PGD-10 | AA |
|---|---|---|---|---|
| Benign | 88.20 ± 0.98 | 12.07 ± 2.59 | 0.84 ± 0.43 | 0.47 ± 0.20 |
| FGSM | 36.54 ± 1.82 | 80.81 ± 2.21 | 1.94 ± 1.73 | 0.42 ± 0.16 |
| Free AT | 70.35 ± 7.52 | 49.06 ± 8.21 | 39.06 ± 8.21 | 31.62 ± 5.77 |
| FGSM-RS | 60.34 ± 8.01 | 86.96 ± 4.37 | 1.72 ± 0.95 | 1.03 ± 0.68 |
| GradAlign | 45.71 ± 13.74 | 38.91 ± 25.41 | 0.78 ± 1.10 | 0.77 ± 1.09 |
| NuAT | 82.73 ± 2.96 | 66.51 ± 4.54 | 25.69 ± 3.93 | 10.43 ± 3.60 |
| ZeroGrad | 67.22 ± 14.19 | 73.85 ± 13.12 | 5.69 ± 3.45 | 1.21 ± 0.52 |
| MultiGrad | 82.11 ± 3.26 | 56.90 ± 6.03 | 37.29 ± 4.88 | 24.94 ± 5.40 |
| N-FGSM | 74.86 ± 6.14 | 55.39 ± 14.43 | 18.84 ± 4.14 | 1.90 ± 0.77 |
| ATAS | 62.50 ± 10.88 | 71.48 ± 5.83 | 1.71 ± 0.58 | 0.94 ± 0.71 |
| AAER | 81.43 ± 3.36 | 63.18 ± 21.36 | 23.75 ± 6.20 | 1.89 ± 0.76 |
| ELLE | 79.80 ± 6.67 | 50.16 ± 6.19 | 40.52 ± 2.38 | 34.60 ± 1.05 |
| SORA (Ours) | 84.88 ± 0.99 | 55.59 ± 1.85 | 43.56 ± 1.66 | 35.54 ± 1.55 |
| PGD-2 | 83.49 ± 0.34 | 55.27 ± 0.96 | 47.50 ± 0.51 | 40.84 ± 0.23 |
| PGD-10 | 77.62 ± 1.45 | 57.63 ± 1.22 | 54.47 ± 0.89 | 50.70 ± 0.53 |
| TRADES | 75.25 ± 0.73 | 54.75 ± 0.86 | 51.94 ± 0.88 | 48.28 ± 0.74 |

*Table 33.* Results for PATHMNIST on the ResNet-18 architecture.

| Method | Clean | FGSM | PGD-10 | AA |
|---|---|---|---|---|
| Benign | 87.25 ± 2.33 | 14.12 ± 0.21 | 1.09 ± 0.89 | 0.86 ± 0.82 |
| FGSM | 34.03 ± 4.50 | 78.02 ± 2.80 | 0.62 ± 0.13 | 0.30 ± 0.20 |
| Free AT | 68.47 ± 10.46 | 45.38 ± 10.93 | 35.24 ± 8.11 | 25.52 ± 3.63 |
| FGSM-RS | 30.70 ± 7.65 | 87.13 ± 2.48 | 2.05 ± 0.21 | 1.50 ± 0.41 |
| GradAlign | 32.62 ± 4.94 | 38.07 ± 32.26 | 2.60 ± 2.11 | 0.72 ± 1.02 |
| NuAT | 88.45 ± 1.62 | 53.96 ± 1.64 | 17.73 ± 2.60 | 4.40 ± 3.24 |
| ZeroGrad | 55.37 ± 17.08 | 60.24 ± 21.54 | 2.66 ± 0.50 | 1.28 ± 0.72 |
| MultiGrad | 81.88 ± 1.15 | 57.89 ± 2.96 | 38.19 ± 4.10 | 29.15 ± 4.48 |
| N-FGSM | 71.90 ± 12.82 | 86.18 ± 5.95 | 14.74 ± 10.11 | 9.15 ± 6.46 |
| ATAS | 66.49 ± 4.94 | 70.95 ± 8.06 | 1.56 ± 0.75 | 0.68 ± 0.34 |
| AAER | 39.19 ± 29.07 | 43.60 ± 35.30 | 17.23 ± 1.98 | 9.16 ± 6.48 |
| ELLE | 79.65 ± 6.09 | 50.49 ± 4.63 | 41.75 ± 1.38 | 36.22 ± 0.49 |
| SORA (Ours) | 84.02 ± 0.36 | 57.73 ± 1.14 | 45.40 ± 0.90 | 38.94 ± 0.74 |
| PGD-2 | 82.79 ± 0.31 | 55.18 ± 0.86 | 47.46 ± 0.58 | 42.36 ± 0.89 |
| PGD-10 | 76.63 ± 1.66 | 55.51 ± 2.21 | 52.13 ± 1.99 | 48.41 ± 1.31 |
| TRADES | 18.64 ± 0.00 | 18.64 ± 0.00 | 18.64 ± 0.00 | 18.64 ± 0.00 |

*Table 34.* Results for PATHMNIST on the WideResNet-28 architecture.

| Method | Clean | FGSM | PGD-10 | AA |
|---|---|---|---|---|
| Benign | 89.45 ± 0.19 | 15.31 ± 0.41 | 2.14 ± 0.69 | 1.55 ± 0.40 |
| FGSM | 56.44 ± 5.94 | 85.24 ± 4.98 | 2.28 ± 0.24 | 1.33 ± 0.42 |
| Free AT | 78.32 ± 1.05 | 52.42 ± 2.05 | 37.79 ± 1.07 | 30.72 ± 1.11 |
| FGSM-RS | 32.01 ± 8.07 | 86.52 ± 2.67 | 2.04 ± 0.33 | 1.28 ± 0.69 |
| GradAlign | 39.37 ± 11.13 | 65.33 ± 8.27 | 1.98 ± 0.54 | 1.69 ± 0.69 |
| NuAT | 87.79 ± 1.85 | 55.68 ± 9.04 | 21.46 ± 1.23 | 13.01 ± 2.38 |
| ZeroGrad | 51.96 ± 19.24 | 85.71 ± 3.27 | 3.24 ± 1.02 | 2.20 ± 0.18 |
| MultiGrad | 85.79 ± 1.22 | 59.65 ± 0.94 | 40.97 ± 2.24 | 29.48 ± 2.50 |
| N-FGSM | 84.01 ± 3.06 | 90.80 ± 5.19 | 16.34 ± 8.90 | 8.77 ± 6.44 |
| ATAS | 55.53 ± 9.27 | 79.78 ± 5.60 | 2.34 ± 0.00 | 2.16 ± 0.24 |
| AAER | 84.20 ± 1.75 | 87.43 ± 3.00 | 22.79 ± 4.16 | 12.92 ± 3.77 |
| ELLE | 18.64 ± 0.00 | 18.64 ± 0.00 | 18.64 ± 0.00 | 18.64 ± 0.00 |
| SORA (Ours) | 86.53 ± 1.35 | 53.89 ± 1.26 | 39.34 ± 2.87 | 31.73 ± 0.28 |
| PGD-2 | 84.10 ± 0.64 | 54.88 ± 0.92 | 46.22 ± 0.68 | 38.68 ± 0.96 |
| PGD-10 | 79.16 ± 1.11 | 58.68 ± 0.34 | 53.74 ± 0.41 | 49.47 ± 0.65 |
| TRADES | 56.96 ± 27.11 | 43.20 ± 17.37 | 41.15 ± 15.92 | 38.00 ± 13.76 |

*Table 35.* Results for PATHMNIST on the SENet-18 architecture.

| Method | Clean | FGSM | PGD-10 | AA |
|---|---|---|---|---|
| Benign | 85.75 ± 2.97 | 11.77 ± 1.45 | 0.28 ± 0.06 | 0.13 ± 0.09 |
| FGSM | 55.05 ± 14.42 | 82.99 ± 2.73 | 3.88 ± 2.40 | 0.42 ± 0.12 |
| Free AT | 69.68 ± 3.46 | 47.64 ± 3.61 | 36.06 ± 2.70 | 29.12 ± 0.23 |
| FGSM-RS | 54.88 ± 14.63 | 89.31 ± 3.15 | 2.92 ± 2.47 | 1.03 ± 0.92 |
| GradAlign | 41.02 ± 14.03 | 65.05 ± 4.76 | 0.86 ± 0.87 | 0.68 ± 0.66 |
| NuAT | 86.37 ± 1.12 | 55.33 ± 7.60 | 24.19 ± 1.34 | 13.40 ± 2.96 |
| ZeroGrad | 60.06 ± 9.01 | 83.25 ± 5.65 | 1.49 ± 0.75 | 0.88 ± 0.68 |
| MultiGrad | 73.65 ± 1.64 | 49.05 ± 3.58 | 31.91 ± 0.74 | 21.31 ± 1.49 |
| N-FGSM | 79.33 ± 1.78 | 77.91 ± 15.64 | 23.99 ± 15.29 | 18.98 ± 16.42 |
| ATAS | 38.22 ± 16.22 | 81.83 ± 1.26 | 7.09 ± 7.50 | 2.59 ± 3.35 |
| AAER | 75.07 ± 1.54 | 69.57 ± 13.60 | 27.79 ± 14.85 | 22.74 ± 15.58 |
| ELLE | 76.45 ± 5.45 | 49.62 ± 2.31 | 44.10 ± 1.36 | 39.88 ± 1.23 |
| SORA (Ours) | 85.32 ± 2.22 | 56.35 ± 3.13 | 41.21 ± 5.30 | 34.76 ± 6.06 |
| PGD-2 | 82.37 ± 0.39 | 55.18 ± 0.88 | 48.43 ± 0.64 | 43.50 ± 1.07 |
| PGD-10 | 76.23 ± 1.67 | 53.72 ± 2.39 | 50.50 ± 2.13 | 46.98 ± 2.15 |
| TRADES | 18.64 ± 0.00 | 18.64 ± 0.00 | 18.64 ± 0.00 | 18.64 ± 0.00 |

*Table 36.* Results for TISSUEMNIST on the PreActResNet-18 architecture.

| Method | Clean | FGSM | PGD-10 | AA |
|---|---|---|---|---|
| Benign | 70.54 ± 0.03 | 14.27 ± 1.73 | 0.00 ± 0.00 | 0.00 ± 0.00 |
| FGSM | 51.80 ± 2.75 | 81.39 ± 12.14 | 0.28 ± 0.11 | 0.04 ± 0.02 |
| FGSM-RS | 52.32 ± 4.00 | 80.82 ± 3.75 | 0.35 ± 0.16 | 0.12 ± 0.05 |
| GradAlign | 55.41 ± 2.72 | 70.13 ± 15.40 | 16.23 ± 22.48 | 15.51 ± 21.91 |
| NuAT | 65.93 ± 0.30 | 27.30 ± 2.37 | 5.17 ± 1.20 | 9.91 ± 1.06 |
| ZeroGrad | 57.36 ± 1.65 | 69.01 ± 2.22 | 0.10 ± 0.01 | 0.01 ± 0.00 |
| MultiGrad | 61.79 ± 2.54 | 42.72 ± 4.99 | 16.92 ± 22.61 | 15.92 ± 22.30 |
| N-FGSM | 57.47 ± 0.10 | 49.75 ± 0.02 | 49.25 ± 0.06 | 48.15 ± 0.07 |
| ATAS | 31.95 ± 10.54 | 56.36 ± 6.04 | 0.08 ± 0.06 | 0.00 ± 0.00 |
| AAER | 57.47 ± 0.10 | 49.75 ± 0.02 | 49.25 ± 0.06 | 48.15 ± 0.07 |
| ELLE | 60.57 ± 0.48 | 47.61 ± 0.67 | 46.07 ± 0.92 | 42.67 ± 1.55 |
| SORA (Ours) | 58.86 ± 0.56 | 49.43 ± 0.37 | 48.55 ± 0.47 | 47.06 ± 0.58 |
| PGD-2 | 59.40 ± 0.23 | 49.15 ± 0.21 | 48.25 ± 0.26 | 46.75 ± 0.32 |
| PGD-10 | 57.75 ± 0.09 | 50.13 ± 0.04 | 49.62 ± 0.09 | 48.56 ± 0.18 |
| TRADES | 53.43 ± 0.00 | 45.30 ± 0.00 | 44.85 ± 0.00 | 42.71 ± 0.00 |

*Table 37.* Results for TISSUEMNIST on the ResNet-18 architecture.

| Method | Clean | FGSM | PGD-10 | AA |
|---|---|---|---|---|
| Benign | 70.52 ± 0.08 | 13.98 ± 0.84 | 0.00 ± 0.00 | 0.00 ± 0.00 |
| FGSM | 50.54 ± 0.16 | 83.73 ± 5.25 | 0.26 ± 0.14 | 0.08 ± 0.06 |
| FGSM-RS | 53.06 ± 1.58 | 85.55 ± 3.23 | 0.24 ± 0.22 | 0.06 ± 0.08 |
| GradAlign | 48.24 ± 4.25 | 58.93 ± 14.77 | 5.90 ± 7.58 | 2.78 ± 3.67 |
| NuAT | 65.95 ± 0.75 | 29.15 ± 1.96 | 5.40 ± 0.67 | 9.94 ± 0.96 |
| ZeroGrad | 55.70 ± 1.05 | 67.73 ± 2.22 | 0.14 ± 0.07 | 0.01 ± 0.01 |
| MultiGrad | 58.28 ± 0.30 | 50.34 ± 0.21 | 49.22 ± 0.06 | 47.61 ± 0.26 |
| N-FGSM | 57.38 ± 0.18 | 49.97 ± 0.03 | 49.51 ± 0.05 | 48.40 ± 0.08 |
| ATAS | 46.32 ± 18.21 | 42.74 ± 4.30 | 29.77 ± 21.03 | 28.40 ± 20.09 |
| AAER | 57.38 ± 0.18 | 49.97 ± 0.03 | 49.51 ± 0.05 | 48.39 ± 0.08 |
| ELLE | 59.18 ± 0.24 | 49.20 ± 0.11 | 48.43 ± 0.16 | 46.92 ± 0.21 |
| SORA (Ours) | 58.71 ± 0.26 | 49.67 ± 0.19 | 48.81 ± 0.17 | 47.36 ± 0.17 |
| PGD-2 | 59.27 ± 0.20 | 49.34 ± 0.10 | 48.55 ± 0.17 | 47.15 ± 0.21 |
| PGD-10 | 57.62 ± 0.18 | 50.23 ± 0.08 | 49.76 ± 0.09 | 48.66 ± 0.16 |
| TRADES | 53.70 ± 0.00 | 44.89 ± 0.00 | 44.45 ± 0.00 | 42.13 ± 0.00 |

*Table 38.* Results for TISSUEMNIST on the WideResNet-28 architecture.

| Method | Clean | FGSM | PGD-10 | AA |
|---|---|---|---|---|
| FGSM | 46.96 ± 5.88 | 80.31 ± 15.49 | 0.07 ± 0.06 | 0.01 ± 0.01 |
| FGSM-RS | 43.32 ± 7.65 | 68.66 ± 15.07 | 0.11 ± 0.15 | 0.01 ± 0.01 |
| N-FGSM | 57.69 ± 0.00 | 50.40 ± 0.00 | 49.99 ± 0.00 | 47.76 ± 0.00 |
| AAER | 57.83 ± 0.00 | 50.39 ± 0.00 | 50.00 ± 0.00 | 48.95 ± 0.00 |
| SORA (Ours) | 60.68 ± 0.87 | 49.33 ± 0.53 | 47.91 ± 0.99 | 45.92 ± 1.36 |

*Table 39.* Results for TISSUEMNIST on the SENet-18 architecture.

| Method | Clean | FGSM | PGD-10 | AA |
|---|---|---|---|---|
| Benign | 70.57 ± 0.10 | 13.50 ± 3.61 | 0.00 ± 0.00 | 0.00 ± 0.00 |
| FGSM | 47.26 ± 1.51 | 84.39 ± 2.67 | 0.26 ± 0.10 | 0.04 ± 0.03 |
| FGSM-RS | 56.10 ± 0.98 | 86.98 ± 3.53 | 0.44 ± 0.21 | 0.06 ± 0.09 |
| GradAlign | 52.50 ± 2.08 | 84.80 ± 5.94 | 0.02 ± 0.03 | 0.01 ± 0.01 |
| NuAT | 65.64 ± 0.02 | 26.34 ± 1.95 | 5.37 ± 0.59 | 10.52 ± 0.35 |
| ZeroGrad | 56.23 ± 0.91 | 70.19 ± 1.10 | 0.06 ± 0.03 | 0.00 ± 0.00 |
| MultiGrad | 59.25 ± 1.65 | 48.02 ± 2.99 | 32.96 ± 22.97 | 31.89 ± 22.50 |
| N-FGSM | 57.21 ± 0.05 | 50.01 ± 0.20 | 49.58 ± 0.20 | 48.56 ± 0.23 |
| ATAS | 48.39 ± 11.60 | 51.68 ± 5.24 | 18.15 ± 17.66 | 15.02 ± 17.16 |
| AAER | 57.21 ± 0.05 | 50.01 ± 0.20 | 49.58 ± 0.20 | 48.56 ± 0.23 |
| ELLE | 60.09 ± 0.47 | 48.21 ± 0.51 | 47.22 ± 0.69 | 44.36 ± 1.32 |
| SORA (Ours) | 57.90 ± 0.37 | 50.18 ± 0.30 | 49.48 ± 0.30 | 48.21 ± 0.44 |
| PGD-2 | 59.11 ± 0.13 | 49.55 ± 0.13 | 48.86 ± 0.13 | 47.43 ± 0.22 |
| PGD-10 | 57.19 ± 0.30 | 50.15 ± 0.12 | 49.73 ± 0.11 | 48.76 ± 0.11 |

## L. Fast Adversarial Training on Vision Transformers

Most prior work in fast adversarial training focuses on the ResNet family (PreActResNet, ResNet, WideResNet); therefore, we included these architectures for consistency and added SENet to increase architectural diversity.

We also explored Vision Transformers (ViTs) to assess the applicability of FAT to modern architectures. Since ViTs are data-hungry and typically perform best on large-scale datasets such as IMAGENET (Shao et al., 2022), our limited computational resources prevented experiments on IMAGENET. Instead, we trained ViTs on CIFAR-10, CIFAR-100, and PATHMNIST using SORA and multiple baselines.

Although Vision Transformers are primarily used in large-scale pre-training settings, we study the ViT-Small (Dosovitskiy et al., 2021) variant on small datasets, following Lin et al. (2024). In this setting, ViTs are generally outperformed by convolutional architectures such as those in the ResNet family. Consequently, the goal of these experiments is not to provide a comprehensive comparative benchmark, but rather to investigate whether FAT can be effectively applied to ViTs.

Consistent with prior findings (Lin et al., 2024) suggesting that ViTs are less susceptible to Catastrophic Overfitting (CO), we observed PertAlign values close to 1 during ViT training, indicating minimal loss distortion. In contrast, PertAlign values for ResNet architectures may drop to around $0.7$. These observations support the hypothesis that ViTs are less prone to CO, although further investigation is needed.

Overall, the results in Tables 40, 41, and 42 show that SORA enables fast adversarial training of ViTs even on small-scale datasets.

*Table 40.* Results for CIFAR-10 on the ViT-Small architecture.

| Method | Clean | FGSM | PGD-10 | AA |
|---|---|---|---|---|
| Benign | 72.95 | 0.34 | 0.00 | 0.00 |
| FGSM | 38.04 | 25.41 | 25.39 | 21.91 |
| FGSM-RS | 42.71 | 26.55 | 26.26 | 22.21 |
| GradAlign | 43.42 | 26.39 | 26.19 | 21.93 |
| NuAT | 58.39 | 22.54 | 20.38 | 16.81 |
| ZeroGrad | 40.03 | 25.74 | 25.64 | 21.77 |
| MultiGrad | 38.55 | 25.71 | 25.58 | 22.00 |
| N-FGSM | 36.56 | 24.82 | 24.84 | 21.52 |
| AAER | 36.56 | 24.82 | 24.84 | 21.52 |
| ELLE | 45.02 | 26.32 | 25.92 | 22.05 |
| SORA | 37.48 | 25.23 | 25.16 | 21.71 |
| PGD-2 | 45.70 | 26.82 | 26.52 | 22.21 |

*Table 41.* Results for CIFAR-100 on the ViT-Small architecture.

| Method | Clean | FGSM | PGD-10 | AA |
|---|---|---|---|---|
| Benign | 44.61 | 0.41 | 0.01 | 0.00 |
| FGSM | 23.01 | 11.37 | 11.31 | 8.28 |
| FGSM-RS | 26.84 | 12.47 | 12.29 | 8.77 |
| GradAlign | 26.98 | 12.24 | 11.94 | 8.64 |
| NuAT | 16.94 | 9.53 | 9.46 | 6.49 |
| ZeroGrad | 25.71 | 11.99 | 11.81 | 8.57 |
| MultiGrad | 23.06 | 11.47 | 11.21 | 8.22 |
| N-FGSM | 22.53 | 11.50 | 11.44 | 8.42 |
| AAER | 22.51 | 11.52 | 11.44 | 8.36 |
| ELLE | 28.05 | 11.88 | 11.44 | 8.28 |
| SORA | 22.98 | 11.52 | 11.51 | 8.31 |
| PGD-2 | 29.08 | 12.13 | 11.80 | 8.61 |

*Table 42.* Results for PATHMNIST on the ViT-Small architecture.

| Method | Clean | FGSM | PGD-10 | AA |
|---|---|---|---|---|
| Benign | 80.85 | 17.87 | 0.47 | 0.47 |
| FGSM | 56.96 | 82.40 | 8.15 | 8.02 |
| FGSM-RS | 45.85 | 74.16 | 8.22 | 7.24 |
| GradAlign | 32.31 | 84.07 | 26.66 | 11.34 |
| NuAT | 83.40 | 86.48 | 15.68 | 13.73 |
| ZeroGrad | 48.36 | 84.72 | 9.43 | 9.36 |
| MultiGrad | 42.92 | 70.15 | 10.49 | 9.22 |
| N-FGSM | 51.89 | 67.70 | 20.43 | 7.47 |
| AAER | 18.64 | 18.64 | 18.64 | 18.64 |
| ELLE | 18.64 | 18.64 | 18.64 | 18.64 |
| SORA | 76.70 | 51.96 | 34.25 | 25.74 |
| PGD-2 | 74.28 | 49.89 | 46.94 | 43.09 |

# M. Research Statement

## M.1. Limitations

### M.1.1. SINGLE-STEP VERSUS MULTI-STEP

As shown in Figures 6 and 7 and the accompanying tables, SORA achieves the best clean-robust accuracy trade-off while remaining one of the most efficient fast adversarial training methods. In most cases, it attains higher clean accuracy than multi-step methods while maintaining competitive robust accuracy. Even on PATHMNIST, where there is a substantial gap in robust accuracy between multi-step and single-step methods, SORA achieves superior clean and robust accuracy compared to other fast methods, while being significantly more efficient than multi-step variants.

That said, in use cases where computational cost is not a major concern, multi-step methods may be more suitable than single-step approaches.

### M.1.2. GENERALIZABILITY VERSUS HYPERPARAMETER SEARCH

Although SORA demonstrates superior generalizability across a wider range of datasets and architectures compared to other single-step methods when using a fixed set of hyperparameters, some alternative methods can achieve robust accuracy close to SORA in certain scenarios through setting-specific hyperparameter search. However, such tuning typically incurs significantly higher computational cost in practice. For more details, see Appendix H.

### M.1.3. COMPUTATIONAL CONSTRAINTS

We attempted to provide a comprehensive evaluation of the baselines by considering as many dataset-architecture pairs as possible. Our evaluations were more diverse in terms of dataset and architecture diversity compared to the Fast AT benchmark (Pan & Yao, 2026). However, certain baselines required substantial memory, which prevented us from evaluating them on larger dataset-architecture pairs. These limitations are discussed in more detail in Appendix K.

## M.2. Reproducibility

### M.2.1. MAIN RESULTS

General hyperparameters and optimizer details are provided in Section 6.1. As recommended by the original authors of the baseline models, we provide a comprehensive account of their hyperparameters and the rationale for their selection in Appendix B. Dataset details and corresponding augmentation strategies are also documented in Appendix D.

All experimental configurations, including random seeds, are accessible in our code repository. Due to computational constraints, we conducted experiments using three distinct random seeds to assess the stability of our results. The only exception to this rule is our results on the IMAGENET-100 which was computationally expensive and the results are due to a single run without any tuning for our method whereas the baselines benefit from hyperparameters reported by the original papers on the same dataset if available. The detailed outcomes for each seed are available in Appendix K.

### M.2.2. FIGURES

The code for generating all figures is included in our repository. The data for Figures 2 and 10 were produced by our main codebase. For visual clarity, these figures were subsequently generated by processing and combining this raw data. We emphasize that the co-occurrence of Catastrophic Overfitting with drops in PertAlign (and GradAlign) to zero was consistently observed across all experimental settings. The figures are intended to illustrate this robust trend, which holds irrespective of specific configurations.

To ensure a fair and accurate comparison of computational cost in Figures 7 and 16, we measured the wall-clock time for the core training loop, explicitly excluding overhead from saving and loading model checkpoints. Each run was repeated multiple times to verify the stability of timing measurements. System resources were monitored to ensure that no extraneous processes interfered, and we confirmed that GPU thermal throttling did not impact the reported durations.

M.2.3. ABLATION STUDIES

The configurations required to replicate our ablation studies are detailed in Appendix F. The results can be directly obtained by executing the corresponding scripts in our codebase.

## M.3. Fair Baseline Comparisons

A central finding of this work is that existing methods often perform poorly on previously untested datasets, such as PATHMNIST and TISSUEMNIST (Yang et al., 2023). This issue is not unique to this field; other areas of machine learning have faced similar challenges due to over-reliance on limited benchmark sets. Evaluation on a narrow range of datasets can create an illusion of progress and lead to indirect overfitting on test data through repeated hyperparameter tuning, even with proper validation splits (Hardt, 2026).

To mitigate this risk and promote generalizability, we evaluated all methods on previously untested datasets and architectures. Nevertheless, we ensured that baseline methods were given a fair chance by diligently tuning their hyperparameters. We adhered to the guidelines provided by the original authors, using their exact hyperparameters for established datasets. For new datasets, we started from the authors' recommended parameters and made reasoned adjustments to improve performance.

The time invested in hyperparameter search for many baseline methods was equal to or greater than that spent on our own model. While our search was necessarily limited by computational resources and the desire to avoid excessive fine-tuning, we carefully reasoned about the impact of key hyperparameters to achieve competitive results. Certain hyperparameter adjustments were found to benefit all methods uniformly on specific datasets; these improved, universally beneficial settings are the ones we report.

A key motivation for fast adversarial training is efficiency and practical viability (Wong et al., 2020). If extensive tuning is required to achieve competitive results, one might as well revert to more reliable but computationally expensive methods like PGD. This rationale underpins our commitment to using a fixed hyperparameter set across all datasets and architectures for our method, while permitting necessary adjustments for baselines to ensure fairness.

Our hyperparameter search was intentionally constrained to PreActResNet-18 and WideResNet-28-10 on CIFAR-10 and CIFAR-100, since nearly all other baselines have also examined these settings (some use WideResNet-34-10 instead). Since these settings were not challenging enough, for our method and the baselines, we did some limited experiments on PATHMNIST as well. The resulting parameters were then fixed for evaluations on TISSUEMNIST and TINYIMA-GENET (Le & Yang, 2015). For TINYIMAGENET, we incorporated published baseline hyperparameters where available; for TISSUEMNIST, no additional tuning was performed.

Notably, our method's key parameters ($\beta$ for reducing PertAlign variance, and $\alpha_0/\alpha_{\max}$ for step-size constraints) required minimal tuning. All results on CIFAR-100, TISSUEMNIST, IMAGENET-100 and TINYIMAGENET represent **first-run outcomes without modification**, whereas baselines benefited from previously optimized settings when available. Similarly, all results on ResNet-18, WideResNet-28, and SENet-18 are also **first-run outcomes without modification**, while we allowed baselines to benefit from previously optimized settings when available.

## M.4. LLM Usage

Large Language Models (LLMs) were used to assist with proofreading and polishing the text of this manuscript. LLMs were also used for initial code scaffolding in some instances (primarily for generating plots or loading raw data). All code generated with LLM assistance was rigorously reviewed, tested, and validated by multiple co-authors to ensure its correctness and integration into our codebase.

