# OpenReview forum: "SORA: Free Second-Order Attacks in Fast Adversarial Training"
_ICML.cc/2026/Conference — ICML 2026 regular_

### Official Review · Reviewer_w3kA · 2026-02-15

**Soundness:** 2
**Presentation:** 2
**Significance:** 1
**Originality:** 1
**Overall Recommendation:** 2
**Confidence:** 5

**Summary:**

This paper investigates the catastrophic overfitting in adversarial training and proposes a metric called PertAlign to predict whether or not CO will happen. Based on this quantitative metric, the authors propose SORA to adaptively adjust the step size to adjust the step size to prevent diverged landscape and thus CO. Numerical experiments are conducted to validate the effectiveness of SORA on various experiments.

**Compliance With Llm Reviewing Policy:**

Affirmed.

**Final Justification:**

I appreciate the authors' responses in the rebuttal phase. However, after reading the comments from other reviewers, I still believe this manuscript needs considerable edits before being considered a top-iter conference, especially considering the limited key insight (distorted loss landscape leads to catastrophic overfitting in fast adversarial training is well known), the missing baselines and the evaluation metrics (i.e. comprehensive robustness evaluation) in the original manuscript. I suggest the authors incorporate these new results in the manuscript and add comprehensive discussion to convince the readers why and how the proposed methods outperforms various baselines in this task.

**Key Questions For Authors:**

The authors are encouraged to address my concerns in the weakness part and I would be open for discussion in the rebuttal period, especially:

1. What is new in this manuscript's motivation, compared with existing literature studying the relationships between CO and loss landscape.

2. Please address my concerns in the experimental part (second point in weakness), including in which cases SORA can perform the best, and more large-scale experiments (like ViT on ImageNet).

3. Please discuss the approximation error in Lemma 4.1.

4. The key robustness metric should be based on AutoAttack.

5. Include NuAT and ATTA as baselines for comparison.

**Limitations:**

The authors have not discussed the societal limitations. This work is generic and may be applied for various downstream applications, but I have not seen key negative societal impact raised by this method.

**Strengths And Weaknesses:**

Strength:

++ The motivation of PertAlign has theoretical guarantee.

++ PertAlign introduces little computational overhead.

Weakness:

-- Limited Novelty: the relationship between the loss landscape (i.e. landscape with high curvature) and the CO issue is well-known and well studies in existing literatures mentioned in this manuscript.

-- Weak experimental support: Based on the results in Table 10 (the robustness performance should be measured in AutoAttack instead of PGD, which is not strong enough), SORA does not outperform the existing baselines, especially ATAS and AAER. In addition, the comprehensive experiments of this work are mainly based on relatively small datasets (CIFAR10) and architectures (ResNet18), results on larger datasets (images of higher resolution, such as ImageNet) and large models (like ViT) should be encouraged, since one of the main contribution of this work is the computational efficiency.

-- the motivation of PerAlign is based on Taylor expansion (Lemma 4.1). However, the step size is usually quite large, especially in one-step attack, the step size is usually larger than $\epsilon$. Therefore, I am concerned whether or not the approximation and thus the result in Lemma 4.1 is sound.

-- The results, such as ablation studies in Appendix F, are mainly demonstrated by PGD-10 attack, which is not a reliable evaluation metric, the authors should use AutoAttack to demonstrate the results.

-- Some key baselines are missing, such as ATTA and NuAT, which also tackles the stability of fast adversarial training.

[1] ATTA: Efficient adversarial training with transferable adversarial examples, CVPR 2020.

[2] NuAT: Towards efficient and effective adversarial training. NeurIPS 2021.

---

> ### Author Rebuttal · Authors · 2026-03-29
>
> Dear reviewer w3kA,
>
> Thank you for your detailed and thorough review. Below we address the concerns and questions you raised.
>
> ## W1 and Q1
> As you noted, it is well known that at the onset of CO the loss landscape becomes distorted and highly non-linear.
> In this work, we characterize the structure of this non-linearity and **show that it is not jagged but instead follows a relatively smooth curve**.
> This observation allows us to exploit this specific form of non-linearity to avoid CO by using a second-order model, leading to improved robustness and higher efficiency with SORA.
> Furthermore, this property enables us to measure the degree of non-linearity using PertAlign, which effectively predicts CO with negligible computational overhead and **can be applied to any other single-step method**.
> We believe this metric may also be useful for developing improved fast adversarial training methods in future work.
>
> ## W2 and Q2
> Thank you for pointing out the results in Table 10.
> The motivation for this experiment was to measure the robustness density with respect to $\varepsilon$ at test time.
> Specifically, we trained a model with each method using $\varepsilon = 8/255$ and evaluated how rapidly the robust accuracy drops as $\varepsilon$ increases.
>
> We observe that methods such as FGSM-RS, GradAlign, ZeroGrad, MultiGrad, ELLE, and ATAS achieve high robust accuracy for small $\varepsilon$ values but lose robustness rapidly as $\varepsilon$ increases.
> In contrast, methods such as NFGSM, AAER, and SORA maintain higher robust accuracy across a wider range of $\varepsilon$ values, exhibiting a slower degradation.
>
> For comparison under the standard adversarial training setting, please refer to Figure 5, Table 1, and Appendix J, where results are reported using AutoAttack.
> As shown, SORA is the only method that consistently matches or exceeds state-of-the-art performance across all settings, while AAER lags behind in robust accuracy on ImageNet-100 and PathMNIST, as well as in clean accuracy.
>
> We also conducted experiments on higher-resolution datasets such as TinyImageNet and ImageNet-100 (Table 1 and Appendix J).
> As requested, we additionally evaluated SORA on ViT architectures; these results are provided in the "W5 and Q5" section of our response to reviewer kRdE.
>
> ## W3 and Q3
> Although the step size $\alpha$ can be larger than $\varepsilon$ (often up to roughly twice its value), $\varepsilon$ itself cannot be very large due to the imperceptibility constraint inherent in the definition of adversarial examples.
> Since the approximation error in our derivation corresponds to $\mathcal{O}(\alpha^3)$, the resulting error term remains significantly smaller than the value of PertAlign.
> For further details and a numerical example, please see the "W2 and Q3" section of our response to reviewer pRMd.
>
> ## W4 and Q4
> Thank you for the suggestion. We report AutoAttack (AA) for all main results (Table 1 and Appendix J). However, for the ablation studies we report PGD-10 for practicality, as these experiments require training and evaluating a large number of method, model, and dataset variants, and AA evaluation is substantially more computationally expensive. PGD-10 provides a reliable proxy for relative robustness trends while keeping the evaluation cost manageable. Importantly, since the final models are evaluated using AA in the main results, the robustness conclusions of the paper remain based on the stronger evaluation protocol. If helpful, we can include AA results for selected ablation settings in the camera-ready version.
>
> ## W5 and Q5
> Thank you for the baseline recommendations.
> We included ATAS [1] instead of ATTA since ATAS is a more advanced and competitive extension of ATTA.
> In addition, the ATAS paper analyzes the limitations of ATTA in detail.
> Overall, we attempted to include a diverse and representative set of well-known and competitive baselines, resulting in a total of 13 comparison methods.
> For more details on our baseline selection criteria, please see the explanation in the "W4 and Q2" section of our rebuttal to reviewer pRMd.
>
> As requested, we also include NuAT results at the following link: [URL](https://drive.google.com/drive/folders/1s2cM55EcOAM_iTaR3FPl0WMW--vPuFtY?usp=sharing).
> We evaluated NuAT on CIFAR-10, CIFAR-100, TinyImageNet, PathMNIST, and TissueMNIST using PreActResNet-18, ResNet-18, WideResNet-28, and SENet-18 architectures.
> We report clean, FGSM, and PGD-10 accuracies, with all results averaged across three seeds. The NuAT authors recommend a default hyperparameter of $\lambda = 4$; however, for some datasets training becomes unstable with this value.
> In those cases, we repeated the experiments using $\lambda = 1$. We also report NuAT results for ViT models in our response to reviewer kRdE.
>
> [1] Fast Adversarial Training with Adaptive Step Size, IEEE Transactions on Image Processing, 2022.

---

> > ### Author Rebuttal · Reviewer_w3kA · 2026-04-02
> >
> > I thank the authors for the rebuttal, which has addressed part of my concerns. I noticed that part of the results in Appendix J do not have AA results, especially for larger datasets. I am confused by some nan results under AA as well. In addition, the actual performance of SORA is low and the gap between multiple-step variant is large (like Table 21 - 29). I double the practical value of the proposed method in challenging situations. Regarding the new result from NuAT, robust accuracy under AA is expected as well.
> >
> > Given the conclusion of this manuscript is not very surprising, and the lack of some supporting evidence, after checking the comments from other reviewers, I decide to maintain my current rating.

---

> > > ### Author Response · Authors · 2026-04-04
> > >
> > > Thank you for your response.
> > > Due to our limited compute resources, we were initially unable to evaluate AA for all tables.
> > > We have now completed AA evaluations for all remaining tables, as well as for the NuAT and Free AT baselines and the ViT architecture.
> > > These results are available at this [URL](https://drive.google.com/drive/folders/1_NsJCrd-E0dVOrGh4acesI_HrfVMzkHj?usp=sharing).
> > >
> > > Regarding performance, as shown in Figures 5 and 6 and in the accompanying tables, SORA achieves the best clean–robust accuracy trade‑off while remaining one of the most efficient fast adversarial training methods.
> > > In most cases, it attains higher clean accuracy than multi‑step methods while maintaining competitive robust accuracy.
> > > Even on PathMNIST, where there is a substantial gap between multi‑step and single‑step methods in robust accuracy, SORA attains superior clean and robust accuracy compared to other fast methods, while being significantly more efficient than multi‑step variants, as presented in [URL](https://drive.google.com/drive/folders/1_NsJCrd-E0dVOrGh4acesI_HrfVMzkHj?usp=sharing).
> > >
> > > To summarize our contributions:
> > >
> > > - We provide a more detailed analysis of the dynamics of Catastrophic Overfitting (CO) and show that a second‑order perspective offers a promising solution.
> > > - However, classical second‑order methods are computationally expensive and therefore unsuitable for fast adversarial training.
> > > - We show, both theoretically and experimentally, that by reusing gradients already computed during standard AT, these costs can be avoided.
> > > - Based on this insight, we introduce PertAlign, an efficient and reliable metric for predicting CO that can also be used by other methods to develop better variants.
> > > - We also introduce SORA, which uses more accurate second‑order attack approximations while maintaining efficiency comparable to the fastest first‑order methods, achieving generalizable state‑of‑the‑art performance across diverse settings.
> > >
> > > For these reasons, we believe our work offers meaningful progress in this area and may be of value to the community.
> > >
> > > We hope that these additional results and clarifications address your concerns, and we would greatly appreciate it if you would consider revising your evaluation.
> > > If any questions or concerns remain, we would be more than happy to address them.

---

### Official Review · Reviewer_kRdE · 2026-03-04

**Soundness:** 2
**Presentation:** 2
**Significance:** 3
**Originality:** 3
**Overall Recommendation:** 2
**Confidence:** 3

**Summary:**

This work proposes SORA, which dynamically controls the perturbation step size based on the geometry of the loss surface to prevent catastrophic overfitting. This work proposes a metric called PertAlign, which measures the local non-linearity of the loss surface. PertAlign is defined as the cosine similarity between the gradient of the loss with respect to the input at the first attack point and the gradient with respect to the input at the perturbed point used for training. The paper presents several lemmas and proofs showing that the proposed metric is linked to local curvature and can be used to predict catastrophic overfitting. Based on these insights, SORA adaptively sets the step size used for the inner maximization when generating adversarial examples, aiming to avoid epsilon overfitting and the onset of catastrophic overfitting. This work evaluates the proposed method on several benchmarks, including ImageNet-100, and compares it against multiple fast (single-step) adversarial training baselines. The results suggest that SORA matches or surpasses the tested baselines while incurring negligible additional computational cost for computing the metric and adapting the step size.

**Compliance With Llm Reviewing Policy:**

Affirmed.

**Final Justification:**

I sincerely appreciate the authors’ response. I suggest adding an explicit gradient obfuscation test that evaluates the individual guidelines discussed in the gradient obfuscation literature. You may also refer to Section 4.4 of the following paper [6]. I also recommend refining the plots and tables so that their formatting is consistent with the rest of the main paper. Because I believe there are still several aspects of the paper that need improvement, I maintain my score.

[6] Hwang et al. “Adversarial Training with Stochastic Weight Average.” ICIP, 2021.

**Key Questions For Authors:**

- Can you include comparisons against additional fast adversarial training methods that are currently omitted?

- Beyond Table 1, can you report AutoAttack (or similarly strong multi-attack) robustness for the broader set of datasets and architectures evaluated in Figure 5, rather than only relative PGD-10 accuracy?

- Have you performed explicit checks for gradient masking, and can you provide evidence that the reported robustness is not an artifact of obfuscated gradients across the main experimental settings?

- Can you clarify why Figure 5 uses a line plot instead of a bar plot, explain the meaning of the shaded gray region, and address the apparent missing or unconnected data points?

- Do the reported conclusions hold for more recent architectures such as Vision Transformers?

**Limitations:**

yes.

**Strengths And Weaknesses:**

# Strengths

- The proposed metric incurs negligible additional cost because the quantities needed to compute it are already available in standard adversarial training (AT).

- The paper identifies and analyzes epsilon overfitting, showing that using a fixed perturbation magnitude can cause the model to overfit to a narrow range of epsilon values and thereby exacerbate catastrophic overfitting.

- The paper includes training time and memory usage plots, which help quantify the efficiency of the proposed method and contextualize it against competing baselines.

- The paper provides a clear algorithm box, which makes the proposed method easy to follow and implement.


# Weaknesses


- This work does not include several well-known fast adversarial training methods, such as [1],[2],[3],[4].

- The main table (Table 1) reports results for only two datasets and one model. For the other datasets and models the paper claims to test, Figure 5 reports only relative PGD accuracy and does not report AutoAttack accuracy.

- The paper does not include an explicit gradient-obfuscation check [5]. Although AutoAttack can help detect gradient masking, it is reported only in Table 1, which covers just two datasets (PathMNIST and ImageNet-100) and a single model (PreActResNet-18).

- Figure 5 is hard to read. It is unclear why a line plot is used instead of a bar plot. In addition, some data points appear to be missing, and some line segments are not connected. The meaning of the shaded gray region is also not explained.

- The evaluated architectures are somewhat dated. The experiments focus on CNN-based models (e.g., ResNet, PreActResNet, and WideResNet). Evaluating on more recent architectures such as Vision Transformers would strengthen the paper’s generality claims.

[1] Shafahi et al., “Adversarial training for free!,” NeurIPS 2019.

[2] Zheng et al., “Efficient adversarial training with transferable adversarial examples,” CVPR 2020.

[3] Ye et al., “AMATA: An annealing mechanism for adversarial training acceleration,” AAAI 2021.

[4] Jung et al., “Fast adversarial training with dynamic batch-level attack control,” DAC 2023.

[5] Athalye et al., “Obfuscated gradients give a false sense of security: Circumventing defenses to adversarial examples,” ICML 2018.

---

> ### Author Rebuttal · Authors · 2026-03-29
>
> Dear reviewer kRdE,
>
> Thank you for your thoughtful and valuable review.
> We appreciate your recognition of the potential and importance of PertAlign and Epsilon Overfitting, and we are glad that the plots and algorithm box were helpful.
>
> ## W1 and Q1
> Thank you for the baseline recommendations.
> We carefully reviewed the suggested methods.
> The main reason for excluding them from our baseline set is that they are not competitive with current state-of-the-art fast adversarial training methods (e.g., SORA, NFGSM, AAER),  which are close to the time and memory efficiency of vanilla FGSM training.
>
> Specifically, both Free AT and AMATA require multi-step PGD updates per batch, which substantially increases training time.
> Similarly, ATTA requires memorizing adversarial perturbations for the entire training set, making it significantly more memory intensive.
> Instead, we include ATAS, which is a more advanced extension of ATTA and is more closely related to SORA.
> Nevertheless, since Free AT is one of the most widely cited works in this area, we will include it in our baseline comparisons.
>
> Overall, we compare SORA against 13 methods, covering a diverse set of competitive and well-known baselines.
> For further details on our baseline selection criteria, please refer to the "W4 and Q2" section of our rebuttal to reviewer pRMd.
>
> ## W2 and Q2
> We report complete results on all datasets in Appendix J, including AutoAttack (AA) accuracies for all evaluated methods, as referenced in the paper.
>
> ## W3 and Q3
> In addition to Table 1, we report AA accuracies in Tables 17, 18, 19, 20, 21, 22, 23, 24, 29, 30, and 31.
> These results cover CIFAR-10, CIFAR-100, PathMNIST, and ImageNet-100 datasets evaluated on ResNet-18, PreActResNet-18, SENet-18, and WideResNet-28 architectures, and are provided in Appendix J.
>
> Following the criteria outlined by [1], several behaviors can indicate gradient masking:
> - **One-step attacks outperform iterative attacks.** We report both FGSM and PGD accuracies across all settings, and in all cases iterative attacks are stronger, indicating that SORA does not suffer from this issue.
> - **Black-box attacks outperform white-box attacks.** The Square attack included in AA is a black-box attack and would substantially reduce robust accuracy in the presence of gradient masking. Since no such drop is observed, this further suggests that SORA does not exhibit gradient masking.
> - **Increasing the distortion bound does not increase attack success.** As shown in Figure 2 and the ablation studies in Appendix F, increasing $\varepsilon$ consistently increases PGD attack success, which is inconsistent with gradient masking behavior.
>
> ## W4 and Q4
> The primary motivation behind Figure 5 was to provide a compact summary of results for 10 baselines across all combinations of 5 datasets and 4 architectures. Due to space constraints, we used a line plot so that each point represents the relative performance of a method under a specific dataset-architecture combination. We would be happy to incorporate a complementary bar plot in the camera-ready version if this would improve clarity.
>
> The shaded regions in Figure 5 correspond to one standard deviation from the mean, averaged over three random seeds. The figure illustrates that, unlike other methods, SORA consistently performs well across all datasets and architectures. In contrast, other baselines often fail to match SORA’s performance and exhibit Catastrophic Overfitting in certain settings. As noted in Appendix J, missing entries correspond to baseline failures, which are detailed in lines 1987-1989 and 2008-2011.
>
> ## W5 and Q5
> Thank you for the suggestion.
> Most prior work in fast adversarial training focuses on the ResNet family (PreActResNet, ResNet, WideResNet); thus, we included these architectures for consistency and added SENet to enhance diversity.
>
> We also explored Vision Transformers (ViT) to assess SORA’s generalizability.
> Since ViTs are data-hungry and best suited for large datasets like ImageNet [2], limited resources prevented ImageNet experiments.
> Instead, we trained ViTs on CIFAR‑10, CIFAR‑100, and PathMNIST with SORA and multiple baselines (results at [URL](https://drive.google.com/drive/folders/1h8mM8NUXlWu8X0GoVmcPapROl3cf7jjh?usp=sharing)).
>
> Consistent with prior findings [3] that ViTs are less susceptible to CO, we observed PertAlign values near 1 during ViT training, indicating minimal loss distortion, unlike ResNets, where PertAlign may drop to $\approx 0.7$.
> This supports the hypothesis that ViTs are less prone to CO but further investigations are required.
>
> Overall, the results show that SORA enables efficient adversarial training of ViTs even on small‑scale datasets.
>
> ## References
>
> [1] Obfuscated Gradients Give a False Sense of Security, ICML 2018
>
> [2] On the Adversarial Robustness of Vision Transformers, TMLR 2022
>
> [3] Layer-Aware Analysis of Catastrophic Overfitting: Revealing the Pseudo-Robust Shortcut Dependency, ICML 2024

---

> > ### Author Rebuttal · Reviewer_kRdE · 2026-03-31
> >
> > I thank the authors for their detailed rebuttal and helpful clarifications. Regarding the response to W2, I checked Appendix J and found that the tables have inconsistent columns: some datasets include an AutoAttack (AA) column, while others do not. Moreover, AutoAttack alone is not sufficient to establish the absence of gradient masking, so adding more explicit gradient-masking evaluations would strengthen the paper. For the omitted baselines, it would also have been helpful to report at least one representative result. In addition, I am hesitant to access external URLs. I will consider these clarifications along with the paper and the other reviews in forming my final recommendation.

---

> > > ### Author Response · Authors · 2026-04-03
> > >
> > > Thank you for your prompt reply.
> > > Unfortunately, due to time constraints, we were unable to report AA accuracy for all configurations.
> > > However, we reported AA for most of the 21 tables and are committed to including AA results for the remaining configurations by the end of the discussion period.
> > > **Update: We have included AutoAttack for all remaining tables. Due to the response character limit we apologize as we were forced to present them as a link available at [URL](https://drive.google.com/drive/folders/1_NsJCrd-E0dVOrGh4acesI_HrfVMzkHj?usp=sharing).**
> > >
> > > Aside from the signs indicating the absence of gradient masking that we discussed in the "W3 and Q3" section of our original response, to the best of our knowledge almost all prior work in this literature reports AutoAttack (AA) results to demonstrate the absence of gradient masking.
> > > For this reason, we followed the same practice. If you have a specific gradient masking evaluation in mind that you would like us to include, we would greatly appreciate your suggestion.
> > >
> > > As requested, following our discussion we report Free AT results below. We present results averaged over three seeds across the CIFAR‑10, CIFAR‑100, and PathMNIST datasets using the PreActResNet‑18, ResNet‑18, WideResNet‑28, and SENet‑18 architectures.
> > >
> > > ## CIFAR10
> > > |  Architecture  |      Clean       |   AutoAttack   |
> > > |:--------------:|:----------------:|:--------------:|
> > > | PreActResNet18 |  77.53 (-2.36%)  | 42.23 (-1.65%) |
> > > |    ResNet18    |  77.61 (-2.07%)  | 42.10 (-1.92%) |
> > > |    SENet18     |  77.43 (-2.74%)  | 42.09 (-1.94%) |
> > > |  WideResNet28  |  80.62 (-2.82%)  | 44.28 (-0.82%) |
> > >
> > > ## CIFAR100
> > > |  Architecture  |     Clean      |   AutoAttack    |
> > > |:--------------:|:--------------:|:---------------:|
> > > | PreActResNet18 | 50.28 (-4.46%) | 19.54 (-2.38%)  |
> > > |    ResNet18    | 49.97 (-4.53%) | 19.47 (-2.15%)  |
> > > |    SENet18     | 48.56 (-5.05%) | 19.30 (-2.25%)  |
> > > |  WideResNet28  | 53.23 (-5.35%) | 20.99 (-2.71%)  |
> > >
> > > ## PathMNIST
> > > |  Architecture  |      Clean      |   AutoAttack    |
> > > |:--------------:|:---------------:|:---------------:|
> > > | PreActResNet18 | 70.35 (-14.66%) | 31.62 (-3.62%)  |
> > > |    ResNet18    | 68.47 (-15.55%) | 25.52 (-12.63%) |
> > > |    SENet18     | 69.68 (-14.63%) | 29.12 (-8.02%)  |
> > > |  WideResNet28  | 78.32 (-8.21%)  | 30.72 (-1.36%)  |
> > >
> > > We used $m = 8$ batch replays, as recommended by the original authors. As shown in the tables, **Free AT is strictly dominated by SORA in both clean and robust accuracy across all configurations**, while also requiring approximately 35\% more training time.
> > > The average accuracy difference between SORA and Free AT is presented in the parentheses.
> > >
> > > Due to the response character limit, we were required to present the ViT results via a link; however, we also include them here for completeness.
> > >
> > > ## CIFAR10 - ViT-Small
> > > |  Method   | Clean | FGSM  |  PGD  |  AA   |
> > > |:---------:|:-----:|:-----:|:-----:|:-----:|
> > > |  Benign   | 62.23 | 0.17  | 0.02  | 0.00  |
> > > |   FGSM    | 42.71 | 28.23 | 28.13 | 23.92 |
> > > |  FGSM-RS  | 46.49 | 28.60 | 28.53 | 23.71 |
> > > | GradAlign | 46.80 | 28.67 | 28.31 | 23.86 |
> > > |   NuAT    | 31.45 | 25.46 | 25.44 | 22.96 |
> > > | ZeroGrad  | 44.99 | 29.24 | 29.11 | 24.73 |
> > > | MultiGrad | 42.88 | 28.20 | 28.10 | 23.95 |
> > > |   NFGSM   | 42.10 | 27.89 | 27.95 | 24.02 |
> > > |   AAER    | 41.72 | 27.88 | 27.80 | 24.05 |
> > > |   ELLE    | 49.49 | 28.39 | 27.92 | 23.43 |
> > > |   SORA    | 41.95 | 28.26 | 28.10 | 24.33 |
> > > |   PGD2    | 49.88 | 28.53 | 27.97 | 23.69 |
> > >
> > > ## CIFAR100 - ViT-Small
> > > |  Method   | Clean | FGSM  |  PGD  |  AA  |
> > > |:---------:|:-----:|:-----:|:-----:|:----:|
> > > |  Benign   | 37.02 | 0.74  | 0.07  | 0.02 |
> > > |   FGSM    | 19.35 | 10.67 | 10.61 | 8.32 |
> > > |  FGSM-RS  | 20.61 | 10.73 | 10.50 | 8.17 |
> > > | GradAlign | 21.05 | 10.69 | 10.56 | 8.16 |
> > > |   NuAT    | 14.94 | 9.27  | 9.20  | 6.96 |
> > > | ZeroGrad  | 19.85 | 10.38 | 10.19 | 7.89 |
> > > | MultiGrad | 19.46 | 10.65 | 10.63 | 8.33 |
> > > |   NFGSM   | 19.13 | 10.76 | 10.66 | 8.36 |
> > > |   AAER    | 19.13 | 10.76 | 10.66 | 8.36 |
> > > |   ELLE    | 21.44 | 10.71 | 10.50 | 7.92 |
> > > |   SORA    | 19.02 | 10.64 | 10.52 | 8.35 |
> > > |   PGD2    | 21.76 | 10.54 | 10.22 | 7.82 |
> > >
> > > ## PathMNIST - ViT-Small
> > > |  Method   | Clean | FGSM  |  PGD  |  AA   |
> > > |:---------:|:-----:|:-----:|:-----:|:-----:|
> > > |  Benign   | 84.83 | 9.30  | 1.69  | 1.43  |
> > > |   FGSM    | 61.53 | 44.83 | 43.62 | 39.32 |
> > > |  FGSM-RS  | 69.44 | 48.76 | 44.87 | 41.59 |
> > > | GradAlign | 67.44 | 47.51 | 44.85 | 40.67 |
> > > |   NuAT    | 51.03 | 45.88 | 45.68 | 44.43 |
> > > | ZeroGrad  | 67.48 | 46.35 | 43.41 | 38.15 |
> > > | MultiGrad | 63.04 | 45.70 | 42.35 | 39.96 |
> > > |   NFGSM   | 61.36 | 46.27 | 45.28 | 42.77 |
> > > |   AAER    | 59.72 | 44.25 | 43.29 | 40.33 |
> > > |   ELLE    | 69.18 | 46.16 | 43.22 | 39.46 |
> > > |   SORA    | 63.12 | 45.97 | 43.82 | 41.06 |
> > > |   PGD2    | 70.32 | 47.55 | 44.97 | 42.01 |

---

### Official Review · Reviewer_Ccgn · 2026-03-08

**Soundness:** 2
**Presentation:** 3
**Significance:** 2
**Originality:** 2
**Overall Recommendation:** 4
**Confidence:** 4

**Summary:**

The authors propose an efficient metric to measure the non-linearity of loss landscape during adversarial training. By utilizing this metric, the optimal step size for adversarial attack can be calculated. The experiments on diverse datasets and architectures demonstrate the effectiveness of the proposed method.

**Compliance With Llm Reviewing Policy:**

Affirmed.

**Final Justification:**

My concerns have been addressed. However, considering the comments from other reviewers, I will maintain my score.

**Key Questions For Authors:**

See weakness

**Limitations:**

yes

**Strengths And Weaknesses:**

**Strengths**
1. The theoretical analysis on PerAlign is rigorous, defining it as the cosine similarity between the input gradient used for adversarial generation and the backpropagation gradient.
2. The authors evaluate SORA across a diverse set of datasets like CIFAR-10/100, ImageNet-100, PATHMNIST and TISSUEMNIST and various architectures, including ResNet, PreActResNet, WideResNet, and SENet.
3. The visualization is useful. The use of loss landscape geometry plots provides intuitive insights into how "Epsilon Overfitting" occurs.

**Weaknesses**
1. **Limited Novelty:** The core mechanism of measuring gradient alignment to detect loss-surface distortion is conceptually similar to GradAlign. Although the authors argue that PertAlign is more computationally efficient ("negligible overhead") and can anticipate CO earlier than existing indicators, the fundamental reliance on gradient directions may limit the perceived technical leap from prior method.
2. **Domain Generalization:** Apart from standard datasets like CIFAR-10 and ImageNet, medical datasets (PATHMNIST and TISSUEMNIST) are included in the evaluation.  The author should explain why these datasets were chosen over other domains. For instance, datasets like **CelebA** (facial recognition) or **Road Sign Recognition** (autonomous driving) would provide more direct evidence of the method's utility in common real-world adversarial settings.
3. **Marginal Performance Gains against Strong Attacks:** The improvement in robust accuracy against AutoAttack (AA), which is regarded as the strongest attack, is sometimes marginal when compared to strong baselines such as AAER. The author should provide stronger experimental results or explanations on this point.

---

> ### Author Rebuttal · Authors · 2026-03-27
>
> Dear Reviewer Ccgn,
>
> Thank you for your valuable insights and feedback. We appreciate that you found our theoretical analysis rigorous and our experimental evaluations diverse across datasets and architectures.
>
> ## W1
>
> Gradients are a natural candidate for monitoring training dynamics because they inherently encode critical information about the current state of the model. While incorporating additional features could potentially provide more information, their computational cost typically grows proportionally, which discourages their use in this setting where speed and efficiency are crucial. Our motivation was therefore to extract richer information from gradient-based features while maintaining the computational efficiency of fast adversarial training. In particular, PertAlign leverages gradient information in a way that remains extremely lightweight.
>
> Although both PertAlign and GradAlign are similarity metrics based on gradients, they differ in several important aspects. As discussed in Appendix C.2, beyond the computational cost difference you noted, PertAlign differs from GradAlign in two key ways:
>
> - **Motivation**: GradAlign seeks to increase gradient alignment, whereas our method focuses on measuring local loss‑surface linearity.
>
> - **Usage**: GradAlign acts as a training regularizer, while PertAlign serves only as a lightweight monitoring metric that does not alter training.
>
> ## W2
>
> Our dataset selection criteria are described in Appendix K.3. We selected datasets with scales comparable to established benchmarks, both in terms of sample size and image resolution. Additionally, medical imaging datasets were prioritized for two main reasons:
>
> - Robustness in medical applications has significant real-world importance.
> - The distribution differences between natural and medical images provide a more challenging test of generalization.
>
> An overview of all datasets used in our work is provided in Appendix D and Table 2. The datasets cover a wide range of characteristics: natural and medical images, color and grayscale images, 8 to 200 classes, balanced and imbalanced class distributions, image resolutions ranging from $28\times28$ to $224\times224$, and training set sizes from 50,000 to 165,466 samples.
>
> Datasets such as CelebA and Road Sign Recognition are indeed interesting for studying domain generalization. However, to the best of our knowledge, most related work in fast adversarial training focuses primarily on single-domain datasets such as CIFAR and ImageNet. Therefore, we placed particular emphasis on evaluating performance on similar single-domain benchmarks for fair comparison. At the same time, unlike prior work, we additionally included several MedMNIST datasets, which we believe better expose the limitations of existing methods. Nevertheless, we agree that domain generalization experiments would be valuable and will attempt to include them in the camera-ready version.
>
> ## W3
>
> As discussed in the Introduction, an effective solution to CO should ideally satisfy robustness across datasets and architectures with hyperparameters that are largely setting-agnostic.
>
> While AAER (and some other baselines) can perform comparably to SORA in certain settings, they do not consistently remain competitive across the wide range of configurations we evaluated. As illustrated in Figure 5, no other baseline remains on top.
>
> We attribute the marginal performance gaps of some baselines, such as AAER in certain settings, to two main factors:
>
> - **Dataset overfitting**: As discussed in Appendix K.2 (lines 2507-2512), extensive evaluation on a limited set of datasets can create a misleading sense of progress.
>
> - **Hyperparameter sensitivity**: Some baselines rely on hyperparameters that are highly dataset-dependent and require careful tuning across multiple runs, which undermines the goal of fast adversarial training.
>
> To avoid these issues, as stated in Appendix K.2 (lines 2536-2539), we deliberately avoided dataset-specific modifications or tuning of our method. Despite this constraint, the results in Table 1 on ImageNet-100 show that SORA achieves 9\% higher clean accuracy and 1.38\% higher robust AA accuracy compared to AAER.
>
> Furthermore, Appendix G shows through grid searches that several baselines either fail to achieve competitive performance or require extensive tuning, with AAER failing to train robust models even across its full recommended hyperparameter range (Table 13). Consistently, on PathMNIST (Table 1), AAER still fails to produce a robust model, while SORA attains a 30.43\% higher AA robust accuracy.
>
> The limitations of other baselines are also discussed in Appendix G. On unseen datasets, these limitations often require multiple training runs to avoid CO or reach acceptable accuracy. This defeats the purpose of fast adversarial training and can make multi‑step methods more practical.

---

> > ### Author Rebuttal · Reviewer_Ccgn · 2026-04-01
> >
> > Thanks for the rebuttal. My concerns have been addressed. However, considering the comments from other reviewers, I will remain my score.

---

> > > ### Author Response · Authors · 2026-04-03
> > >
> > > We are glad that all of your concerns were resolved. Please know that we look forward to addressing any further concerns or questions regarding our work.

---

### Official Review · Reviewer_pRMd · 2026-03-13

**Soundness:** 2
**Presentation:** 2
**Significance:** 3
**Originality:** 3
**Overall Recommendation:** 4
**Confidence:** 4

**Summary:**

This paper proposes an adversarial training approach to address catastrophic overfitting caused by epsilon overfitting. First, the authors introduce PertAlign, a metric designed to predict catastrophic overfitting without incurring additional computational overhead. Second, the authors propose SORA for adversarial training, which dynamically adjusts the step size from a second-order optimization perspective. Experimental results on a variety of benchmark datasets and model architectures demonstrate that the proposed SORA approach achieves superior efficiency while maintaining competitive performance compared with state-of-the-art methods.

**Compliance With Llm Reviewing Policy:**

Affirmed.

**Final Justification:**

I thank the authors for their rebuttal. My concerns have been adequately addressed, and I maintain my original positive score.

**Key Questions For Authors:**

1. Could the authors provide additional analysis reflecting the curvature magnitude or the smoothness of the loss function to further justify the use of the PertAlign metric and the second-order modeling assumption?

2. Could the authors include more recent methods (e.g., [1]) in the experimental comparison and provide a more comprehensive comparison in Table 1 by including methods such as [1], MultiGrad, and ELLE?

3. Could the authors further elaborate on the validity of the condition $\alpha \ll 1$ in practical training settings, or analyze the performance of the method when this condition is not strictly satisfied?

[1] C. Pan, K. Tang, Q. Li and X. Yao, “Mitigating Catastrophic Overfitting in Fast Adversarial Training via Label Information Elimination,” in Proc. IEEE/CVF International Conference on Computer Vision, 2025

**Limitations:**

No. This paper should include a dedicated section discussing the limitations of the proposed method.

**Strengths And Weaknesses:**

Strengths:

1. This paper proposes PertAlign, a metric that can effectively predict the occurrence of catastrophic overfitting without incurring additional computational overhead.

2. The paper introduces SORA, an adaptive step-size adversarial training method that effectively addresses the catastrophic overfitting issue.

3. The theoretical analysis appears reasonable and helps justify the proposed adversarial training mechanism.

Major Weakness:

1. The methodological approach lacks rationality. As shown in Figure 2, the accuracy rate exhibits a smooth yet non-monotonic behavior when $\epsilon$ is small, prompting the authors to employ a second-order model to approximate the region near $\epsilon=0$. However, the simple accuracy curve fails to indicate whether the loss function is smooth or to quantify its curvature. Further elaboration is required to characterize other features that can reflect the magnitude of curvature.

2. The theoretical assumptions may lack practical considerations. In Lemma 4.1, the authors state that PertAlign can be approximated when $\alpha \ll 1$, However, it is not entirely clear whether the step size $\alpha$ can satisfy this condition in practical settings, and its practical interpretation or determination remains unclear.

3. The method explanation is insufficiently detailed. In Algorithm 1, the update of $v$ in line 10 is not clearly specified in the method section.

4. The comparative analysis lacks comprehensiveness. First, the authors should consider including more recent fast adversarial training (AT) methods, such as [1], in the comparison. In addition, although three adversarial training methods are compared in Table 1, high-accuracy methods such as MultiGrad and ELLE, which demonstrate superior performance in Figure 6, are not included in Table 1. This omission limits the completeness of the comparative evaluation and makes it difficult to fully assess the performance advantages of the proposed method.

[1] C. Pan, K. Tang, Q. Li and X. Yao, “Mitigating Catastrophic Overfitting in Fast Adversarial Training via Label Information Elimination,” in Proc. IEEE/CVF International Conference on Computer Vision, 2025.

Minor Weakness:

1. In line 215 of Section 4, there is an unexpected line break after the word “and”.

2. In Algorithm 1, the definitions of $\eta$ in line 3 and $\eta_i$ in line 6 are inconsistent.

3. The manuscript extensively cites arXiv preprints in the references without citing the corresponding published versions when available.

---

> ### Author Rebuttal · Authors · 2026-03-27
>
> Dear Reviewer pRMd,
>
> Thank you for your thoughtful and detailed feedback. We appreciate that you found PertAlign to be an effective metric for predicting CO with negligible computational overhead and SORA to be an effective method for addressing the CO issue. Below, we address the concerns and questions you raised.
>
> ## W1 and Q1
>
> We agree that the smoothness of the accuracy curve alone may leave some uncertainty regarding the smoothness of the loss landscape. Therefore, in our experiments we also tracked the evolution of the loss landscape during CO by visualizing the loss values along the FGSM and PGD directions for different perturbation magnitudes. We used the visualization code from [1].
>
> The loss landscape visualizations are shown per epoch alongside the accuracy plot and are available at [URL](https://drive.google.com/file/d/1BshF0WqNxYm1rOAPbMRjMylFin3e4nNt/view?usp=sharing). Each unit along the FGSM and PGD axes corresponds to $\varepsilon$. By examining the FGSM axis when PGD $=0$, we observe a non-jagged (smooth) and non-monotonic behavior of the loss distortion after the occurrence of CO, which corresponds to the behavior observed in the accuracy curve in Figure 2.
>
> In the paper, we used the accuracy plot together with the abstract loss visualization in Figure 1 instead of the empirical loss landscapes to improve conceptual clarity. However, we will include these empirical visualizations in the camera-ready version.
>
> ## W2 and Q3
>
> Although in single-step attacks the step size is often larger than $\varepsilon$, the imperceptibility property of adversarial examples [2] limits how large the step size can practically be. Moreover, since the remainder term in the Taylor expansion is of order $\mathcal{O}(\alpha^3)$, it remains small even when the step size is moderately larger than $\varepsilon$.
>
> For example, when $\alpha = 16/255$, the error term is on the order of $3 \times 10^{-4}$, which is much smaller than $1$. In practice, PertAlign behaves reliably across all of our experiments, with example samples shown in Appendix I.
>
> ## W3
>
> The parameter $v$ denotes the scalar value $\frac{p^{T} g'}{\|g\|_1}$ used in the denominator of SORA's adaptive step size selection. Intuitively, $v$ correlates with the local curvature of the loss landscape in the direction of the perturbation ($v < 1$ indicates negative curvature, $v = 1$ approximately zero curvature, and $v > 1$ positive curvature). Values closer to $1$ therefore indicate a more locally linear loss surface.
>
> To reduce noise and stabilize this quantity, we update it using an exponential moving average, as shown in line 10 of Algorithm 1. We will further clarify this explanation in the methodology section to remove any ambiguity.
>
> ## W4 and Q2
>
> Thank you for the baseline recommendation.
>
> Excluding SORA, we included 9 single-step methods, 3 multi-step methods, and a benign training baseline, all published in top-tier venues. Our goal was to include both competitive recent methods such as NFGSM (NeurIPS 2022), AAER (NeurIPS 2024), and ELLE (ICLR 2024), as well as established methods such as FGSM-RS (ICLR 2020), GradAlign (NeurIPS 2020), and ZeroGrad / MultiGrad (Elsevier 2021) for single-step training, and TRADES (ICML 2019) for multi-step training, in addition to the standard FGSM and PGD methods.
>
> We also included ATAS (IEEE TIP 2022), which is another adaptive step-size method but based on a different mechanism. Given the large number of methods proposed in this area and our goal of evaluating across many dataset–architecture combinations, to be as diverse and extensive as possible.
>
> We also reviewed the work you mentioned. The LIET method is based on an interesting idea using class-label information; however, it requires approximately $30$ percent more training time than SORA and additional memory proportional to the number of classes, whereas SORA only requires negligible constant memory similar to FGSM. Thus, SORA is more efficient in both runtime and memory. Nevertheless, we are committed to evaluating LIET within our experimental setup and comparing its accuracy with SORA.
>
> Regarding ELLE and MultiGrad, these methods incur substantially higher computational and memory costs than the fast methods listed in Table 1, which makes evaluation on a high-resolution dataset such as ImageNet-100 infeasible with our available resources (NVIDIA RTX 4090 GPU). However, we evaluated them on all other datasets, including PathMNIST and TinyImageNet, as reported in Appendix J. Since the distribution of ImageNet-100 is closer to TinyImageNet and CIFAR-style datasets, we compared SORA with the strongest baselines from those datasets
>
> ## Minor Weaknesses
>
> Thank you for pointing out these typographic errors. We will correct them in the camera-ready version of the manuscript.
>
> ## References
>
> [1] Understanding Catastrophic Overfitting in Single-step Adversarial Training, AAAI 2021.
>
> [2] Intriguing Properties of Neural Networks, ICLR 2014.

---

> > ### Author Rebuttal · Reviewer_pRMd · 2026-04-04
> >
> > I thank the authors for their detailed responses to the raised weaknesses and questions, as well as for providing additional visualizations. My concerns have been adequately addressed. Taking into account other reviewers’ concerns, I will maintain my original score.

---

> > > ### Author Response · Authors · 2026-04-04
> > >
> > > We are glad all of your concerns were resolved and that our visualizations were helpful. Please know that we are more than happy to address any further concerns or questions regarding our work.

---

### Decision · Program_Chairs · 2026-04-30

**Decision:**

Accept (regular)

**Comment:**

This paper received mixed original scores (4, 4, 2, 2). The authors have provided a comprehensive rebuttal and follow-up replies to the reviewers' concerns. The two reviewers who rated 4 have selected "fully resolved" in their Rebuttal Acknowledgement, while the other two reviewers who rated 2 have disappeared and did not consider the follow-up replies in their final justifications. The AC has carefully checked the remaining concerns by those two reviewers and believes that the authors' follow-up replies are satisfactory. Specifically, Reviewer 3 suggests adding an explicit gradient obfuscation test as in [6] (which does not seem to be a severe problem for the AC), and the authors have replied that almost all prior related work does not do so, but only considers AA. To the best of the knowledge of the AC, AT-based methods do not usually suffer from gradient masking.

For Reviewer 4, the authors have provided additional AA results to address their remaining concern regarding AA. It is also worth noting that both these two reviewers ground their rejection on other reviewers' concerns; however, it is clear that the other two reviewers have agreed to accept the paper.